# DNA methylation atlas of the mouse brain at single-cell resolution

Hanqing Liu[1,2,16], Jingtian Zhou[1,3,16], Wei Tian[1], Chongyuan Luo[1,4], Anna Bartlett[1], Andrew Aldridge[1], Jacinta Lucero[5], Julia K. Osteen[5], Joseph R. Nery[1], Huaming Chen[1], Angeline Rivkin[1], Rosa G. Castanon[1], Ben Clock[6], Yang Eric Li[7], Xiaomeng Hou[8,9,10,11], Olivier B. Poirion[8,9,10,11], Sebastian Preissl[8,9,10,11], Antonio Pinto-Duarte[5], Carolyn O'Connor[12], Lara Boggeman[12], Conor Fitzpatrick[12], Michael Nunn[1], Eran A. Mukamel[13], Zhuzhu Zhang[1], Edward M. Callaway[14], Bing Ren[7,8,9,10,11], Jesse R. Dixon[6], M. Margarita Behrens[5] & Joseph R. Ecker[1,15 ✉]

Mammalian brain cells show remarkable diversity in gene expression, anatomy and function, yet the regulatory DNA landscape underlying this extensive heterogeneity is poorly understood. Here we carry out a comprehensive assessment of the epigenomes of mouse brain cell types by applying single-nucleus DNA methylation sequencing[1,2] to profile 103,982 nuclei (including 95,815 neurons and 8,167 non-neuronal cells) from 45 regions of the mouse cortex, hippocampus, striatum, pallidum and olfactory areas. We identified 161 cell clusters with distinct spatial locations and projection targets. We constructed taxonomies of these epigenetic types, annotated with signature genes, regulatory elements and transcription factors. These features indicate the potential regulatory landscape supporting the assignment of putative cell types and reveal repetitive usage of regulators in excitatory and inhibitory cells for determining subtypes. The DNA methylation landscape of excitatory neurons in the cortex and hippocampus varied continuously along spatial gradients. Using this deep dataset, we constructed an artificial neural network model that precisely predicts single neuron cell-type identity and brain area spatial location. Integration of high-resolution DNA methylomes with single-nucleus chromatin accessibility data[3] enabled prediction of high-confidence enhancer–gene interactions for all identified cell types, which were subsequently validated by cell-type-specific chromatin conformation capture experiments[4]. By combining multi-omic datasets (DNA methylation, chromatin contacts, and open chromatin) from single nuclei and annotating the regulatory genome of hundreds of cell types in the mouse brain, our DNA methylation atlas establishes the epigenetic basis for neuronal diversity and spatial organization throughout the mouse cerebrum.

Epigenomic dynamics are associated with cell differentiation and maturation in the mammalian brain and have an essential role in regulating neuronal functions and animal behaviour[5,6]. Cytosine DNA methylation (5mC) is a stable covalent modification that persists in post-mitotic cells throughout their lifetime and is critical for proper gene regulation[6]. In mammalian genomes, 5mC occurs predominantly at CpG sites (mCG), showing dynamic patterns at regulatory elements with tissue and cell-type specificity[1,6–8], modulating binding affinity of transcription factors[9] and controlling gene transcription[5]. Non-CpG cytosines are also abundantly methylated (mCH, H denotes A, C, or T)—uniquely in neurons—in the mouse and human brain[6,10], which can directly affect DNA binding of methyl CpG binding protein 2 (MeCP2)[11–13], causing Rett syndrome[14]. Levels of mCH at gene bodies are anti-correlated with gene expression and show high heterogeneity across neuronal cell types[1,7].

A deeper understanding of epigenomic diversity in the mouse brain provides a complementary approach to transcriptome-based profiling methods for identifying brain cell types and allows genome-wide prediction of the regulatory elements and transcriptional networks

[1]Genomic Analysis Laboratory, The Salk Institute for Biological Studies, La Jolla, CA, USA. [2]Division of Biological Sciences, University of California, San Diego, La Jolla, CA, USA. [3]Bioinformatics and Systems Biology Program, University of California, San Diego, La Jolla, CA, USA. [4]Department of Human Genetics, University of California Los Angeles, Los Angeles, CA, USA. [5]Computational Neurobiology Laboratory, The Salk Institute for Biological Studies, La Jolla, CA, USA. [6]Peptide Biology Laboratory, The Salk Institute for Biological Studies, La Jolla, CA, USA. [7]Ludwig Institute for Cancer Research, La Jolla, CA, USA. [8]Center for Epigenomics, University of California, San Diego School of Medicine, La Jolla, CA, USA. [9]Department of Cellular and Molecular Medicine, University of California, San Diego School of Medicine, La Jolla, CA, USA. [10]Institute of Genomic Medicine, University of California, San Diego School of Medicine, La Jolla, CA, USA. [11]Moores Cancer Center, University of California, San Diego School of Medicine, La Jolla, CA, USA. [12]Flow Cytometry Core Facility, The Salk Institute for Biological Studies, La Jolla, CA, USA. [13]Department of Cognitive Science, University of California, San Diego, La Jolla, CA, USA. [14]Systems Neurobiology Laboratories, The Salk Institute for Biological Studies, La Jolla, CA, USA. [15]Howard Hughes Medical Institute, The Salk Institute for Biological Studies, La Jolla, CA, USA. [16]These authors contributed equally: Hanqing Liu, Jingtian Zhou. ✉e-mail: ecker@salk.edu

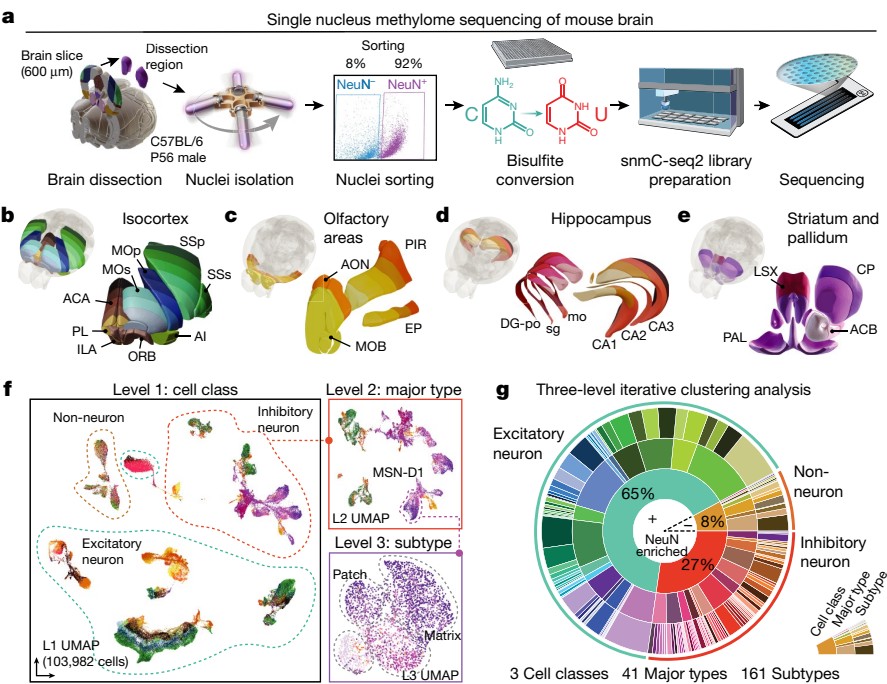

**Fig. 1 | A survey of single-cell DNA methylomes in the mouse brain. a**, The workflow of dissection, FANS and snmC-seq2 sequencing. **b**–**e**, Dissected regions of isocortex (**b**), OLF (**c**), HIP (**d**) and CNU (**e**). **f**, Three-level UMAP from iterative analysis, colour coded as in **b**–**e**, panels show an example in which MSN-D1 neurons are separated into subtypes. **g**, Proportions of cells in clusters defined in the three-level iterative analysis. Brain atlas images in **a**–**d** were created based on Wang et al.[16] and © 2017 Allen Institute for Brain Science. Allen Brain Reference Atlas. Available from: atlas.brain-map.org.

underlying this diversity. Previous studies have demonstrated the utility of studying brain cell types and regulatory diversity using single-nucleus methylome sequencing (snmC-seq)[1]. This study uses snmC-seq2[2] to perform thorough methylome profiling with detailed spatial dissection in the adult postnatal day 56 (P56) male mouse brain. In Li et al.[3], the same tissue samples were profiled using single-nucleus assay for transposase-accessible chromatin using sequencing (snATAC-seq) to identify genome-wide accessible chromatin[15], providing complementary epigenomic information to aid in cell-type-specific regulatory genome annotation. Moreover, to further study *cis*-regulatory elements and their potential target genes across the genome, we applied single-nucleus methylation and chromosome conformation capture sequencing (sn-m3C-seq)[4] to profile the methylome and chromatin conformation in the same cells.

These epigenomic datasets provide a detailed and comprehensive census of the diversity of cell types across mouse brain regions, allowing identification of cell-type-specific regulatory elements and their candidate target genes and upstream transcription factors. Here we construct a single-cell base-resolution DNA methylation dataset containing 103,982 methylomes from 45 dissected brain regions and use an iterative analysis framework to identify 161 predicted mouse brain subtypes. Comparing subtype-level methylomes enables us to identify 3.9 million genomic regions showing cell-type-specific mCG variation, covering approximately 50% (1,240 Mb) of the mouse genome. We show that differentially methylated transcription factor genes and binding motifs can be associated with subtype taxonomy branches, allowing the prediction of cell-type gene regulatory programs specific for each developmental lineage. Integration of these data with cell clusters identified on the basis of chromatin accessibility validates most methylome-derived subtypes, enabling the prediction of 1.6 million enhancer-like genomic regions. We identify *cis*-regulatory interactions between enhancers and genes using computational prediction and single-cell chromatin conformation profiling (in the hippocampus (HIP)). We also identify spatial methylation gradients in cortical excitatory neurons and dentate gyrus granule cells and associated

transcription factors and motifs. We apply an artificial neural network (ANN) model to precisely predict single-neuron cell-type identity and brain area spatial location using its methylome profile as input and develop the brain cell methylation viewer (http://neomorph.salk.edu/omb) as a portal for querying and visualization of cell- and cluster-level methylation data.

## Single-cell DNA methylome atlas

We used snmC-seq2[2] to profile genome-wide 5mC at single-cell resolution (Fig. 1a) across the cortex, HIP, striatum and pallidum (or cerebral nuclei, CNU), and olfactory areas (OLF) (Fig. 1b–e) using adult male C57BL/6 mice[16]. In total, we analysed 45 dissected regions in two replicates (Extended Data Fig. 1, Supplementary Table 2). Fluorescence-activated nuclei sorting (FANS) of antibody-labelled nuclei was applied to capture NeuN-positive neurons (NeuN+, 92% of neurons), while also sampling a smaller number of NeuN-negative (NeuN−, 8% of neurons) non-neuronal cells (Fig. 1a). In total, we profiled the DNA methylomes of 103,982 single nuclei, yielding, on average, 1.5 million stringently filtered reads per cell ($1.50 \times 10^6 \pm 0.58 \times 10^6$, mean ± s.d.) covering 6.2 ± 2.6% of the cytosines in the mouse genome in each cell. These enabled reliable quantification of the DNA methylation fraction for 25,905 ± 1,090 (95 ± 4%) 100-kb bins and 44,944 ± 4,438 (81 ± 8%) gene bodies (Extended Data Fig. 2a). The global methylation levels range from 0.2% to 7.6% in non-CpG sites and 61.6% to 88.8% in CpG sites (Extended Data Fig. 2b, c).

On the basis of the mCH and mCG profiles in 100-kb bins throughout the genome, we performed a three-level iterative clustering analysis to categorize the epigenomic cell populations (Fig. 1f, g). After quality control and preprocessing (Methods), in the first level (cell class), we clustered 103,982 cells as 67,472 (65%) excitatory neurons, 28,343 (27%) inhibitory neurons, and 8,167 (8%) non-neurons (Supplementary Table 3). The second round of iterative analysis of each cell class identified 41 cell major types in total (cluster size range 95–11,919),

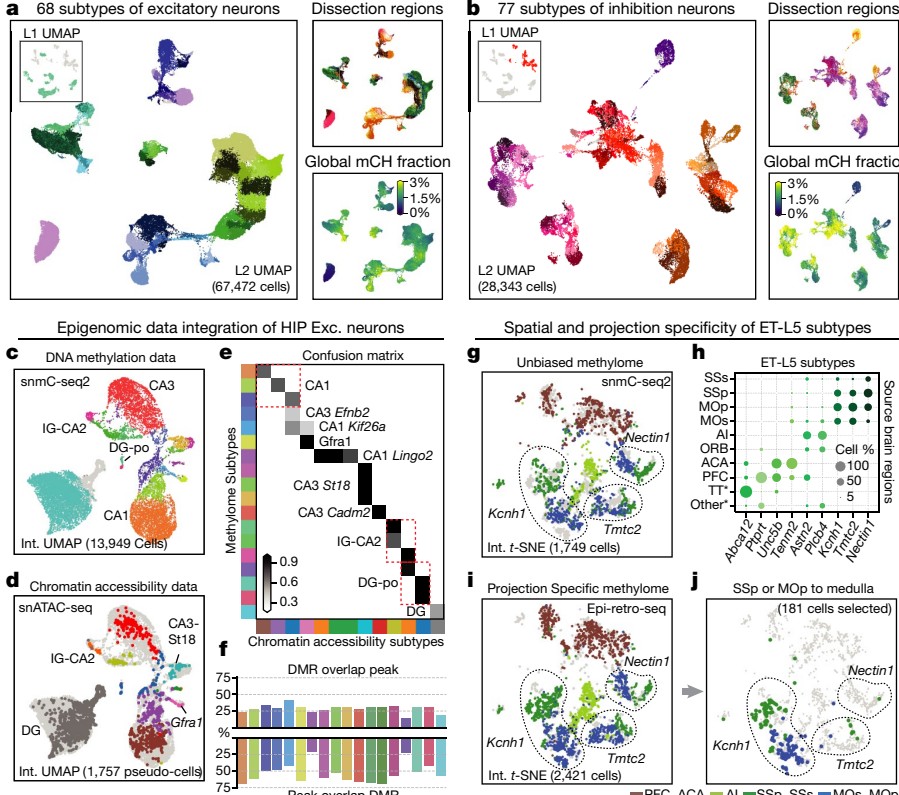

**Fig. 2 | Epigenomic diversity of neurons. a, b,** Level 2 UMAP of excitatory (**a**) and inhibitory (**b**) neurons, coloured by subtype, dissection region and global mCH fraction. **c, d,** Integration UMAP of the HIP excitatory neurons profiled by snmC-seq2 (**c**) and snATAC-seq (**d**; shows pseudo-cells). **e,** Overlap score of a-types and m-types. **f,** Overlap of CG-DMR and ATAC peaks in matched subtypes. **g, i, j,** Integration *t*-SNE of ET-L5 neurons profiled by snmC-seq2 (**g**) and epi-retro-seq (**i, j**), coloured by dissection region. Three SSp- and MOp-enriched subtypes are labelled by their marker gene. **j,** Medulla projecting neurons from SSp or MOp only. **h,** Spatial composition of ET-L5 subtypes.

and the third round separated these major types further into 161 cell subtypes (cluster size range 12–6,551). All subtypes are highly conserved across replicates, and replicates from the same brain region are co-clustered compared with samples from other brain regions (Extended Data Fig. 2d–g).

The spatial distribution of each cell type is assessed based on where the cells were dissected (Supplementary Table 5). Here we used uniform manifold approximation and projection (UMAP)[17] to visualize cell spatial locations (Fig. 1f, Extended Data Fig. 3) and major cell types (Extended Data Fig. 2h). Major non-neuronal cell types have a similar distribution across brain regions (Extended Data Fig. 1g), except adult neuron progenitors (ANPs). We found two subtypes of ANPs, presumably corresponding to neuronal precursors in the subgranular zone of the dentate gyrus (DG)[18] (ANP anp-dg) and the rostral migratory stream[18] in CNU and OLF (ANP anp-olf-cnu). Excitatory neurons from isocortex, OLF and HIP formed different major types, with some exceptions, potentially owing to overlaps in dissected regions (Supplementary Table 2). Cells from the isocortex were further separated on the basis of their projection types[1,19,20]. The intratelencephalic (IT) neurons from all cortical regions contain four major types corresponding to the laminar layers (L2/3, L4, L5 and L6), each of which includes cells from all cortical regions, except L4, which lack cells from the prefrontal cortex (PFC) and anterior cingulate area (ACA). Excitatory neurons from the HIP were further partitioned into major types corresponding to DG granule cells and different subfields of cornus ammonis (CA). We also identified major types from cortical subplate structures, including the claustrum (CLA) and endopiriform nucleus (EP) from isocortex and OLF dissections. GABAergic inhibitory neurons from isocortex and HIP cluster

together into five major types, whereas interneurons from CNU and OLF group into nine major types.

In total, we identified 68 excitatory and 77 inhibitory subtypes (Fig. 2a, b, Supplementary Table 7). Although there is no one-to-one correspondence between subtypes and brain regions, individual subtypes show differential regional enrichment (Fig. 2a, b, top right) and distinct global mCH levels, ranging from 0.98% (DG dg-all) to 4.64% (PAL-Inh Chat, an inhibitory subtype in pallidum (PAL)) (Fig. 2a, b, bottom right). Specifically, isocortical excitatory subtypes usually consist of cells majorly derived from either the sensorimotor (primary motor (MOp), secondary motor (MOs), primary somatosensory (SSp), and secondary somatosensory (SSs) cortex), medial (PFC and ACA), or frontal areas (orbital (ORB) and agranular insular (AI) area). In the OLF, excitatory cells from the anterior olfactory nucleus (AON) and main olfactory bulb (MOB) are enriched in the subtype OLF-Exc *Bmpr1b*, whereas cells from the piriform area (PIR) are relatively enriched in the other OLF-Exc subtypes. Similarly, some inhibitory subtypes in CNU and OLF also correspond to different substructures in these two regions (Supplementary Note 1), indicating substantial spatial-related methylation diversity among CNU and OLF interneurons. By contrast, most caudal (CGE) or medial (MGE) ganglionic eminence-derived inhibitory subtypes contain cells derived predominantly from all cortical or hippocampal regions. To better demonstrate the unprecedented level of neuronal subtype and spatial diversity in their DNA methylomes, we provide a web application to interactively display this information at different granularity (http://neomorph.salk.edu/omb). We also provide a detailed discussion of how exemplified subtypes correspond to cell types with known functional and spatial features (Extended Data Fig. 4) in Supplementary Note 1.

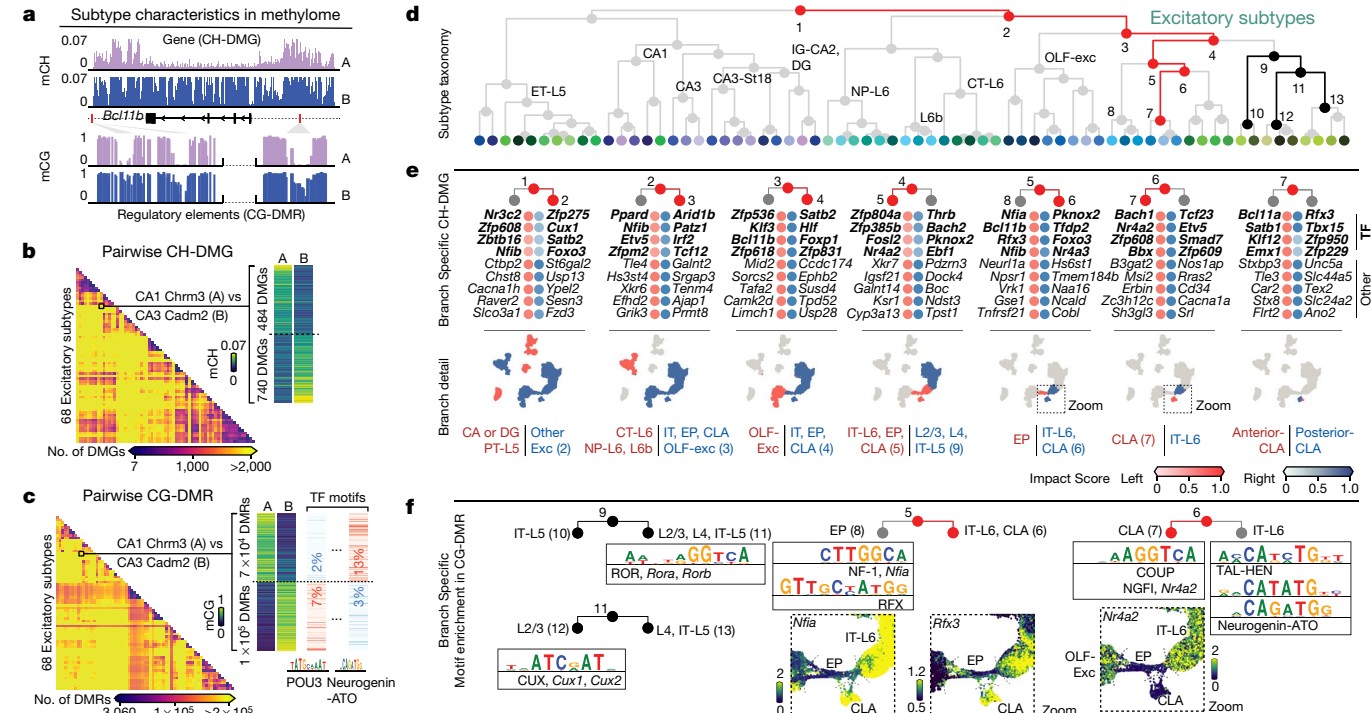

**Fig. 3 | Relating genes and regulatory elements to cell subtype taxonomy.**
**a**, Schematic of the two characteristics contained in the methylome profiles.
**b, c**, Pairwise CH-DMG (**b**) and CG-DMR (**c**) counts between 68 excitatory subtypes. **c**, In each CG-DMR set, we further identify differentially enriched motifs (left). TF, transcription factor. **d**, Excitatory subtype taxonomy tree. **e**, Top impact scores of ranked genes for the left and right branches of nodes

1–7 in **d**. The top four genes are transcription factor genes (bold); these are followed by other protein-coding genes. The scatter plots below show cells involved in each branch. **f**, Branch-specific transcription factor motif families. The zoomed UMAPs show individual transcription factor genes in those families, whose differential mCH fractions are concordant with their motif enrichment.

## Consensus epigenomic profiles

Integrating single-cell datasets collected using different molecular profiling modalities can help to establish a consensus cell-type atlas[20,21]. By integrating the methylome data with the chromatin accessibility data profiled using snATAC-seq on the same brain samples from a parallel study[3], the two modalities validated each other at the subtype level (Fig. 2c, d, Extended Data Fig. 5a–f, Supplementary Table 10). We then calculated overlap scores between the original methylation subtypes (m-types) and the chromatin accessibility subtypes (a-types), which further quantified the matching of subtypes between the two modalities (Fig. 2e, Extended Data Fig. 5e, Methods). Moreover, the mCG DMRs (see below) highly overlap with open chromatin peaks in the hippocampal subtypes (Fig. 2f). Their mCG fractions and chromatin accessibility levels show similar cell-type-specificity across hippocampal subtypes, confirming the correct match of cell-type identities (Extended Data Fig. 5f).

## Projection specificity of ET-L5 neurons

To further infer the projection targets of cell subtypes, we integrated our extra-telencephalic (ET) L5 neurons with epi-retro-seq data[22]. Epi-retro-seq uses retrograde viral labelling to select neurons projecting to specific brain regions, followed by methylome analysis of their epigenetic subtypes. Cells from the same brain region of the two datasets are colocalized on t-distributed stochastic neighbour embedding (t-SNE) analysis, validating the subtypes' spatial distribution (Fig. 2g–i, Extended Data Fig. 5g–i). The overlap scores between unbiased (snmC-seq2) and targeted (epi-retro-seq) profiling experiments (Extended Data Fig. 5j) indicate that some subtypes identified from the same cortical area show different projection specificity. For

example, SSp and MOp neurons were mainly enriched in three subtypes marked by *Kcnh1*, *Tmtc2* and *Nectin1*, respectively. However, neurons projecting to the medulla in the MOp and SSp only integrate with the subtype marked with *Kcnh1* (Fig. 2j), suggesting that the subtypes identified in unbiased methylome profiling have distinct projection specificities.

## Regulatory taxonomy of neuronal subtypes

Having developed a consensus map of cell types based on their DNA methylomes, we identified 16,451 differentially CH-methylated genes (CH-DMGs) and 3.9 million CG-differentially methylated regions (CG-DMRs, 624 ± 176 base pairs (bp) mean ± s.d.) between the subtypes (Extended Data Fig. 6, Methods, Supplementary Note 2). snmC-seq2 captures both cell-type-specific gene expression and predicted regulatory events[1,2]. Specifically, both gene body mCH and mCG negatively correlate with gene expression in neurons, with mCH showing a stronger correlation than mCG[1,6,7,13]. CG-DMRs provide predictions about cell-type-specific regulatory elements and transcription factors whose motifs enriched in these CG-DMRs predict the crucial regulators of the cell type[1,7,8].

To further explore the gene regulatory relationship between neuronal subtypes, we constructed taxonomy trees for excitatory and inhibitory subtypes, based on gene body mCH of CH-DMGs (Extended Data Fig. 7a, b, Methods). The dendrogram structures represent the similarities between these discrete subtypes and may reflect the developmental history of neuronal type specification[19,23]. Next, we used both CH-DMGs and CG-DMRs to annotate the tree and explore the features specifying cell subtypes (excitatory in Fig. 3a–c and inhibitory in Extended Data Fig. 7c, d). Specifically, we calculated a branch-specific methylation impact score for each gene or transcription factor motif

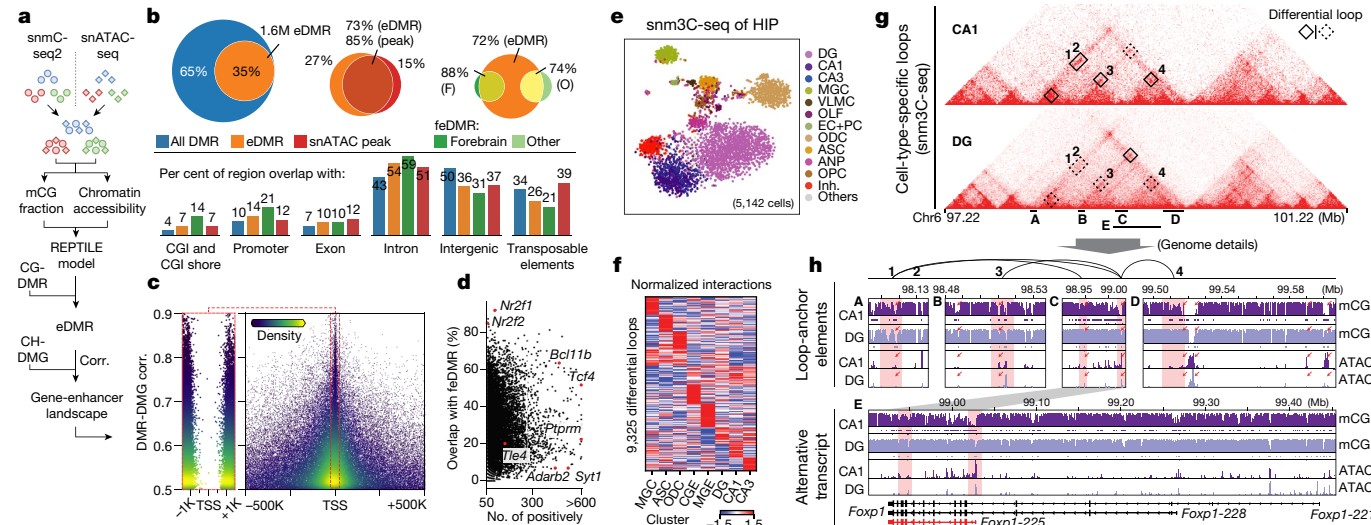

**Fig. 4 | Gene–enhancer landscapes in neuronal subtypes. a**, Schematic of enhancer calling using matched DNA methylome and chromatin accessibility subtype profiles. Corr., correlation. **b**, Overlap of regulatory elements identified in this study and other epigenomic studies (ATAC peaks[3] and feDMRs[8]). **c**, DMR–DMG correlation and the distance between DMR centre and gene TSS; each point is a DMR–DMG pair coloured by kernel density. **d**, Percentage of positively correlated eDMRs that overlap with forebrain feDMRs in each gene. **e**, $t$-SNE of cells analysed by sn-m3C-seq coloured by assigned major cell types. **f**, Interaction level ($z$-score across rows and columns) of differential loops in eight clusters at 25-kb resolution. **g**, **h**, Epigenomic signatures surrounding *Foxp1*. **g**, Triangle heat maps showing CA1 and DG chromatin contacts and differential loops. **h**, Genome browser sections showing detailed mCG and ATAC profiles near anchors of four CA1-specific loops. Red rectangles indicate loop anchors and red arrows indicate notable regulatory elements.

that summarizes all of the pairwise comparisons related to that branch (Extended Data Fig. 7e; Methods). The impact score ranges from 0 to 1, with a higher score predicting stronger functional relevance to the branch. We assign 6,038 unique genes to branches within the excitatory taxonomy (5,975 in inhibitory taxonomy), including 406 transcription factor genes (412 in inhibitory taxonomy) using genes with impact scores greater than 0.3. For example, motifs from the ROR (also known as NR1F) family were assigned to the branch that separates superficial layer IT neurons from deeper layer IT neurons (Fig. 3d–f, node 9), whereas motifs from the CUX family were assigned to the IT-L2/3 branch, separating it from IT-L4/5 neurons (Fig. 3d–f, node 11). Both of these families contain members, such as *Cux1*, *Cux2* and *Rorb*, that show laminar expression in the corresponding layers and regulate cortical layer differentiation during development[19].

After impact score assignment, each branch of this taxonomy was associated with multiple transcription factor genes and motifs, which potentially function in combination to shape cell-type identities[24] (Fig. 3e, f). For example, we focused on two brain structures of interest: the CLA and the EP[25,26]. At the major-cell-type level, distinct clusters are marked by *Npsr1* (EP) and *B3gat2* (CLA). The known EP and CLA marker transcription factor *Nr4a2*[25] also shows hypomethylation in both clusters compared to other clusters. Accordingly, the NR4A2 motif is also associated with a branch that splits CLA neurons from IT-L6 neurons (Fig. 3d–f, node 6). On another branch separating EP from CLA and IT-L6 neurons, genes for several transcription factors, including NF-1 family members *Nfia* and *Nfib* and the RFX family member *Rfx3*, together with corresponding motifs (Fig. 3d–f, node 5) rank near the top. Our findings suggest that these transcription factors may function together with *Nr4a2*, potentially separating EP neurons from CLA and IT-L6 neurons.

Beyond identifying specific cell-subtype characteristics, we derived total impact (TI) scores to summarize the methylation variation of genes and motifs to understand their relative importance in cell type diversification and function (Extended Data Fig. 7f–k, Supplementary Note 3). By comparing the TI scores of genes and motifs calculated from the inhibitory and excitatory taxonomies, we found that there were

more transcription factor genes and motifs having large TI scores in both cell classes than in either one or the other (Extended Data Fig. 7f, i). For instance, *Bcl11b* distinguishes OLF-Exc and IT neurons in the excitatory lineage and distinguishes CGE-*Lamp5* and CGE-*Vip* in the inhibitory lineage. Similarly, *Satb1* separates IT–L4 from IT-L2/3 and MGE from CGE in excitatory and inhibitory cells. These findings indicate broad repurposing of transcription factors for cell-type specification among distinct developmental lineages.

## Enhancer–gene Interactions

To systematically identify enhancer-like regions in specific cell types, we predicted enhancer-DMRs (eDMR) by integrating matched DNA methylome and chromatin accessibility profiles[3] of 161 subtypes (Fig. 4a, Methods). We identified 1,612,198 eDMR (34% of CG-DMRs), 73% of which overlapped with separately identified snATAC-seq peaks (Fig. 4b). Fetal-enhancer DMRs (feDMR) (that is, eDMRs between development time points) of forebrain bulk tissues[8] show high (88%) overlap with eDMRs. Surprisingly, the eDMRs also cover 74% of the feDMRs from other fetal tissues[8], indicating extensive reuse of enhancer-like regulatory elements across mammalian tissue types (Fig. 4b).

Next, we examined the relationship between the cell-type-signature genes and their potential regulatory elements. We calculated the partial correlation between all DMG–DMR pairs within 1 Mb distance using methylation levels across 145 neuronal subtypes (Methods). We identified a total of 1,038,853 (64%) eDMRs that correlated with at least one gene (correlation >0.3 with empirical $P < 0.005$, two-sided permutation test, Extended Data Fig. 8a). Notably, for those strongly positive-correlated DMR–DMG pairs (correlation >0.5), the DMRs are largely (63%) within 100 kb of the transcription start sites (TSSs) of the corresponding genes but are depleted from ±1 kb (Fig. 4c, Extended Data Fig. 8b), whereas for the negatively correlated DMR–DMG pairs, only 11% of DMRs are found within 100 kb of the TSS (Extended Data Fig. 8c).

Using the gene–enhancer interactions predicted by this correlation analysis, we assigned eDMRs to their target genes. The percentages

of feDMR-overlapping eDMRs vary markedly among genes (Fig. 4d, Extended Data Fig. 8d, e). Of note, DMRs assigned to the same gene show different mCG specificity among subtypes. For example, *Tle4*-correlated eDMR could be partitioned into three groups (Extended Data Fig. 8e–g). One group (G2) of elements that displayed little diversity in bulk data showed highly specific mCG and open-chromatin signals in MSN–D1/D2 neurons, whereas another group (G3) was specific to CT–L6 neurons. These two groups of DMRs suggest that possible alternative regulatory elements are used to regulate the same gene in different cell types, although further experiments are required to validate this hypothesis.

Together, these analyses allow us to carefully chart the specificity of regulatory elements identified in bulk tissues to the subtype level. Besides, we identified many regulatory elements that show more restricted specificity (for example, eDMRs correlated with *Tle4* in MSN–D1/D2), providing abundant candidates for further pursuing enhancer-driven adeno-associated viruses (AAVs) that target highly specific cell types[27].

## 3D genome structure of hippocampus

Distal enhancers typically regulate gene expression through physical interaction with promoters[28]. Therefore, to examine whether physical chromatin contacts support our correlation-based predictions of enhancer–gene associations, we generated sn-m3C-seq[4] data for 5,142 single nuclei from the HIP (152,000 contacts per cell on average). We assigned these cells, on the basis of the sn-m3C-seq data, to eight major cell types based on integration with the snmC-seq2 HIP data. In total, 19,151 chromosome loops were identified in at least one of the cell types at 25-kb resolution (range from 1,173 to 12,614 chromosome loops per cell type).

Using DG and CA1 as examples, a notably higher correlation was observed between enhancers and genes at loop anchors than between random enhancer–gene pairs (Extended Data Fig. 8h). Reciprocally, the enhancer–gene pairs showing stronger correlation with methylation were more likely to be found linked by chromosome loops or within the same looping region (Extended Data Fig. 8i). We also compared the concordance of methylation patterns between genes and enhancers linked by different methods and found the pairs linked by loop anchors or closest genes had the highest correlation of methylation (Extended Data Fig. 8j). Together, these analyses validate the physical proximity of enhancer–gene pairs predicted by our correlation-based method in specific cell types.

Additionally, we observed significant cell-type-specific 3D genome structures. The major cell types could be distinguished on UMAP embedding on the basis of chromosome interaction (Fig. 4e), indicating the dynamic nature of genome architecture across cell types. Among the 19,151 chromosome loops, 48.7% showed significantly different contact frequency between cell types (Fig. 4f). eDMRs were highly enriched at these differential loop anchors (Extended Data Fig. 8k). mCG levels at distal *cis*-elements are typically anti-correlated with enhancer activity[8]. Thus, we hypothesized that enhancers at differential loop anchors might also be hypomethylated in the corresponding cell type. Indeed, using the loops identified in DG and CA1 as examples, we observed that enhancers at the anchor of cell-type-specific loops show corresponding hypomethylation in the same cell type that the loop is specific to (Extended Data Fig. 8l).

Many differential loops were observed near marker genes of the corresponding cell type. For example, *Foxp1*, a gene for a CA1-specific transcription factor[29], has chromosome loops surrounding its gene body in CA1 but not DG (Fig. 4g, h). eDMRs and open chromatin were observed at these loop anchors. Notably, three loops in CA1 anchored at the TSS of the same transcript of *Foxp1* (Fig. 4h). Stronger demethylation and chromatin accessibility were also observed at the same transcript than in other transcripts (Fig. 4h, box E). These epigenetic patterns might suggest a specific transcript of *Foxp1* (Foxp1-225) is

selectively activated in CA1. by contrast, *Lrrtm4*, encoding a DG specific presynaptic protein that mediates excitatory synapse development[30], shows extensive looping to distal elements in DG but not CA1 (Extended Data Fig. 8n). Notably, among 34 genes showing alternative loop usage, 20 genes expressed in both DG and CA1[31]; for example, the TSS of *Grm7* interacts with an upstream enhancer in DG and gene body enhancers in CA1 (Extended Data Fig. 8o).

## mC gradients in IT neurons

Cortical excitatory IT neurons are classified into major types corresponding to their laminar layers: L2/3, L4, L5 and L6 (Fig. 5a). In agreement with the anti-correlation between transcript levels and DNA methylation, we found hypomethylation in IT neurons of the layer marker genes[19] (Extended Data Fig. 9a). Furthermore, UMAP embedding (Fig. 5a) reveals a continuous gradient of IT neurons resembling the medial–lateral distribution of the cortical regions (Fig. 5b), strongly suggesting that the arealization information is well preserved in the DNA methylome.

To systematically explore the spatial gradient of DNA methylation, we merged the cells into spatial groups on the basis of their cortical layer and region and generated a taxonomy between them (Methods). The taxonomy split the cells into four layer groups, followed by cortical-region separation within each layer (Extended Data Fig. 9c), providing a clear structure for investigating layer-related or region-related methylation variation. Specifically, the layer-related transcription factors included many known laminar marker genes and their DNA-binding motifs (Extended Data Fig. 9d), whereas some also show regional specific methylation differences. For example, *Cux1*, encoding a homeobox transcription factor specific to L2/3 and L4 neurons, is hypomethylated in motor (MO) and somatosensory (SS) cortex, but is hypermethylated in L2/3 of other regions, in agreement with patterns from in situ hybridization[32]. *Cux2*, which encodes another homeobox transcription factor, does not show the same regional specificity (Extended Data Fig. 9a). We also identified genes for many additional transcription factors that showed cortical region specificity (Fig. 5c, Extended Data Fig. 9e). For example, *Etv6* is only hypomethylated in medial dissection regions across layers, whereas *Zic4* is hypermethylated in those regions. By contrast, *Rora* shows an anterior–posterior methylation gradient within the L4 and L5 cells. Together, these observed methylome spatial gradients demonstrated the value of our dataset for further exploring the cortical arealization with cell-type resolution.

## mC gradients in DG granule cells

Global methylation gradients are observed within large cell types. For example, DG granule cells were continuously distributed in the UMAP embedding from low to high global mCH and mCG (Fig. 5d, global mCH fraction 0.5–1.9%, mCG fraction 69–79%). This gradient correlated with the anterior–posterior position of brain sections. Granule cells from the most posterior DG regions had higher global methylation than cells from anterior regions (Fig. 5d).

mCH accumulates throughout the genome during postnatal brain development[6,8]. We reasoned that DG granule cells, which are continuously replenished by ongoing neurogenesis throughout the lifespan, may accumulate mCH during their post-mitotic maturation. If so, global mCH should correlate with the age and maturity of granule cells. To investigate this, we divided DG granule cells into four groups on the basis of their global mCH levels and investigated regions of differential methylation between the groups. We identified 219,498 gradient CG-DMRs between the four groups, among which 139,387 showed a positive correlation with global mCH (+DMR), and 80,111 were negatively correlated (−DMR) (Fig. 5e). Notably, genes overlapping +DMRs or −DMRs have different annotated functions: genes enriched in +DMRs

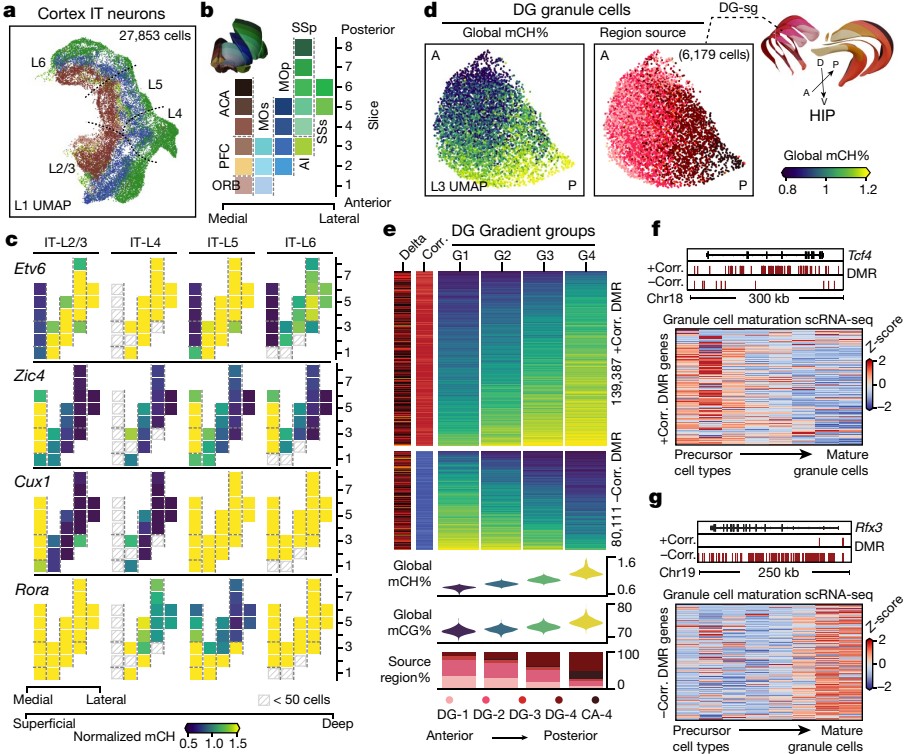

**Fig. 5 | Brain-wide spatial gradients of DNA methylation. a**, UMAP for cortex IT neurons coloured by dissection regions. **b**, The 21 cortical dissection regions organized by spatial axis. **c**, Normalized mCH fraction of spatial CH-DMGs, with the same layout as **b**. **d**, UMAPs for DG granule cells coloured by their cell global mCH fractions and dissection regions. A, anterior; P, posterior; D, dorsal; V, ventral. **e**, Compound figure showing four cells groups organized according to DG gradient and the two gradient DMR groups separated according to the sign of the correlation to the cell's global mCH level. **f**, **g**, Bottom, expression (z-score across rows) of genes positively (**f**) or negatively (**g**) correlated with DMRs across cell types along the granule cell maturation pathway[33]. Top, genome browser views of representative genes.

(+DMRgenes, n = 328) were associated with developmental processes, whereas those enriched in −DMRs (−DMRgenes, n = 112) were related to synaptic function (Extended Data Fig. 10a, b).

To further test the relationship between the +DMRgenes, −DMRgenes and DG development, we examined the expression patterns of these genes across time using a single-cell RNA-seq dataset that grouped DG cells into eight cell types, along their developmental trajectory from radial glia to mature granule cells[33]. The +DMRgenes were more highly expressed in immature cell types than in mature cell types (for example, *Tcf4*; Fig. 5f, Extended Data Fig. 10c), whereas the −DMRgenes showed the reverse trend (for example, *Rfx3*; Fig. 5g, Extended Data Fig. 10d). These results are consistent with the hypothesis that young DG granule cells have low global mCH and low methylation at genes associated with neural precursors. Conversely, older DG granule cells accumulate greater global mCH and have low methylation at genes associated with mature neurons. Notably, the global mCH levels also correlate with the brain dissections (Fig. 5d), indicating that the spatial axis can partially explain the methylation gradient (Supplementary Note 4).

Next, we investigated whether the global methylation level is correlated with 3D genome architecture. By plotting the chromatin interaction strength against the anchors' genomic distance, we observed a higher proportion of short-range contacts and a smaller proportion of long-range contacts in the groups with higher global mCH (Extended Data Fig. 10f). Although compartment strengths were not correlated with the global methylation changes (Extended Data Fig. 10g), the number of intra-domain contacts was positively correlated with global mCH across single cells (Extended Data Fig. 10h). After normalizing for the effect of decay, we found that insulation scores at domain boundaries were significantly lower in the groups with high global mCH levels (Extended Data Fig. 10i; all $P < 1 \times 10^{-10}$, two-sided Wilcoxon signed-rank test). Together these suggest that local structures may be more condensed over flanking regions in the high-mCH cell groups.

## Cell type and spatial prediction model

To further quantify the spatial and cell-type information encoded in a single cell's DNA methylome, we built a multi-task deep ANN using cell-level methylome profiles from this study (Fig. 6a). Specifically, mCH levels of 100-kb bins were used to train and test the network with fivefold cross-validation (Method). The ANN predicted neuronal subtype identity and spatial location simultaneously for each testing cell with 95% and 89% accuracy, respectively (Fig. 6b–d). Notably, the location prediction accuracy of the ANN was higher than using only the spatial distribution information of subtypes (overall increased by 38%, Extended Data Fig. 11c), suggesting that spatial diversity is well-preserved in the neuronal DNA methylome. We also notice higher levels of errors in location prediction of some cell types, especially in the cortical MGE and CGE inhibitory neurons (Fig. 6c, Extended Data Fig. 11c). This finding is consistent with previous transcriptome-based studies[19,31], suggesting these neurons do not display strong cortical region specificity. Many cell-type marker genes are also enriched in features that capture most spatial information (Fig. 6e, f). For example, besides distinguishing CT-L6 neurons from other cell types, *Foxp2* shows notable mCH differences among dissected regions within CT-L6 (Fig. 6f). Notably, we also observed the moderate spatial specificity of astrocytes and oligodendrocytes using a separate model trained with methylomes of non-neuronal cells (Supplementary Note 5).

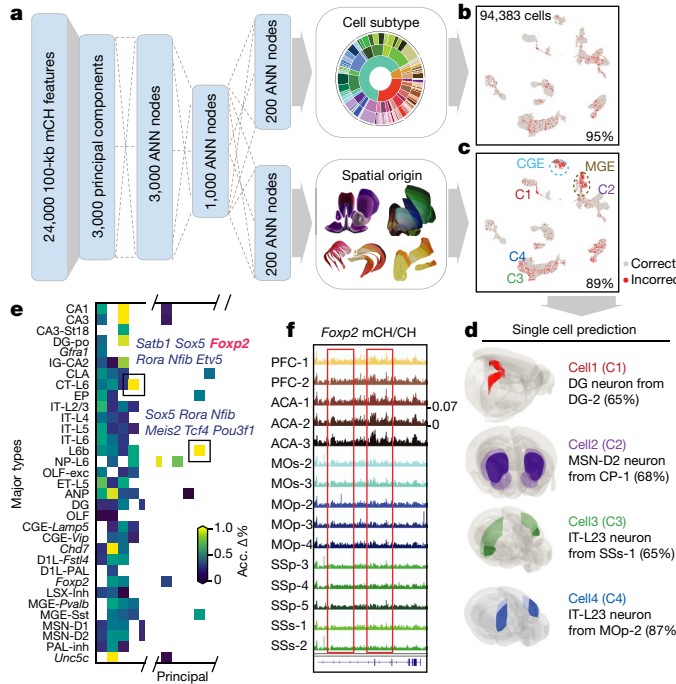

**a**

24,000 100-kb mCH features → 3,000 principal components → 3,000 ANN nodes → 1,000 ANN nodes → 200 ANN nodes → Cell subtype

200 ANN nodes → Spatial origin

**b** 94,383 cells — 95%

**c** CGE, MGE, C1, C2, C3, C4 — 89% — Correct / Incorrect

**d** Single cell prediction

Cell1 (C1) DG neuron from DG-2 (65%)

Cell2 (C2) MSN-D2 neuron from CP-1 (68%)

Cell3 (C3) IT-L23 neuron from SSs-1 (65%)

Cell4 (C4) IT-L23 neuron from MOp-2 (87%)

**e** Major types: CA1, CA3, CA3-St18, DG-po, Gfra1, IG-CA2, CLA, CT-L6, EP, IT-L2/3, IT-L4, IT-L5, IT-L6, L6b, NP-L6, OLF-exc, ET-L5, ANP, DG, OLF, CGE-Lamp5, CGE-Vip, Chd7, D1L-Fstl4, D1L-PAL, Foxp2, LSX-Inh, MGE-Pvalb, MGE-Sst, MSN-D1, MSN-D2, PAL-inh, Unc5c

Satb1 Sox5 *Foxp2*; Rora Nfib Etv5

Sox5 Rora Nfib; Meis2 Tcf4 Pou3f1

Acc. Δ% — 1.0 / 0.5 / 0

Principal components

**f** *Foxp2* mCH/CH

PFC-1, PFC-2, ACA-1, ACA-2, ACA-3, MOs-2, MOs-3, MOp-2, MOp-3, MOp-4, SSp-3, SSp-4, SSp-5, SSs-1, SSs-2 — 0.07 / 0

**Fig. 6 | A methylome-based predictive model captures both cellular and spatial characteristics of neurons. a**, Schematic of the model that predicts both cell-type identity and spatial origin. **b, c**, Model performance on the prediction of cell subtypes (**b**), and dissection regions (**c**). Per cent accuracy is shown. **d**, Examples of using the model to predict cell spatial origin (maximum prediction probability in parentheses). **e**, Evaluation of importance of features (principal components) for spatial origin prediction accuracy. 'Acc. Δ%' denotes the average prediction accuracy decrease percentage. **f**, The *Foxp2* gene body mCH fraction in each cortical dissection region group.

## Discussion

In this Article, we present a single-cell DNA methylomic atlas of the mouse brain with detailed spatial dissection. This comprehensive dataset enables high-throughput cell-type classification, marker gene prediction and identification of regulatory elements. The three-level iterative clustering defined 161 subtypes representing excitatory (68), inhibitory (77) and non-neuronal cells (16). The development of a hierarchical taxonomic architecture for cell subtypes on the basis of CH-DMGs allowed us to assign specific genes and transcription factor binding motifs to taxonomy branches using the methylation impact score. These assignments describe cell-type specificity at different levels, potentially relating to different developmental stages of each neuronal lineage. Notably, we found that transcription factor genes and their corresponding DNA-binding motifs were co-associated with the same branch in the taxonomy, providing a rich source of candidate transcription factors for future study.

Through integration with snATAC-seq[3], we matched subtypes classified in both epigenomic modalities and used the combined information to predict 1.6 million active-enhancer-like eDMRs, including 72% of cell-type-specific elements missed from previous tissue-level bulk studies[8]. To examine the associations of eDMRs and their targeting genes, we applied multi-omic methods to establish an eDMR–gene landscape using correlation-based prediction and chromatin conformation profiling using sn-m3C-seq, resulting in the identification of chromatin loops between eDMRs and their potential targeting genes in specific cell types.

Our brain-wide epigenomic dataset reveals extraordinary spatial diversity encoded in the DNA methylomes of neurons. The ANN trained on the single-cell methylome profiles accurately reproduced the detailed brain-dissection information within most subtypes, indicating the existence of large spatial methylation gradients throughout the brain. Echoing cortex development studies[34], glutamatergic neurons are regionalized by a protomap formed from an early developmental gradient of transcription factor expression. Similarly, we observed that many transcription factor genes and their corresponding DNA-binding motifs showed gradients of DNA methylation in adult IT neurons from distinct cortical regions. Additionally, we also found intra-subtype methylation gradients in DG granule cells that correlate with the spatial axis in the DG. These gradient-related CG-DMRs are enriched in essential neurodevelopmental and synaptic genes[33,35], suggesting that these spatially resolved DNA methylation gradients reflect past regulatory events occurring during brain maturation. We qualify our findings by noting that snmC-seq2 is a sodium bisulfite-based method and cannot distinguish between 5-methylcytosine and 5-hydroxymethylcytosine, which has been shown to accumulate in some brain regions[36]. New methods will be needed to simultaneously measure the full complement of cytosine base modifications at the single-cell level.

Overall, our analysis highlights the power of this dataset power for characterizing cell types using gene activity information from both coding regions and the regulatory elements in the non-coding regions of the genome. This comprehensive epigenomic dataset provides a valuable resource for answering fundamental questions about gene regulation in specifying cell-type spatial diversity and provides the raw material to develop new genetic tools for targeting specific cell types and functional testing.

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

## Methods

### Mouse brain tissues

All experimental procedures using live animals were approved by the Salk Institute Animal Care and Use Committee under protocol number 18-00006. Adult (P56) C57BL/6J male mice were purchased from Jackson Laboratories and maintained in the Salk animal barrier facility on 12 h dark-light cycles with food ad libitum for a maximum of 10 days. Brains were extracted and sliced coronally at 600 μm from the frontal pole across the whole brain (for a total of 18 slices) in an ice-cold dissection buffer containing 2.5 mM KCl, 0.5 mM CaCl$_2$, 7 mM MgCl$_2$, 1.25 mM NaH$_2$PO$_4$, 110 mM sucrose, 10 mM glucose and 25 mM NaHCO$_3$. The solution was kept ice-cold and bubbled with 95% O$_2$, 5% CO$_2$ for at least 15 min before starting the slicing procedure. Slices were kept in 12-well plates containing ice-cold dissection buffers (for a maximum of 20 min) until dissection aided by an SZX16 Olympus microscope equipped with an SDF PLAPO 1XPF objective. Olympus cellSens Dimension 1.8 was used for image acquisition. Each brain region was dissected from slices along the anterior-posterior axis according to the Allen Brain reference Atlas CCFv3[16] (see Extended Data Fig. 1 for the depiction of a posterior view of each coronal slice). Slices were kept in ice-cold dissection media during dissection and immediately frozen in dry ice for posterior pooling and nuclei production. For nuclei isolation, each dissected region was pooled from 6–30 animals, and two biological replicas were processed for each slice.

### Fluorescence-activated nuclei sorting

Nuclei were isolated as previously described[1,6]. Isolated nuclei were labelled by incubation with 1:1,000 dilution of Alexa Fluor 488-conjugated anti-NeuN antibody (MAB377X, Millipore) and a 1:1,000 dilution of Hoechst 33342 at 4 °C for 1 h with continuous shaking. FANS of single nuclei was performed using a BD Influx sorter with an 85-μm nozzle at 22.5 PSI sheath pressure. Single nuclei were sorted into each well of a 384-well plate preloaded with 2 μl of proteinase K digestion buffer (1 μl M-Digestion Buffer (Zymo, D5021-9), 0.1 μl of 20 μg μl$^{-1}$ proteinase K and 0.9 μl H$_2$O). The alignment of the receiving 384-well plate was performed by sorting sheath flow into wells of an empty plate and making adjustments based on the liquid drop position. Single-cell (one-drop single) mode was selected to ensure the stringency of sorting. For each 384-well plate, columns 1–22 were sorted with NeuN$^+$ (488+) gate, and column 23-24 with NeuN$^-$ (488−) gate, reaching an 11:1 ratio of NeuN$^+$ to NeuN$^-$ nuclei. BD Influx Software v1.2.0.142 was used to select cell populations.

### Library preparation and Illumina sequencing

Detailed methods for bisulfite conversion and library preparation were previously described for snmC-seq2[1,2]. The snmC-seq2 and sn-m3C-seq (see below) libraries generated from mouse brain tissues were sequenced using an Illumina Novaseq 6000 instrument with S4 flow cells using the 150-bp paired-end mode. Freedom EVOware v2.7 was used for library preparation, and Illumina MiSeq control software v3.1.0.13 and NovaSeq 6000 control software v1.6.0/Real-Time Analysis (RTA) v3.4.4 were used for sequencing.

### The sn-m3C-seq specific steps of library preparation

Single-nucleus methyl-3C sequencing (sn-m3C-seq) was performed as previously described[4]. In brief, the same batch of dissected tissue samples from the dorsal dentate gyrus (DG-1 and DG-2, Supplementary Table 2), ventral dentate gyrus (DG-3 and DG-4), dorsal HIP (CA-1 and CA-2), and ventral HIP (CA-3 and CA-4), were frozen in liquid nitrogen. The samples were then pulverized while frozen using a mortar and pestle, and then immediately fixed with 2% formaldehyde in DPBS for 10 min. The samples were quenched with 0.2 M glycine and stored at −80 °C until ready for further processing. After isolating nuclei as previously described[4], nuclei were digested overnight with NlaIII and

ligated for 4 h. Nuclei were then stained with Hoechst 33342 (but not stained with NeuN antibody) and filtered through a 0.2-μm filter, and sorted similarly to the snmC-seq2 samples. Libraries were generated using the snmC-seq2 method.

### Mouse brain region nomenclature

The mouse brain dissection and naming of anatomical structures in this study followed the Allen Mouse Brain common coordinate framework (CCF)[16]. On the basis of the hierarchical structure of the Allen CCF, we used a three-level spatial region organization to facilitate description: (1) the major region, for example, isocortex, HIP; (2) the sub-region, for example, MOp, SSp, within isocortex; (3) the dissection region, for example, MOp-1 and MOp-2, within MOp. Supplementary Table 1 contains the full names of all abbreviations used in this study. All brain atlas images were created based on Wang et al.[16] and ©2017 Allen Institute for Brain Science. Allen Brain Reference Atlas. Available from: http://atlas.brain-map.org/.

### Analysis stages

The following method sections were divided into three stages. The first stage, 'Mapping and feature generation', describes mapping and generating files in the single-cell methylation-specific data format. The second stage, 'Clustering related', describes clustering, identifying DMGs, or integrating other datasets, which all happened at the single-cell level. The third stage, 'Cell-type-specific regulatory elements', describes the identification of putative cell-type-specific regulatory elements using cluster-merged methylomes. Other figure-specific analysis topics may combine results from more than one stage.

### Mapping and feature generation

**Mapping and feature-count pipeline.** We implemented a versatile mapping pipeline, YAP (https://hq-1.gitbook.io/mc/), for all the single-cell-methylome-based technologies developed by our group[1,2,37]. The main steps of this pipeline include: (1) demultiplexing FASTQ files into single cells; (2) reads level quality control (QC); (3) mapping; (4) BAM file processing and QC; and (5) final molecular profile generation. The details of the five steps for snmC-seq2 were previously described[2]. We mapped all of the reads to the mouse mm10 genome. We calculated the methylcytosine counts and total cytosine counts for two sets of genomic regions in each cell after mapping. Non-overlapping chromosome 100-kb bins of the mm10 genome (generated by "bedtools makewindows -w 100000") were used for clustering analysis and ANN model training, and the gene body regions ±2 kb defined by the mouse GENCODE vm22 were used for cluster annotation and integration with other modalities.

**sn-m3C-seq-specific steps or read mapping and chromatin contact analysis.** Methylome sequencing reads were mapped following the TAURUS-MH pipeline, as previously described[4]. Specifically, reads were trimmed for Illumina adaptors, and then an additional 10 bp was trimmed on both sides. Then R1 and R2 reads were mapped separately to the mm10 genome using Bismark with Bowtie. The unmapped reads were collected and split into shorter reads representing the first 40 bp, the last 40 bp, and the middle part of the original reads (if read length >80 bp after trimming). The split reads were mapped again using Bismark with Bowtie. The reads with MAPQ <10 were removed. The filtered bam files from split and unsplit R1 and R2 reads were deduplicated with Picard and merged into a single bam file to generate the methylation data. Methylpy (v1.4.2)[38] was used to generate an ALLC file (base-level methylation counts) from the bam file for every single cell. We paired the R1 and R2 bam files where each read-pair represents a potential contact to generate the Hi-C contact map. For generating contact files, read pairs where the two ends mapped within 1 kbp of each other, were removed.

### Clustering-related methods

**Single-cell methylome data quality control and preprocessing.** Cell filtering. We filtered the cells on the basis of these main mapping

metrics: (1) mCCC level <0.03; (2) overall mCG level >0.5; (3) overall mCH level <0.2; (4) total final reads >500,000; and (5) Bismark mapping rate >0.5. Other metrics such as genome coverage, PCR duplicates rate and index ratio were also generated and evaluated during filtering. However, after removing outliers with the main metrics 1–5, few additional outliers were found. Note the mCCC level is used as the estimation of the upper bound of bisulfite non-conversion rate[1].

Feature filtering. 100 kb genomic bin features were filtered by removing bins with mean total cytosine base calls <250 (low coverage) or >3,000 (unusually high-coverage regions). Regions that overlap with the ENCODE blacklist[39] were also excluded from further analysis.

Computation and normalization of the methylation level. For CG and CH methylation, the methylation level computation from the methylcytosine and total cytosine matrices contains two steps: (1) prior estimation for the beta-binomial distribution, and (2) posterior level calculation and normalization per cell.

Step 1: for each cell, we calculated the sample mean $m$ and variance $v$ of the raw methylcytosine level (mc/cov), where cov is the total cytosine base coverage and mc is the methylcytosine base coverage, for each sequence context (CG or CH). The shape parameters $(\alpha, \beta)$ of the beta distribution were then estimated using the method of moments:

$$\alpha = m(m(1-m)/v - 1)$$

$$\beta = (1-m)(m(1-m)/v - 1)$$

This approach used different priors for different methylation types for each cell and used weaker priors to cells with more information (higher raw variance).

Step 2: we then calculated the posterior: $\widehat{mc} = \frac{\alpha + mc}{\alpha + \beta + cov}$ for all bins in each cell. Like the counts per million reads (CPM) normalization in the single-cell RNA-seq analysis, we normalized this posterior methylation ratio by the cell's global mean methylation, $m = \alpha/(\alpha + \beta)$. Thus, all the posterior $\widehat{mc}$ values with 0 cov will have a constant value of 1 after normalization. The resulting normalized mc level matrix contains no NA (not available) value, and features with lower cov tend to have a mean value close to 1.

Selection of highly variable features. Highly variable methylation features were selected with a modified approach using the scanpy.pp.highly_variable_genes function from the scanpy 1.4.3 package[40]. In brief, the scanpy.pp.highly_variable_genes function normalized the dispersion of a gene by scaling with the mean and standard deviation of the dispersions for genes falling into a given bin for mean expression of genes. In our modified approach, we reasoned that both the mean methylation level and the mean cov of a feature (100 kb bin or gene) could impact mc level dispersion. We grouped features that fall into a combined bin of mean and cov. We then normalized the dispersion within each mean–cov group. After dispersion normalization, we selected the top 3,000 features based on normalized dispersion for clustering analysis.

Dimension reduction and combination of different mC types. For each selected feature, mc levels were scaled to unit variance and zero mean. We then performed principal component analysis (PCA) on the scaled mc level matrix. The number of principal components (PCs) was selected by inspecting the variance ratio of each PC using the elbow method. The CH and CG PCs were then concatenated together for further analysis in clustering and manifold learning (Supplementary Table 6 for parameters of PCA and clustering analysis).

**Consensus clustering.** Consensus clustering on concatenated PCs. We used a consensus clustering approach based on multiple Leiden clustering[41] over $k$-nearest neighbour (KNN) graph to account for the randomness of the Leiden clustering algorithms. After selecting dominant PCs from PCA in both mCH and mCG matrices, we concatenated the PCs together to construct a KNN graph using scanpy.pp.neighbours

with Euclidean distance. Given fixed resolution parameters, we repeated the Leiden clustering 300 times on the KNN graph with different random starts and combined these cluster assignments as a new feature matrix, where each single Leiden result is a feature. We then used the outlier-aware DBSCAN algorithm from the scikit-learn package to perform consensus clustering over the Leiden feature matrix using the hamming distance. Different epsilon parameters of DBSCAN are traversed to generate consensus cluster versions with the number of clusters that range from the minimum to the maximum number of clusters observed in the multiple Leiden runs. Each version contained a few outliers; these usually fall into three categories: (1) cells located between two clusters had gradient differences instead of clear borders, for example, border of IT layers; (2) cells with a low number of reads potentially lack information in essential features to determine the specific cluster; and (3) cells with a high number of reads that were potential doublets. The number of type 1 and 2 outliers depends on the resolution parameter and is discussed in the choice of the resolution parameter section. The type 3 outliers were very rare after cell filtering. The supervised model evaluation below then determined the final consensus cluster version.

Supervised model evaluation on the clustering assignment. We performed a recursive feature elimination with cross-validation (RFECV)[42] process from the scikit-learn package to evaluate clustering reproducibility for each consensus clustering version. We first removed the outliers from this process, and then we held out 10% of the cells as the final testing dataset. For the remaining 90% of the cells, we used ten-fold cross-validation to train a multiclass prediction model using the input PCs as features and sklearn.metrics.balanced_accuracy_score[43] as an evaluation score. The multiclass prediction model is based on BalancedRandomForestClassifier from the imblearn package, which accounts for imbalanced classification problems[44]. After training, we used the 10% testing dataset to test the model performance using the score from balanced_accuracy_score. We kept the best model and corresponding clustering assignments as the final clustering version. Finally, we used this prediction model to predict outliers' cluster assignments. We rescued the outlier with prediction probability >0.3, otherwise labelling them as outliers.

Manifold learning for visualization. In each round of clustering analysis, the $t$-SNE[45,46] and UMAP[17] embedding were run on the PC matrix the same as the clustering input using the implementation from the scanpy[40] package. The coordinates from both algorithms were in Supplementary Table 5.

Choice of resolution parameter. Choosing the resolution parameter of the Leiden algorithm is critical for determining the final number of clusters. We selected the resolution parameter by three criteria: (1). the portion of outliers <0.05 in the final consensus clustering version; (2) the ultimate prediction model accuracy >0.9; and (3) the average cell per cluster ≥ 30, which controls the cluster size to reach the minimum coverage required for further epigenome analysis such as DMR calls. All three criteria prevented the over-splitting of the clusters; thus, we selected the maximum resolution parameter under meeting the criteria using a grid search.

Cell class (level 1 clustering) annotation. We annotated non-neuron cells based on both the NeuN[−] gate origin and low global mCH fraction. Given the strong anti-correlation between CH methylation and gene expression, we used hypo-CH-methylation at gene bodies ±2 kb of pan-excitatory markers such as *Slc17a7* and *Sv2b*, and pan-inhibitory markers such as *Gad1* and *Gad2* to annotate excitatory and inhibitory cell classes, respectively.

Major type (level 2) and subtypes (level 3) annotations. We used both gene body ±2 kb hypo-CH-methylation (or hypo-CG-methylation for non-neurons) of well-known marker genes and the dissection information to annotate neuron and non-neuron clusters. All cluster marker genes are listed in Supplementary Table 7, together with the description of the cluster names, references to the marker gene information, and

the URL to the data browser. The major cell types were annotated based on well-known marker genes reported in the previous studies[1,19,31,47–49]. Whenever possible, we name these clusters with canonical names (for example, IT-L23, L6b) or using descriptive names that reflect the specific spatial location of the cluster (for example, EP, CLA, IG-CA2). For subtypes, we named the clusters via its parent major type name followed by a subtype marker gene name.

**Pairwise DMG identification.** We used a pairwise strategy to calculate DMGs for each pair of clusters within the same round of analysis. We used the gene body ±2 kb regions of all the protein-coding and long non-coding RNA genes with evidence level 1 or 2 from the mouse GENCODE vm22. We used the single-cell level mCH fraction normalized by the global mCH level (as in 'Computation and normalization of the methylation level' in the clustering step above) to calculate markers between all neuronal clusters. We compared non-neuron clusters separately using the mCG fraction normalized by the global mCG level. For each pairwise comparison, we used the Wilcoxon rank-sum test to select genes with a significant decrease (hypo-methylation). Marker genes were chosen based on adjusted $P < 10^{-3}$ with multitest correction using the Benjamini–Hochberg procedure, delta-normalized methylation level change $<-0.5$ (hypo-methylation) and area under the receiver-operating curve (AUROC) $>0.8$. We required each cluster to have $\geq 5$ DMGs compared to any other cluster. Otherwise, the smallest cluster that did not meet this criterion was merged to the closest cluster based on Euclidean distance between cluster centroids in the PC matrix used for clustering. Then the marker identification process was repeated until all clusters found enough marker genes.

**Three levels of iterative clustering analysis.** On the basis of the consensus clustering steps described above, we used an iterative approach to cluster the data into three levels of categories. In the first level, termed CellClass, clustering analysis is done using all cells and then manually merged into three canonical classes: excitatory neurons, inhibitory neurons, and non-neurons based on marker genes. Within each CellClass, we performed all the preprocessing and clustering steps again to obtain clusters for the MajorType level using the same stop criteria. Furthermore, within each MajorType, we obtained clusters for the SubType level. All clusters' annotations and relationships are presented in Supplementary Table 7.

**Subtype taxonomy tree.** To build the taxonomy tree of subtypes, we selected the top 50 genes that showed the most significant changes for each subtypes' pairwise comparisons. We then used the union of these genes from all subtypes and obtained 2,503 unique genes. We calculated the median mCH level of these genes in each subtype and applied bootstrap resampling-based hierarchical clustering with average linkage and the correlation metric using the R package pvclust (v.2.2)[50].

**Impact score and total impact score.** We defined the impact score (IS) to summarize pairwise comparisons for two subtype groups, where one group, A, contains $M$ clusters and the other group, B, contains $N$ clusters. For each gene or motif, the number of total related pairwise comparisons is $M \times N$, the number of significant comparisons with desired change (hypo-methylation for gene or enrichment for motif) is $a$ in group A and $b$ in group B. The IS is then calculated as $IS_A = \frac{a - b}{M \times N}$ and $IS_B = \frac{b - a}{M \times N}$ for the two directions. For either group, IS ranges from $-1$ to 1, and 0 means no impact, 1 means full impact and $-1$ means full impact in the other group (Extended Data Fig. 7e).

We explored two scenarios using the IS to describe cluster characteristics (Extended Data Fig. 7e). The first scenario is considering each pair of branches in the subtype taxonomy tree as comprising group A and group B. Thus, the IS can quantify and rank genes or motifs to the upper nodes based on the leaves' pairwise comparisons (Fig. 3d–f). The second scenario summarizes the total impact for specific genes or

motifs regarding the taxonomy tree based on the calculation in the first scenario (Extended Data Fig. 7f–k). In a subtype taxonomy tree with $n$ subtypes, the total non-singleton node was $n - 1$, and each node $i$ had a height $h_i$ and associated $IS_A$ for one of the branches ($IS_B = -IS_A$). The node-height-weighted total IS ($IS_{total}$) was then calculated as:

$$IS_{total} = \sum_{i=1}^{n-1} h_i \times |IS_A|$$

The larger total IS indicated that a gene or motif shows more cell-type-taxonomy-related significant changes. The total IS can also be calculated in a sub-tree or any combination of interests to rank genes and motifs most related to that combination (See 'Figure-specific methods' for Fig. 5 regarding calculating layer and region total IS from the same tree).

**Integration with snATAC-seq data.** A portion of the same brain tissue sample used in this study for methylome profiling was also processed using snATAC-seq in a parallel study of chromatin accessibility[3]. The final high-quality snATAC-seq cells were assigned to 160 chromatin accessibility clusters (a-types). The snATAC-seq-specific data analysis steps are described in Li et al.[3]. Here, we performed cross-modality data integration and label-transferring to assign the 160 a-types to the 161 methylome subtypes in the following steps:

(1) We manually grouped both modalities into five integration groups (for example, all IT neurons as a group) and only performed the integration of cells within the same group to decrease computation time. These groups were distinct in the clustering steps of both modalities and can be matched with great confidence using known marker genes. Steps 2–6 were repeated for each group. See Extended Data Fig. 5 for the group design.

(2) We used a similar approach as described above to identify pairwise differential accessible genes (DAGs) between all pairs of a-types. The cut-off for DAG is adjusted $P < 10^{-3}$, fold change $>2$ and AUROC $>0.8$.

(3) We then gathered DMGs from comparisons of related subtypes in the same group. Both DAGs and DMGs were filtered according to whether they recurred in $>5$ pairwise comparisons. The intersection of the remaining genes was used as the feature set of integration.

(4) After identifying DAGs using cell-level snATAC-seq data, we merged the snATAC-seq cells into pseudo-cells to increase snATAC-seq data coverage. Within each a-type, we did a $k$-means clustering ($k$ = no. of cells in that cluster/50) on the same PCs used in snATAC-seq clustering. We discarded small $k$-means clusters with less than 10 cells (about 5% of the cells) and merged each remaining $k$-means cluster into a pseudo-cell. On average, a pseudo-cell had about 50 times more fragments than a single cell.

(5) We then used the MNN based Scanorama[51] method with default parameters to integrate the snmC-seq cells and snATAC-seq pseudo-cells using genes from step 3. After Scanorama integration, we did co-clustering on the integrated PC matrix using the clustering approaches described above.

(6) We used the intermediate clustering assignment from step 5 to calculate the overlap score (below) between the original methylome subtypes and the a-types. We used overlap score $>0.3$ to assign a-types to each methylome subtype. For those subtypes that have no match under this threshold, we assigned the top a-type ranked by the overlap score (Supplementary Tables 10, 11).

**Overlap score.** We used the overlap score to match a-type and methylome subtypes. The overlap score, range from 0 to 1, was defined as the sum of the minimum proportion of samples in each cluster overlapped within each co-cluster[52]. A higher score between one methylome subtype and one a-cluster indicates they consistently co-clustered within one or more co-clusters. Besides matching clusters in integration

analysis, the overlap score was also used in two other cases: (1) to quantify replicates and region overlaps over methylome subtypes (Extended Data Fig. 2e–g); and (2) to quantify the overlap of each L5-ET subtype overlapping with 'soma location' and 'projection target' labels from epi-retro-seq cells (Extended Data Fig. 5j) through integration with the epi-retro-seq dataset.

## Cell-type-specific regulatory elements

**DMR analysis.** After clustering analysis, we used the subtype cluster assignments to merge single-cell ALLC files into the pseudo-bulk level and then used methylpy (v1.4.2)[38] DMRfind function to calculate mCG DMRs across all subtypes. The base calls of each pair of CpG sites were added before analysis. In brief, the methylpy function used a permutation-based root mean square test of goodness of fit to identify differentially methylated sites (DMS) simultaneously across all samples (subtypes in this case), and then merge the DMS within 250 bp into the DMR. We further excluded DMS calls that have low absolute mCG level differences by using a robust-mean-based approach. For each DMR merged from the DMS, we ordered all the samples by their mCG fraction and calculated the robust mean $m$ using the samples between 25th and 75th percentiles. We then reassigned hypo-DMR and hyper-DMR to each sample when a region met two criteria: (1) the sample mCG fraction of this DMR is lower than $(m - 0.3)$ for hypo-DMR or $(m + 0.3)$ for hyper-DMR, and (2) the DMR is originally a significant hypo- or hyper-DMR in that sample judged by methylpy. DMRs without any hypo- or hyper-DMR assignment were excluded from further analyses. On the basis of these filtering criteria, we estimate the false discovery rate of calling DMRs is 2.7% (Supplementary Note 2, Extended Data Fig. 6).

**Enhancer prediction using DNA methylation and chromatin accessibility.** We performed enhancer prediction using the REPTILE[53] algorithm. REPTILE is a random-forest-based supervised method that incorporates different epigenomic profiles with base-level DNA methylation data to learn and then distinguish the epigenomic signatures of enhancers and genomic background. We trained the model in a similar way as in the previous studies[8,53], using CG methylation, chromatin accessibility of each subtype and mouse embryonic stem cells (mouse ES cells). The model was first trained on mouse ES cell data and then predicted a quantitative score that we termed enhancer score for each subtype's DMRs. The positives were 2 kb regions centred at the summits of the top 5,000 EP300 peaks in mouse ES cells. Negatives include randomly chosen 5,000 promoters and 30,000 2-kb genomic bins. The bins have no overlap with any positive region or gene promoter[8].

Methylation and chromatin accessibility profiles in bigwig format for mouse ES cells were from the GEO database (GSM723018). The mCG fraction bigwig file was generated from subtype-merged ALLC files using the ALLCools package (https://github.com/lhqing/ALLCools). For chromatin accessibility of each subtype, we merged all fragments from snATAC-seq cells that were assigned to this subtype in the integration analysis and used deeptools bamcoverage to generate CPM normalized bigwig files. All bigwig file bin sizes were 50 bp.

**Motif-enrichment analysis.** We used 719 motif PWMs from the JASPAR 2020 CORE vertebrates database[54], where each motif was able to assign corresponding mouse transcription factor genes. The specific DMR sets used in each motif-enrichment analysis are described in figure specific methods below. For each set of DMRs, we standardized the region length to the centre ±250bp and used the FIMO tool from the MEME suite[55] to scan the motifs in each enhancer with the log-odds score $P < 10^{-6}$ as the threshold. To calculate motif enrichment, we use the adult non-neuronal mouse tissue DMRs[10] as background regions unless expressly noted. We subtracted enhancers in the region set from the background and then scanned the motifs in background regions using the same approach. We then used Fisher's exact test to find motifs enriched in the region set and the Benjamini–Hochberg procedure to

correct multiple tests. We used the TFClass[56] classification to group transcription factors with similar motifs.

**DMR–DMG partial correlation.** To calculate DMR–DMG partial correlation, we used the mCG fraction of DMRs and the mCH fraction of DMGs in each neuronal subtype. We first used linear regression to regress out variance due to global methylation difference (using scanpy.pp.regress_out function), then use the residual matrix to calculate the Pearson correlation between DMR and DMG pairs where the DMR centre is within 1 Mb of the TSSs of the DMG. We shuffled the subtype orders in both matrices and recalculated all pairs 100 times to generate the null distribution.

**Identification of loops and differential loops from sn-m3C-seq data.** After merging the chromatin contacts from cells belonging to the same type, we generated a .hic file of the cell-type with Juicer tools pre. HICCUPS[57] was used to identify loops in each cell type. The loops from eight major cell types were concatenated and deduplicated and used as the total samples for differential loop calling. A loop-by-cell matrix was generated, in which each element represents the number of contacts supporting each loop in each cell. The matrix was used as input of EdgeR to identify differential interactions with ANOVA tests. Loops with FDR $<10^{-5}$ and minimum–maximum fold change >2 were used as differential loops. Note that the abundance of cell types is highly variable, leading to different coverages of contact maps after merging all the cells from each cell type. Since *HICCUPS* loop calling is sensitive to the coverage, more loops were identified in the abundant cell types (for example, 12,614 loops were called in DG, containing 1,933 cells) compared to the less abundant ones (for example, 1,173 loops were called in MGE, containing 145 cells). Therefore, we do not compare the feature counts related to the loops across cell types directly in our analyses.

## Figure-specific methods

3D model of dissection regions (Fig. 1b–e). We created in silico dissection regions based on the Allen CCF[16] 3D model using Blender 2.8 that precisely follow our dissection plan. To ease visualization of all different regions, we modified the layout and removed some of the symmetric structures, but all the actual dissections were applied symmetrically to both hemispheres.

Calculating the genome feature detected ratio (Extended Data Fig. 2a). The detected ratio of chromosome 100-kb bins and gene bodies is calculated as the percentage of bins with >20 total cytosine coverage. Non-overlapping chromosome 100-kb bins were generated by bedtools makewindows -w 100000; gene bodies were defined by GENCODE vm22.

Integration with epi-retro-seq L5-ET cells (Fig. 2g–j, Extended Data Fig. 5g–j). Epi-retro-seq is an snmC-seq2-based method that combines retrograde AAV labelling[22]. The L5-ET cells' non-overlapping chromosome 100-kb bin matrix gathered by the epi-retro-seq dataset was concatenated with all the L5-ET cells from this study for co-clustering and embedding as described in 'Clustering-related methods'. We then calculated the OS between subtypes in this study and the 'soma location' or 'projection target' labels of epi-retro-seq cells. The first OS helped quantify how consistent the spatial location is between the two studies; the second OS allowed us to impute the projection targets of subtypes in this study.

Pairwise DMR and motif-enrichment analysis (Fig. 3c, f). The total subtype DMRs were identified as described in 'Cell-type-specific regulatory elements' by comparing all subtypes. We then assigned DMRs to each subtype pair if the DMRs were: (1) significantly hypomethylated in only one of the subtypes; and (2) the mCG fraction difference between the two subtypes is >0.4. Each subtype pair was associated with two exclusive sets of pairwise DMRs. We carried out motif-enrichment analysis described in 'Cell-type-specific regulatory elements' on each DMR set using the other set as background. Motifs enriched in either

direction were then used to calculate the impact score and were associated with upper nodes of the taxonomy.

Overlapping eDMR with genome regions (Fig. 4b). The cluster-specific snATAC-seq peaks were identified in Li et al.[3]. We used bedtools merge to aggregate the total non-overlap peak regions and bedtools intersect to calculate the overlap between peaks and eDMRs. The developing forebrain and other tissue feDMRs were identified in He et al.[8] using methylC-seq[58] for bulk whole-genome bisulfite sequencing. All of the genome features used in Fig. 4b were defined as in He et al.[3], except using an updated mm10 CGI region and RepeatMaster transposable elements lists (UCSC table browser downloaded on 9 October 2019).

Heat maps of the gene–enhancer landscape (Extended Data Fig. 8e). The eDMRs for each gene were selected by eDMR–gene correlation of >0.3. Sections of the heat maps in Extended Data Fig. 8e were gathered by (1) mCG fraction of each eDMR in 161 subtypes from this study; (2) snATAC-seq subtype-level fragments per kilobase of transcript per million mapped reads (FPKM) of each eDMR in the same subtype orders. The subtype snATAC profiles were merged from integration results as described in 'Clustering-related methods'; (3) mCG fraction of each eDMR in forebrain tissue during ten developing time points from embryonic day 10.5 (E10.5) to P0 (data from He et al.[8]); (4) H3K27ac FPKM of each eDMR in 7 developing time points from E11.5 to P0 (data from Gorkin et al.[59]); (5) H3K27ac FPKM of each eDMR in P56 frontal brain tissue (data from Lister et al.[6]); and (6) eDMR is overlapped with forebrain feDMR using bedtools intersect.

Embedding of cells with chromosome interactions (Fig. 4e). scHi-Cluster[60] was used to generate the $t$-SNE embedding of the sn-m3C-seq cells. Specifically, a contact matrix at 1-Mb resolution was generated for each chromosome of each cell. The matrices were then smoothed by linear convolution with pad = 1 and random walk with restart probability = 0.5. The top 20th percentile of strongest interactions on the smoothed map was extracted, binarized and used for PCA. The first 20 PCs were used for $t$-SNE.

IT layer dissection region group DMG and DMR analysis (Fig. 5a–c). To collect enough cells for dissection region analysis, we used only the major types (corresponding to L2/3, L4, L5 and L6) of IT neurons. We grouped cells into groups according to layer dissection region and kept groups with >50 cells for further analysis (Extended Data Fig. 9b). We performed pairwise DMG, DMR and motif-enrichment analysis, the same as the subtype analysis in Fig. 3, but using the layer dissection region group labels. We then built a spatial taxonomy for these groups and used it to calculate impact scores. To rank layer-related or dissection-region-related genes and motifs separately, we used two sets of the branches (Extended Data Fig. 9c, top set for layers, bottom set for regions) in the taxonomy and calculated two total impact scores using the equations above.

DG cell group and gradient DMR analysis (Fig. 5e). DG cells were grouped into four evenly sized groups according to the cells' global mCH levels, with cut-off thresholds at 0.45%, 0.55% and 0.69%. We then randomly chose 400 cells from each group to call gradient-DMRs using methods described in 'Clustering-related methods'. To ensure the DMRs identified between intra-DG groups were not due to stochasticity, we also randomly sampled 15 groups of 400 cells from all DG cells regardless of their global mCH and called DMRs among them as control-DMRs (2,003 using the same filtering condition). Only 0.04% of gradient DMRs overlapped with the control DMRs; these were removed from further analysis. Pearson correlations ($\rho$) of mCG fractions of each gradient DMR was calculated against a linear sequence (1, 2, 3, 4) to quantify the gradient trend. DMRs with $\rho < -0.75$ or $\rho > 0.75$ were considered to be significantly correlated. Weakly correlated DMRs (10% of DMRs) were not included in further analysis.

DMR- and DMS-enriched genes (Fig. 5f,g). To investigate the correlated DMR or DMS enrichment in specific gene bodies, we compared the number of DMS and cytosine inside the gene body with the number of DMS and cytosine in the ±1 Mb regions using Fisher's exact test. We chose

genes passing two criteria: (1) adjusted $P < 0.01$ with multitest correction using the Benjamini–Hochberg procedure, and (2) overlap with >20 DMSs. Gene ontology analysis of DMR and DMS enriched genes was carried out using GOATOOLS[61]. All protein-coding genes with gene body length >5 kb were used as background to prevent gene-length bias.

Compartment strength analysis (Extended Data Fig. 10g). We normalized the total chromosome contacts by $z$-score in each 1-Mb bin of the DG contact matrix, and the bins with normalized coverage between −1 and 2 were kept for the analysis. After filtering, the PC1 of the genome-wide Knight–Ruiz-normalized[62] contact matrix was used as the compartment score. The score was divided into 50 categories with equal sizes from low to high, and bins were assigned to the categories. The intra-chromosomal observation/expectation (ove) matrices of each group were used to quantify the compartment strength. We computed the average ove values within each pair of categories to generate the 50 × 50 saddle matrices. The compartment strength was computed with the average of the upper left and lower right 10 × 10 matrices divided by the average of the upper right and lower left 10 × 10 matrices[63].

Domain analysis (Extended Data Fig. 10i). We identified 4,580 contact domains at 10-kb resolution in DG using Arrowhead[57]. For bin $i$, the insulation score $I$ is computed by

$$I_i = \frac{\mathrm{mean}_{i-10 \leq i' < i; i \leq j' < i+10} A_{i'j'}}{\max(\mathrm{mean}_{i-10 \leq i' < i; i-10 \leq j' < i} A_{i'j'}, \mathrm{mean}_{i \leq i' < i+10; i \leq j' < i+10} A_{i'j'})}$$

where $A$ is the ove of Knight–Ruiz-normalized matrices and mean is the average of $A$ over the range in the subcript. For each group, insulation scores of domain boundaries and 100-kb flanking regions were computed and averaged across all boundaries.

**Prediction model description.** Related to Fig. 6. To reduce the computing complexity, we applied PCA on the dataset of 100-kb bin mCH features to obtain the first 3,000 PCs, which retain 61% of the variance of the original data. These 3,000 PCs were then used to train and test the predicting model. We used an ANN with two hidden layers to simultaneously predict cell subtypes and their dissection regions. The input layer contains 3,000 nodes, followed by a shared layer with 1,000 nodes. The shared layer is further connected simultaneously to two branch hidden layers of the subsection region's subtype, each containing 200 nodes. The corresponding one-hot encoding output layers follow branch hidden layers. We used fivefold cross-validation to access the model performance. We applied the dropout technique[64] with a dropout rate $P = 0.5$ on each hidden layer to prevent overfitting during the training. Adam optimization[65] was used to train the network with a cross-entropy loss function. The training epoch number and batch size are 10 and 100, respectively. The training and testing processes were conducted via TensorFlow 2.0[66].

Model performance. The two output layers generate two probabilistic vectors for each single cell input as the prediction results for cell subtypes and dissection regions, respectively. The subtype and dissection region label with the highest probabilities were used as the prediction results for each cell to calculate accuracy. When calculating the cell dissection region accuracy (Fig. 6c), we defined two kinds of accuracy with different stringency: (1) the exact accuracy using the predicted label, and (2) the fuzzy accuracy using predicted labels or its potential overlap neighbours. The potential overlap neighbours curated based on Allen CCF (Extended Data Fig. 11a, Supplementary Table 2) stood for adjacent regions of a particular dissection region. The exact accuracy of the ANN model is 69% and the fuzzy accuracy is 89%. To evaluate how much of the dissection region accuracy was improved via ANN, we calculated fuzzy accuracy based only on naive guesses in each subtype based on the dissection region composition (grey dots in Extended Data Fig. 11c). We also trained additional models using logistic regression and random forest for benchmarks. The performance of ANN on subtype prediction is comparable with logistic

regression and random forest. By contrast, the performance in location prediction is substantially improved against the other two models (Extended Data Fig. 11b), suggesting that distinguishing the cells from different dissected regions may require nonlinear relationships between genomic regions. We used scikit-learn (v0.23) for logistic regression and random forest implementation and the multinomial objective function for multi-class classification. N_estimators were set to 1,000 for the random forest.

Biological feature importance for dissection region prediction (Fig. 6e). To assess which DNA regions store information of cell spatial origins that is distinguishable using our model, we evaluated the importance of PC features by examining how permutation of each PC feature across cells affects prediction accuracy. We tested five permutations for each feature and used decreasing average accuracy to indicate PC feature importance. We examined genes contained in the 100-kb bins with the top 1% PCA factor loadings for the most important PC feature for a given cell type.

## Reporting summary

Further information on research design is available in the Nature Research Reporting Summary linked to this paper.

## Data availability

Single-cell raw and processed data included in this study were deposited to NCBI Gene Expression Omnibus and Sequence Read Archive with accession number GSE132489 (each experiment has a separate accession number recorded in GSE132489; see Supplementary Table 13), and to the NeMO archive: https://assets.nemoarchive.org/dat-vmivr5x. Single-cell methylation data can be visualized at the Brain Cell Methylation Viewer: http://neomorph.salk.edu/omb/home. Cluster merged methylome profiles can be visualized at http://neomorph.salk.edu/mouse_brain.php. Other datasets used in the paper include single-nuclei ATAC-seq data[3] from http://catlas.org, mouse embryo forebrain development data[8] from the ENCODE portal (https://www.encodeproject.org/), the developing hippocampal single-cell RNA-seq data from GSE104323, DNA methylation and chromatin accessibility profiles for mouse ES cells from GSM723018 and the JASPAR 2020 CORE vertebrates database from http://jaspar.genereg.net/.

## Code availability

The mapping pipeline for snmC-seq2 data is available at https://hq-1.gitbook.io/mc/; the ALLCools package for post-mapping analysis and snmC-seq2 related data structure are available at https://github.com/lhqing/ALLCools; the jupyter notebooks for reproducing specific analysis are at https://github.com/lhqing/mouse_brain_2020; and the source code of the Brain Cell Methylation Viewer is at https://github.com/lhqing/omb.

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

**Acknowledgements** We thank Y. He for the advice on the methylpy and REPTILE analysis; T. Sejnowski for the advice on the ANN analysis. This work is supported by NIMH U19MH11483 to J.R.E. and E.M.C, and NHGRI R01HG010634 to J.R.E. and J.R.D. The Flow Cytometry Core Facility of the Salk Institute is supported by funding from NIH-NCI CCSG: P30 014195 and Shared Instrumentation Grant S10-OD023689. J.R.E is an investigator of the Howard Hughes Medical Institute.

**Author contributions** J.R.E., H.L., B.R., M.M.B., C.L. and J.R.D. conceived the study. H.L., J.Z. and W.T. analysed the snmC-seq data and drafted the manuscript. J.R.E., C.L., E.A.M., J.R.D. and M.M.B. edited the manuscript. J.R.E., H.L., M.M.B., A.B., J.L. and S.P. coordinated the research. M.M.B., A.B., A.A., H.L., J.L., J.R.N., A.R., J.K.O., A.P.-D., C.O., L.B., C.F., C.L. and J.R.E. generated the snmC-seq2 data. J.R.D., B.C., A.B., J.L., J.Z., A.A., J.K.O., C.L., J.R.N., C.O., L.B., C.F., R.G.C., M.M.B. and J.R.E. generated the sn-m3C-seq data. S.P., M.M.B., X.H., J.L., O.B.P., Y.E.L., J.K.O. and B.R. generated the snATAC-seq data. Z.Z., J.Z., E.M.C., M.M.B., J.R.E., A.B., A.A., J.R.N., C.O., L.B., C.F., R.G.C. and A.R. generated the epi-retro-seq data. H.L., H.C., E.A.M., M.N. and C.L. contributed to data archive/infrastructure. J.R.E. supervised the study.

**Competing interests** J.R.E serves on the scientific advisory board of Zymo Research Inc. B.R. is a shareholder of Arima Genomics.

**Additional information**
**Correspondence and requests for materials** should be addressed to J.R.E.

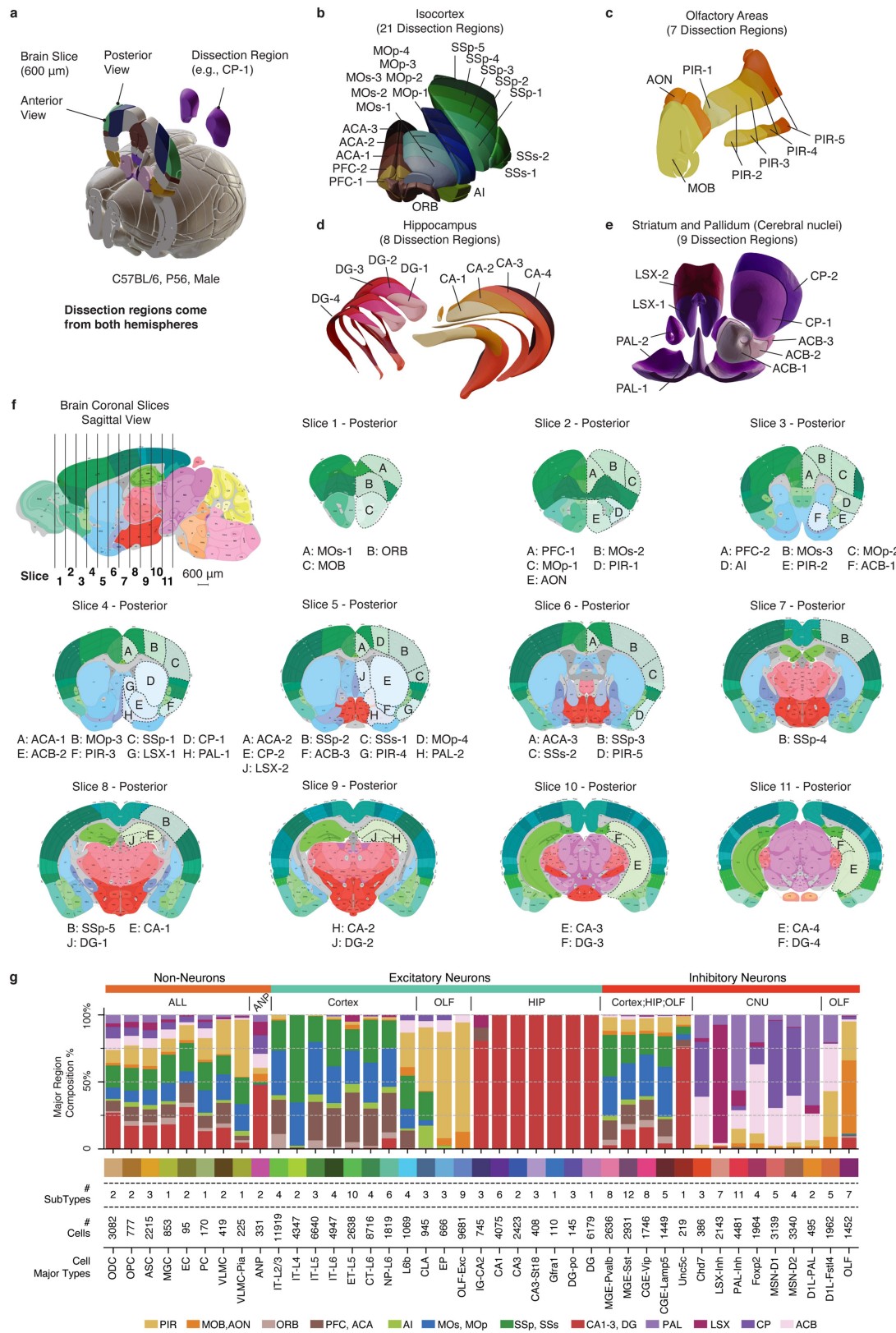

**Extended Data Fig. 1 | Brain dissection regions. a**, Schematic of brain dissection steps. Each male C57BL/6 mouse brain (age P56) was dissected into 600-µm slices. We then dissected brain regions from both hemispheres within a specific slice. **b**–**e**, 3D mouse brain schematic adapted from Allen CCFv3 to display the four major brain regions and 45 dissection regions. Each colour represents a dissection region. **f**, 2D mouse brain atlas adapted from Allen Mouse Brain Reference Atlas, the first sagittal image showing the location of each coronal slice, followed by 11 posterior view images of all coronal slices, the same 45 dissection regions are labelled on the corresponding slice. All coronal images follow the same scale as the sagittal image. The posterior view of each slice is the anterior view of the next slice. **g**, An integrated overview of brain region composition, subtype and cell numbers of the major types. All brain atlas images were created based on Wang et al.[16] and © 2017 Allen Institute for Brain Science. Allen Brain Reference Atlas. Available from: http://www.atlas. brain-map.org.

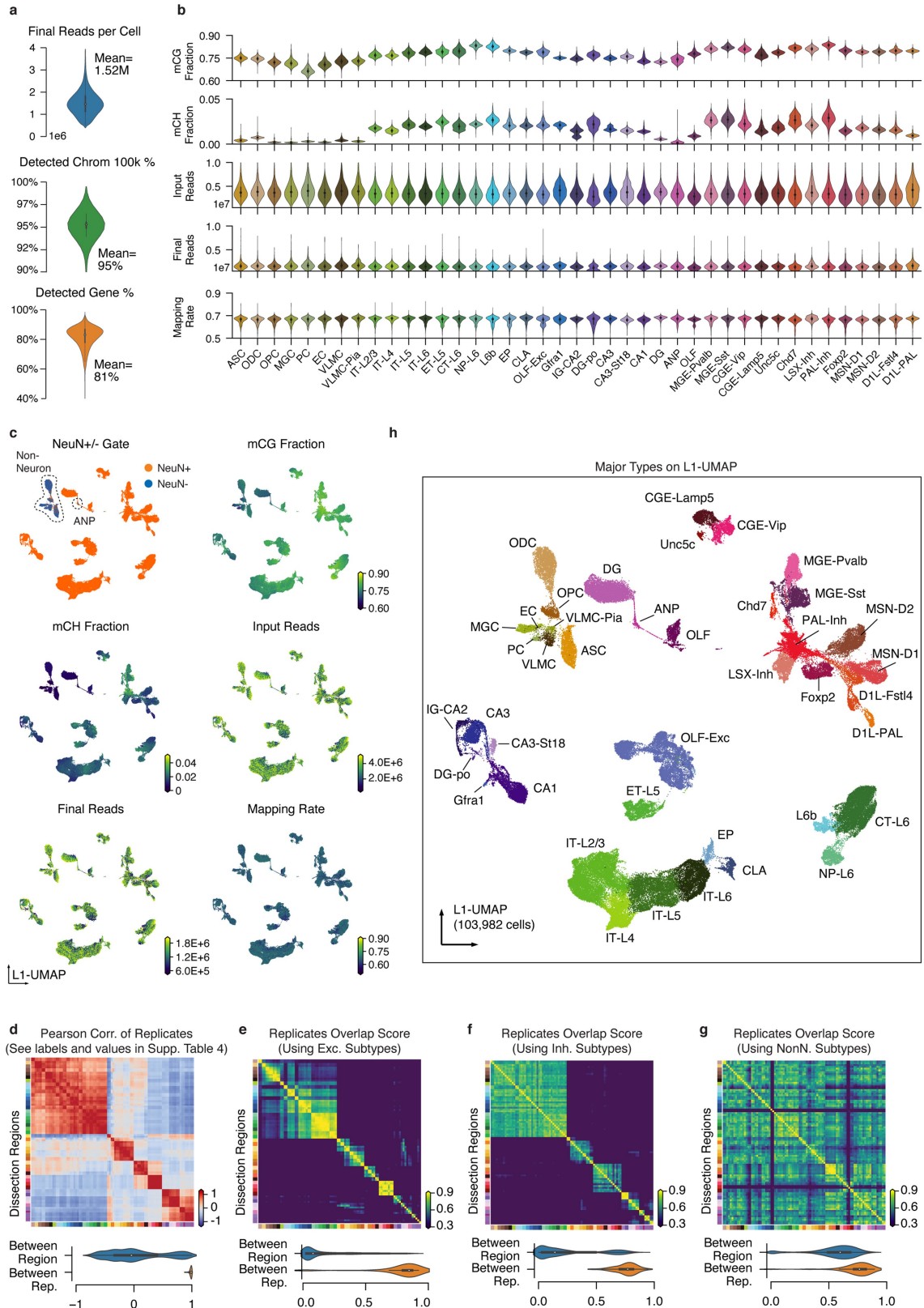

**Extended Data Fig. 2 | Major Type labelling and basic mapping metrics of snmC-seq2. a**, The number of final pass QC reads, the percentage of non-overlapping chromosome 100-kb bins detected, and the percentage of GENCODE vm22 genes detected per cell. **b**, Violin plots for all of the key metrics, group by major types. **c**, L1 UMAP coloured by NeuN antibody FACS gates and other snmC-seq2 key read mapping metrics. **d**, Heat map of Pearson correlation between the average methylome profiles (mean mCH and mCG fraction of all chromosome 100-kb bins across all cells belong to a replicate sample) of the 92 replicates from 45 brain regions. The violin plot below summarizes the value between replicates within the same brain region or between different brain regions. **e**–**g**, Pairwise overlap score (measuring co-clustering of two replicates) of excitatory subtypes (**e**), inhibitory subtypes (**f**), and non-neuronal subtypes (**g**). The violin plots summarize the subtype overlap score between replicates within the same brain region or between different brain regions. **h**, L1 UMAP coloured and labelled by major cell types.

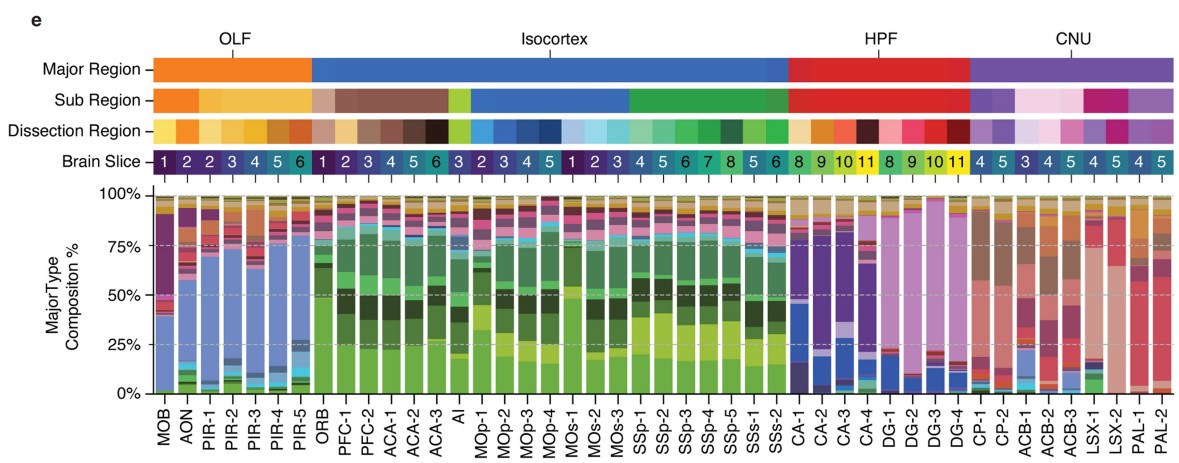

**Extended Data Fig. 3 | Cell-type composition of dissection regions. a-d**, L1 UMAP labelled by major types and partially coloured by dissection regions for cells from isocortex (**a**), OLF (**b**), HIP (**c**) and cerebral nucleus (**d**). Other cells are shown in grey as background. **e**, Similar compound bar plot as Extended Data Fig. 1g, arranged top to bottom, showing the organization of dissection regions and the major type composition of each dissection region.

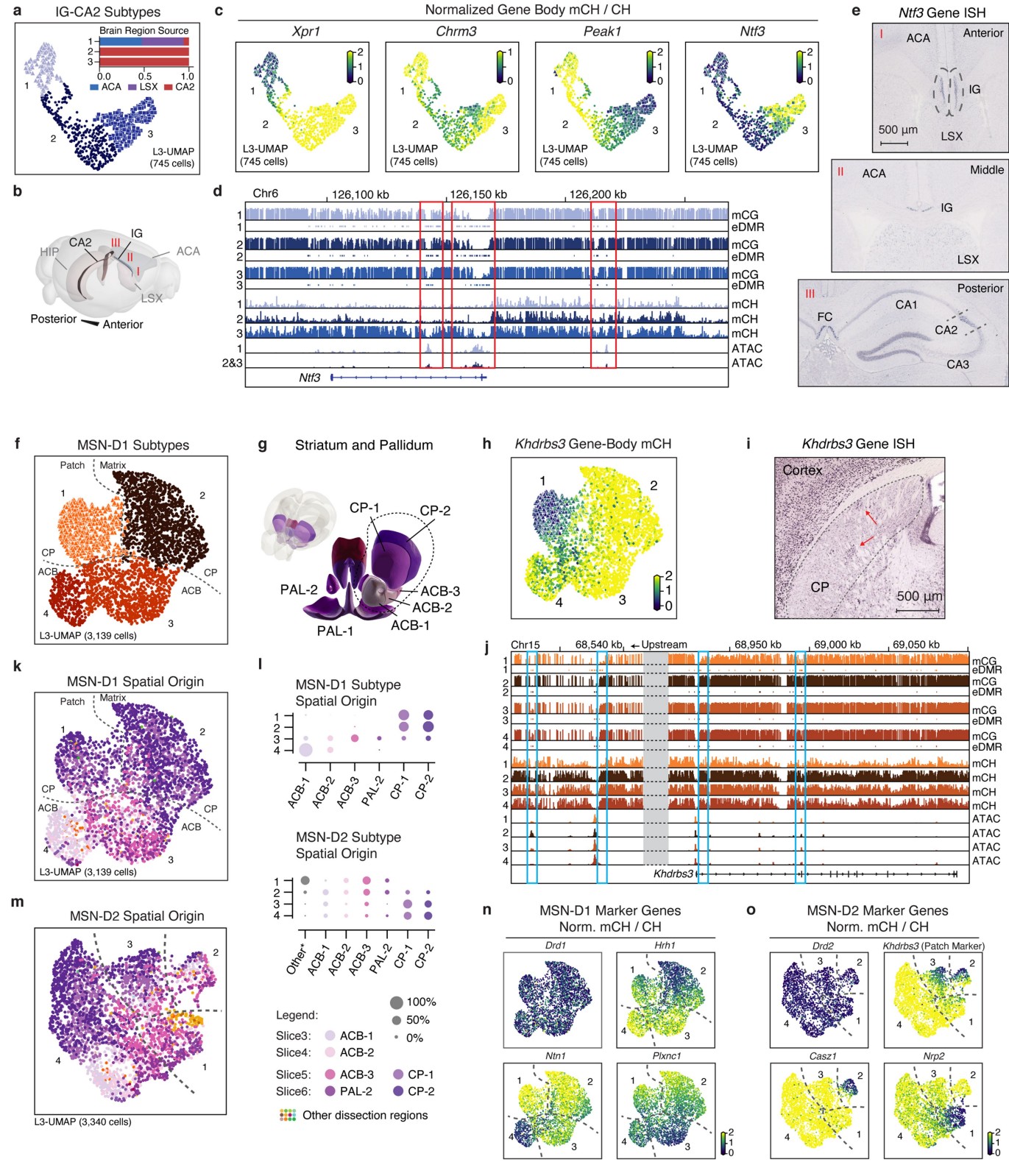

**Extended Data Fig. 4** | See next page for caption.

**Extended Data Fig. 4 | Supporting details of cellular and spatial diversity of neurons at the subtype level. a**, Level 3 UMAP of IG-CA2 neurons coloured by subtypes. Bar plot showing sub-region composition of the subtypes: (1) Xpr1, (2) Chrm3, and (3) Peak1. **b**, The 3D model illustrates the spatial relationships between related anatomical structures. **c**, mCH fraction of marker genes in IG-CA2 cells. **d**, Methylome, and chromatin accessibility genome browser view of *Ntf3* genes and its upstream regions. ATAC and eDMR information are from Fig. 4 analysis. **e**, Three different views of the in situ hybridization experiment (source: https://mouse.brain-map.org/gene/show/17972; the same patterns were shown in three biological replicates) from Allen Brain Atlas[48], showing the *Ntf3* gene expressed in both IG and CA2. **f**, Level 3 UMAP of MSN-D1 neurons coloured by subtype. Numbers indicate four subtypes: (1) Khdrbs3, (2) Hrh1 (3) Plxnc1, and (4) Ntn1. **g**, The 3D model of related striatum dissection regions. **h**, **i**, mCH fraction (**h**), and an in situ hybridization experiment (source: https://mouse.brain-map.org/gene/show/13769; the same patterns were shown in two biological replicates) from Allen Brain Atlas[48] (**i**) of the *Khdrbs3* gene, red arrows indicate patch regions in CP. **j**, Genome browser view of *Khdrbs3* genes similar to **d**. **k**, Level 3 UMAP of MSN-D1 neurons coloured by dissection regions. **l**, The region composition of each subtype of MSN-D1 and MSN-D2. **m**, MSN-D2 subtypes: (1) Nrp2, (2) Casz1, (3) Col14a1, and (4) Slc24a2. **n**, **o**, mCH fraction of MSN-D1 (**n**) and MSN-D2 (**o**) subtype marker genes. All brain atlas images (**b**, **g**) were created based on Wang et al.[16] and © 2017 Allen Institute for Brain Science. Allen Brain Reference Atlas. Available from: http://atlas.brain-map.org.

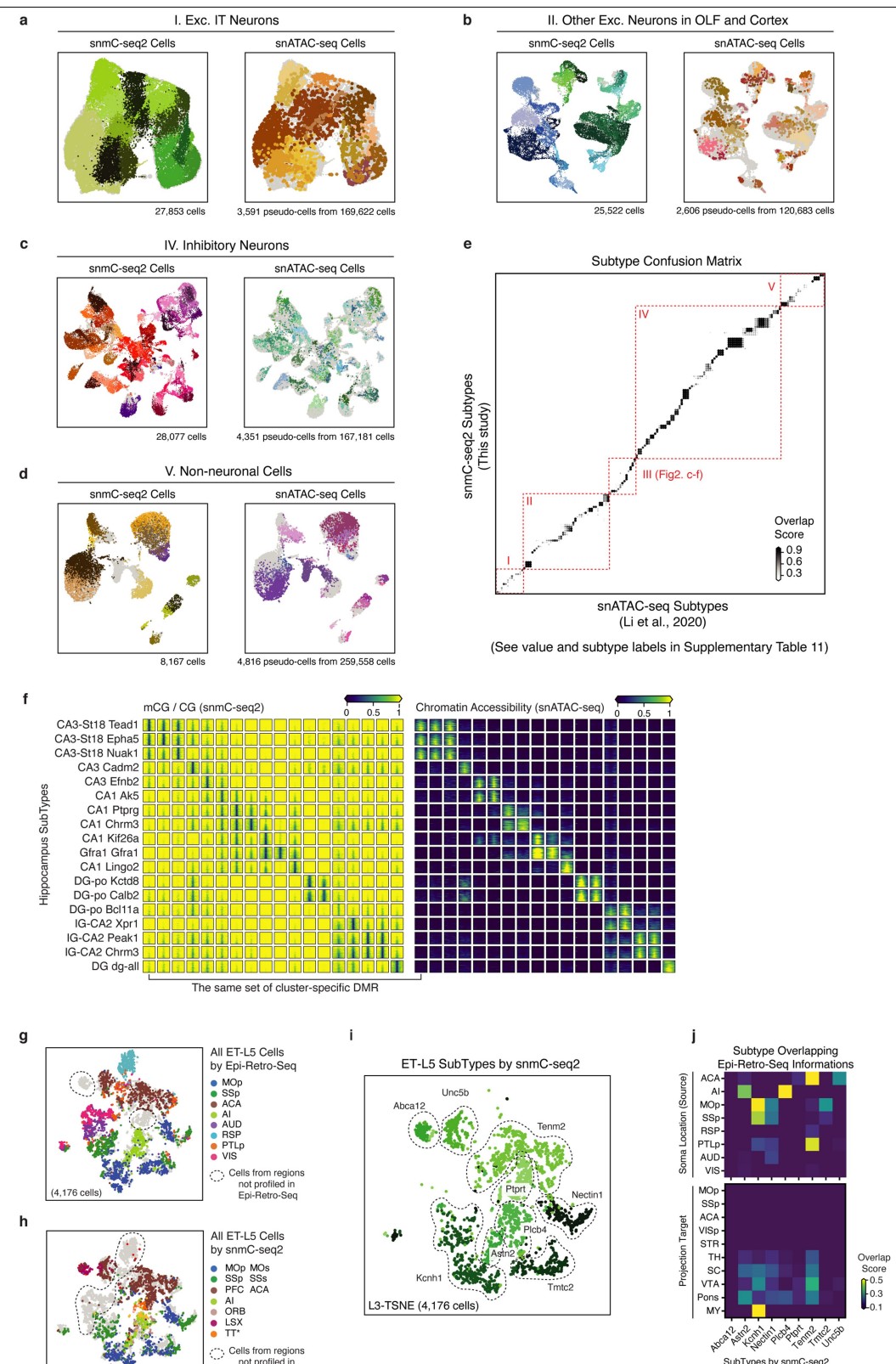

**Extended Data Fig. 5 | Integration with snATAC-seq and epi-retro-seq.**
**a**–**d**, Integration UMAP for snmC-seq2 cells and snATAC-seq pseudo-cells from each cell group: excitatory IT neurons (**a**), other excitatory neurons (**b**), inhibitory neurons (**c**) and non-neuronal cells (**d**). Each panel is coloured by subtypes from the corresponding study, the other dataset is shown in grey in the background. **e**, Overlap score matrix matching the 160 a-types to the 161 m-types. **f**, mCG fraction (left), and chromatin accessibility (right) of cluster-specific CG-DMRs (columns) in HIP subtypes (rows). **g**, **h**, Same integration *t*-SNE as Fig. 2g, i coloured by the dissection regions but using all cells profiled by epi-retro-seq (**g**) or snmC-seq2 (**h**), cells from brain regions that have only been profiled via one of the methods are circled out. **i**, Same *t*-SNE as (**h**) coloured by snmC-seq2 subtypes. **j**, Overlap score matrix matching the subtypes to the 'Soma Location (source)' and 'Projection target' information labels of epi-retro-seq cells.

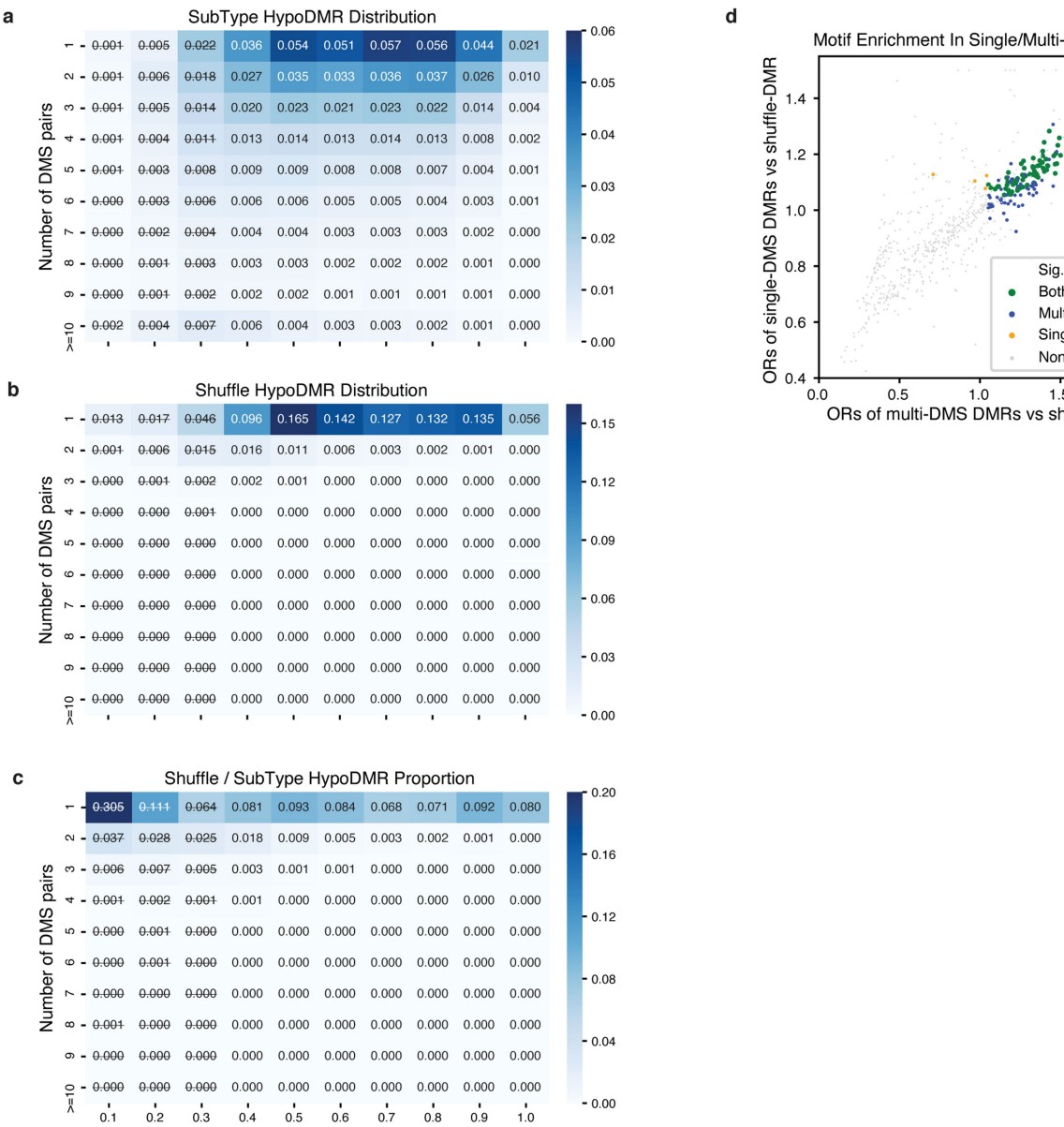

**Extended Data Fig. 6 | Controlling the FDR of CG-DMRs. a**, **b**, The proportion of subtype (**a**) or shuffled DMRs (**b**) in each block of specific effect size and number of DMSs. **c**, The empirical FDR of each block, calculated by (no. of shuffle-DMRs/no. of subtype-DMRs) in each block. DMRs with effect size <0.3 were excluded in further analyses. **d**, The odds ratio of transcription factor motif enrichment in single-DMS DMRs (sDMRs) and multi-DMS DMRs (mDMRs). Each dot represents a transcription factor. Transcription factors whose motifs are significantly enriched in both sDMRs and mDMRs are coloured in green, and transcription factors that are significant only in sDMR or mDMRs are coloured in red or blue, respectively. Non-significant transcription factors are shown in grey.

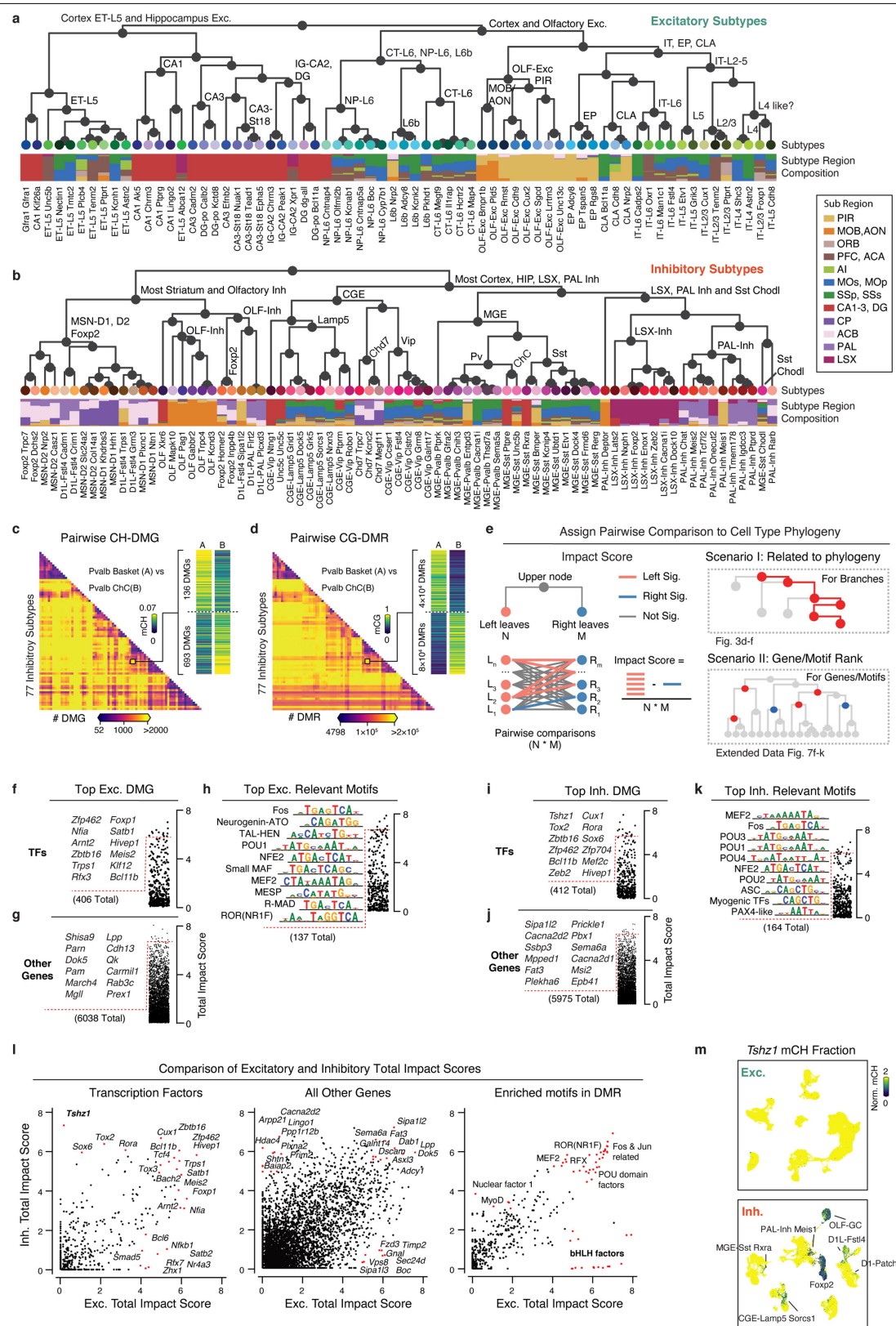

**Extended Data Fig. 7 | Subtype taxonomy with related genes and motifs.**
**a**, **b**, Subtype taxonomy of excitatory (**a**) and inhibitory (**b**) neurons. Leaf nodes are coloured by subtypes, and the bar plot shows subregion composition. **c**, **d**, Counts heat map of pairwise CH-DMG (**c**) and CG-DMR (**d**) between 77 inhibitory subtypes. **e**, Schematic of impact score calculation (left), and two scenarios of discussing impact scores (right). **f**–**h**, Top transcription factors (**f**), other genes (**g**) and enriched motifs (**h**) ranked by total impact score based on

the excitatory subtype taxonomy. **i**–**k**, Top transcription factors (**i**), other genes (**j**) and enriched motifs (**k**) ranked by total impact score based on the inhibitory subtype taxonomy. **l**, Comparison of the total impact scores calculated from either excitatory subtype taxonomy (*x*-axis) or inhibitory subtype taxonomy (*y*-axis) for transcription factors, other genes and enriched motifs. **m**, An example gene *Tshz1* only shows subtype diversity in inhibitory subtypes but not in excitatory subtypes.

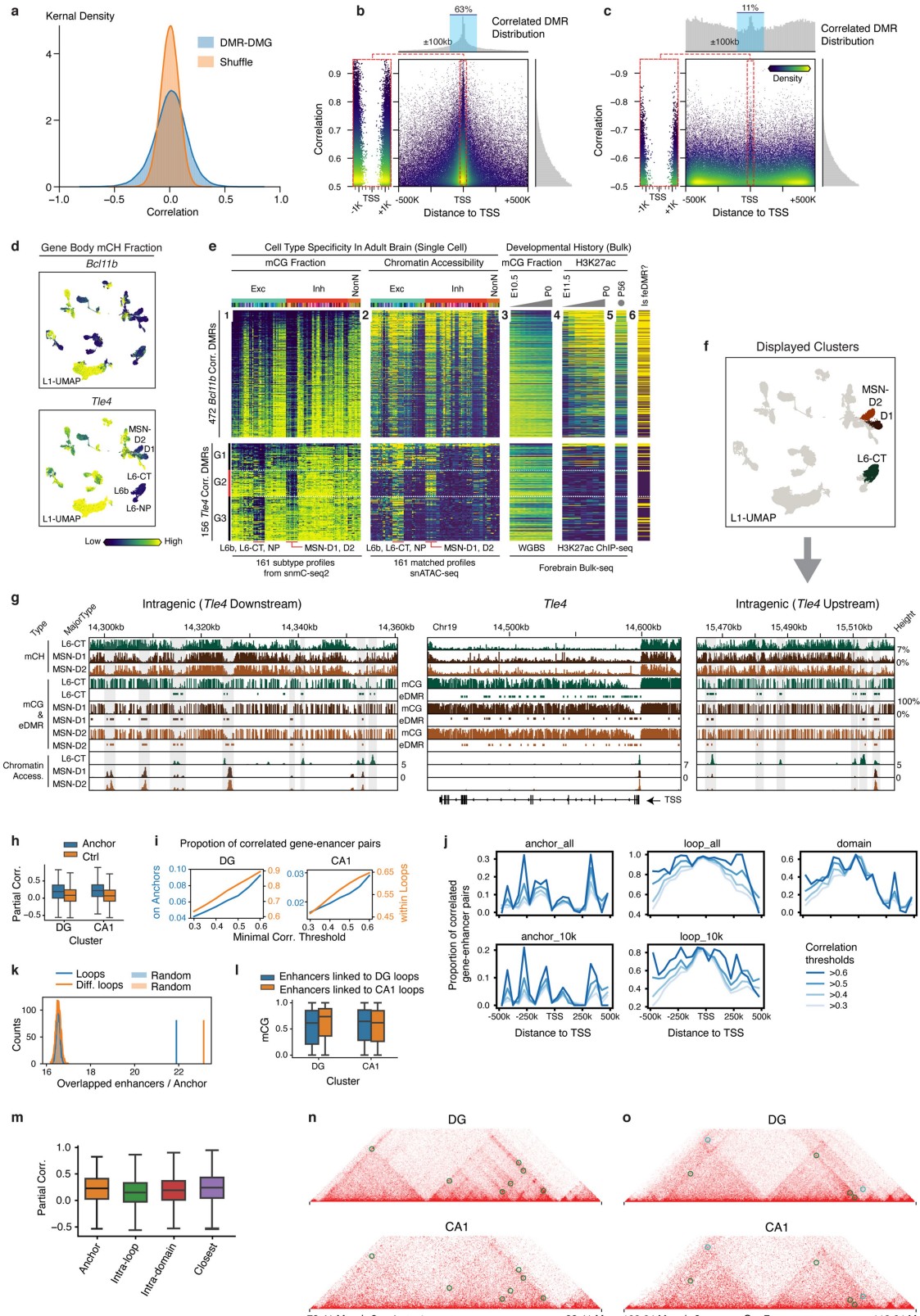

**Extended Data Fig. 8** | See next page for caption.

**Extended Data Fig. 8 | Gene-Enhancer landscape related. a**, Distribution of actual DMR–DMG partial correlation compared to the shuffled null distribution. **b**, **c**, DMR–DMG correlation (*y*-axis), and the distance between DMR centre and gene TSS (*x*-axis), each point is a DMR–DMG pair, colour represents points kernel density. The positively (**b**) and negatively (**c**) correlated DMRs are shown separately, owing to very different genome location distributions that are plotted on the top histograms. **d**, The gene body mCH fraction of *Bcl11b* (top) and *Tle4* (bottom) gene. **e**, The predicted enhancer landscape of *Bcl11b* (top) and *Tle4* (bottom). Each row is a correlated eDMR to the gene, columns from left to right are: (1) mCG fraction and (2) ATAC FPKM in 161 subtypes; (3) bulk developing forebrain tissue mCG fraction and (4) H3K27ac FPKM; (5) adult frontal cortex H3K27ac FPKM; and (6) feDMR or not. **f**, detailed view of surrounding eDMRs that are correlated with *Tle4* gene body mCH. Alternative eDMRs appear only in either CT-L6 or MSN-D1/D2 can be seen both upstream and downstream of the gene. **g**, Level 1 UMAP coloured by corresponding cell major types shown in **f**. **h**, Partial correlation between mCG of enhancers and mCH of genes on separated loop anchors of DG (left) and CA1 (right) compared to random anchors with comparable distance ($n$ = 4,171, 4,036, 4,326, 5,133 (left to right)), $P$ = 5.9 × 10$^{-74}$ for DG and 3.0 × 10$^{-158}$ for CA1, two-sided Wilcoxon rank-sum tests. **i**, Proportion of loop supported enhancer-gene pairs among the pairs linked by correlation analyses surpassing different correlation thresholds in DG (left) and CA1 (right). The proportion of pairs that the gene and enhancer located on separated anchors of the same loop (blue, left *y*-axis) or within the same loop (orange, right *y*-axis) is shown. **j**, Proportion of loop supported enhancer-gene pairs among those linked by correlation analyses surpassing different correlation thresholds at each specific distance. **k**, Number of enhancers per loop anchor (blue) or per differential loop anchor (orange) compared to randomly selected 25-kb regions across the genome.; $P$ < 0.005, two-sided permutation test with 2,000 times repeats. **l**, mCG of enhancers linking to DG specific loops (blue, $n$ = 13,854) and CA1-specific loops (orange, $n$ = 14,373) in DG (left, $P$ = 2.9 × 10$^{-3}$) or CA1 (right, $P$ = 3.5 × 10$^{-5}$). $P$ values were computed with two-sided Wilcoxon rank-sum tests. **m**, Partial correlation between mCG of enhancers and mCH of genes linked by different methods ($n$ = 4,171, 127,730, 28,203, 10,058 (left to right)). The elements of box plots are defined as: centre line, median; box limits, first and third quartiles; whiskers, 1.5 × interquartile range. **n**, **o**, Interaction maps, mCH, mCG, ATAC and differential loops tracks surrounding *Lrrtm4* (**n**) and *Grm7* (**o**). Circles on the interaction maps represent differential loops between DG and CA1, where green represents DG loops, and cyan represents CA1 loops.

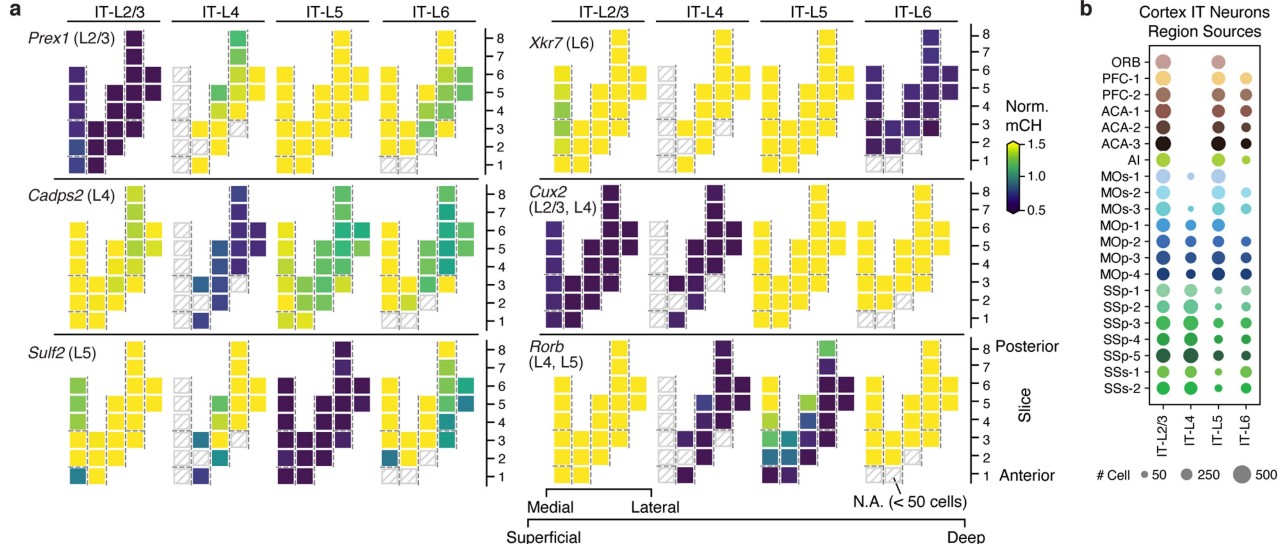

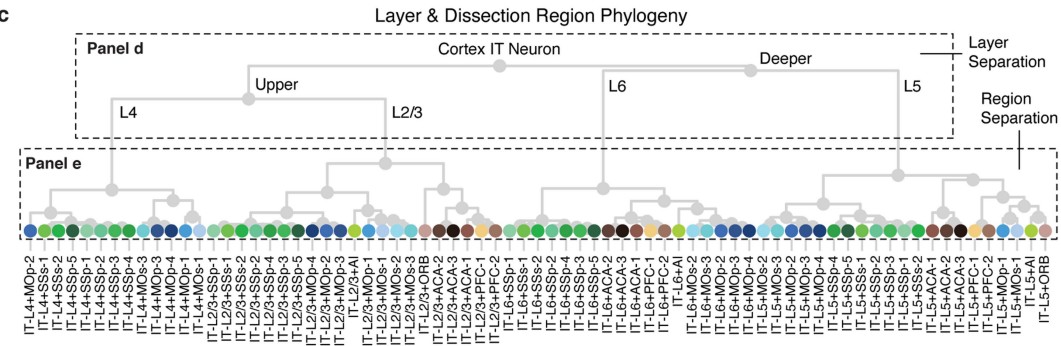

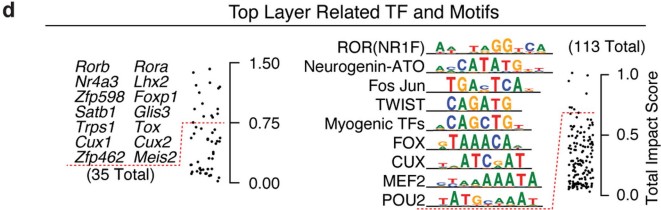

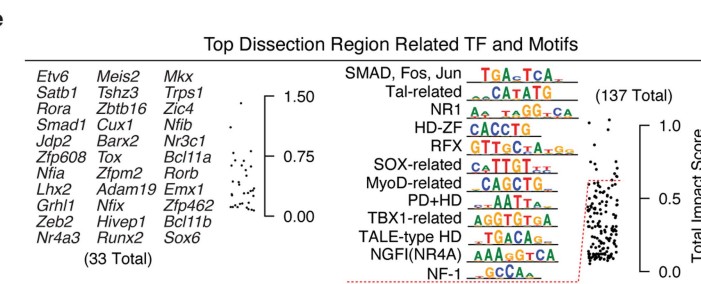

**Extended Data Fig. 9 | DNA methylation gradient of IT neurons.**
**a**, Representative marker genes for laminar layers separation. The same dissection region layout in Fig. 5b was used here. **b**, Layer-dissection-region cell group taxonomy. **c**, Dot plot sized by the number of cells in each layer-dissection-region combination in excitatory IT neurons. Each group needs at least 50 cells to be included in the analysis. **d**, **e**, The top layer (**d**) and dissection region (**e**) for related TFs and JASPAR motifs ranked by total impact score.

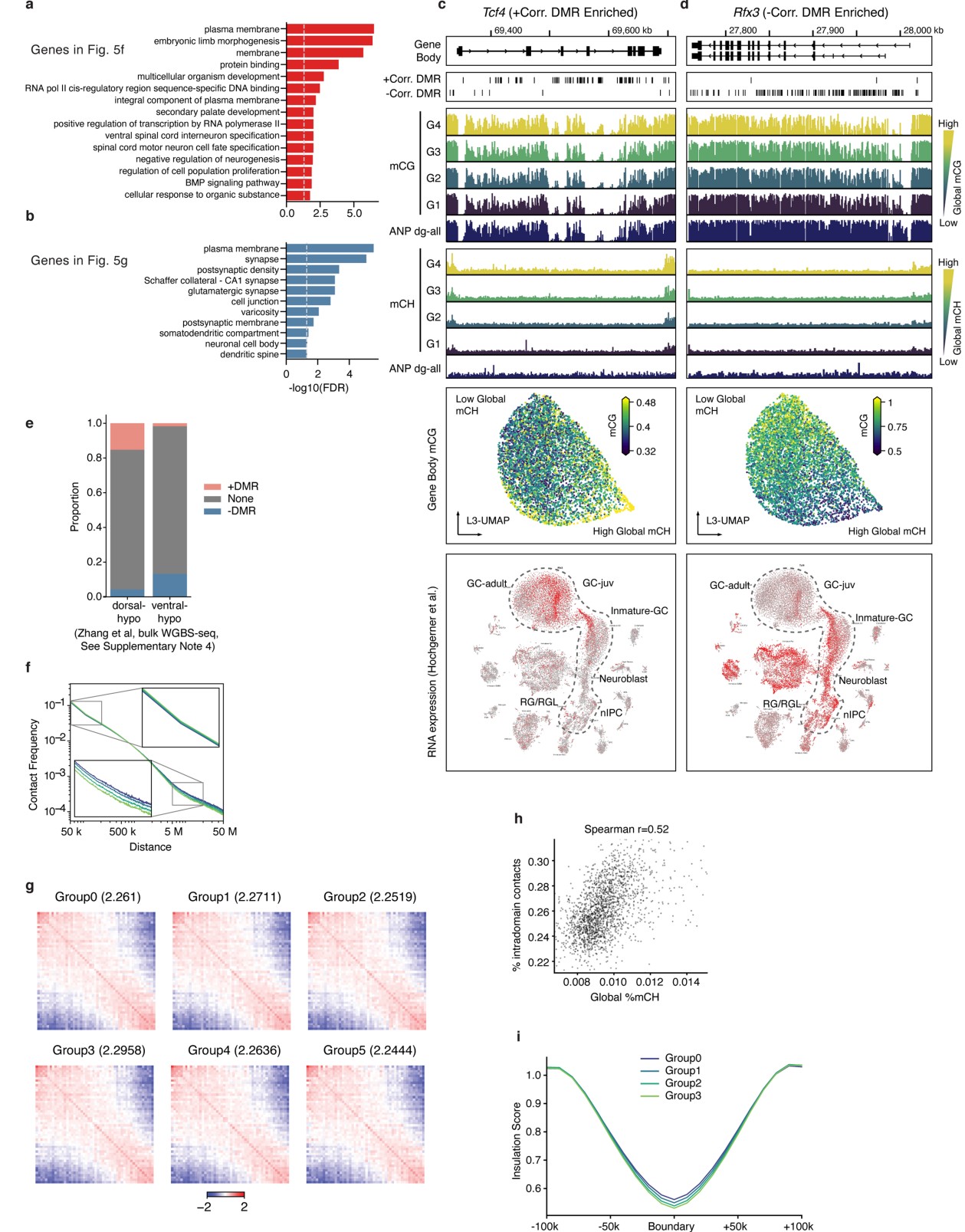

**Extended Data Fig. 10 | DNA methylation gradient of DG granule cells.**
**a**, **b**, Top enriched Gene Ontology (GO) terms for +DMRgenes (**a**) and −DMRgenes (**b**). Significant +DMRgenes and −DMRgenes are coloured in red and blue, respectively. **c**, **d**, For the +DMRgene *Tcf4* (**c**) or the −DMRgene *Rfx3* (**d**), the browser view of +DMRs or −DMRs, mCG or mCH in each DG cells groups and adult neural progenitors (ANP), L3 UMAP coloured by gene body mCG, and scRNA-Seq UMAP coloured by gene expression are shown. **e**, The proportion of dorsal or ventral DMRs that overlap with +DMRs and −DMRs. **f**, Interaction frequency decays with increasing genome distances in different groups. **g**, Saddle plots for different groups of DG cells separated by global mCH. Values in the title represent the compartment's strengths. **h**, Correlation between global mCH and proportion of intra-domain contacts across 1,904 DG cells. **i**, Insulation scores of 9,160 domain boundaries and flanking 100-kb regions.

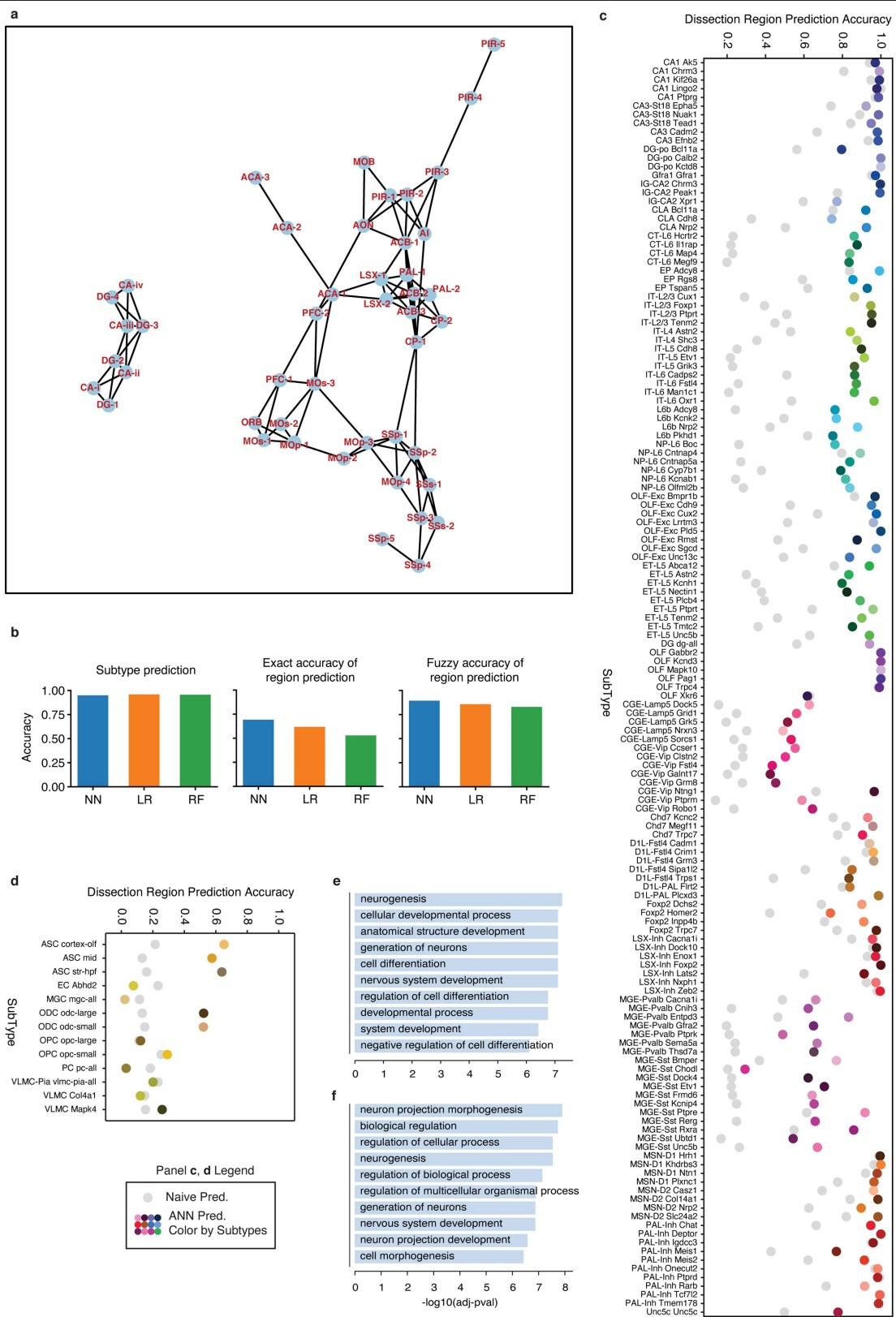

**Extended Data Fig. 11 | Evaluation of the predictive model. a**, The neighbour relation among the potential overlapping dissection regions. The network is constructed based on information of the dissection scheme and the 'Potential overlap' column in Supplementary Table 2 and is used to compute the fuzzy accuracy. **b**, The exact accuracy of subtype prediction (top), dissected region prediction (middle), and fuzzy accuracy of dissected region prediction (bottom) of neural network (NN, blue), logistic regression (LR, orange) and random forest (RF, green). **c**, **d**, Prediction accuracy of dissection region at cell subtype level of neurons (**c**) and non-neuronal cells (**d**). Coloured points denote the prediction accuracy of the model, whereas grey points denote the random guess accuracy when cell subtypes and corresponding spatial distributions are given. **g**, **h**, GO-term enrichment of top-loading genes of features that are important for predicting the spatial location of CT-L6 (**g**) and L6b (**h**).

# Reporting Summary

Nature Research wishes to improve the reproducibility of the work that we publish. This form provides structure for consistency and transparency in reporting. For further information on Nature Research policies, see our Editorial Policies and the Editorial Policy Checklist.

## Statistics

For all statistical analyses, confirm that the following items are present in the figure legend, table legend, main text, or Methods section.

| n/a | Confirmed | |
|---|---|---|
| ☐ | ☒ | The exact sample size (*n*) for each experimental group/condition, given as a discrete number and unit of measurement |
| ☐ | ☒ | A statement on whether measurements were taken from distinct samples or whether the same sample was measured repeatedly |
| ☐ | ☒ | The statistical test(s) used AND whether they are one- or two-sided *Only common tests should be described solely by name; describe more complex techniques in the Methods section.* |
| ☐ | ☒ | A description of all covariates tested |
| ☐ | ☒ | A description of any assumptions or corrections, such as tests of normality and adjustment for multiple comparisons |
| ☐ | ☒ | A full description of the statistical parameters including central tendency (e.g. means) or other basic estimates (e.g. regression coefficient) AND variation (e.g. standard deviation) or associated estimates of uncertainty (e.g. confidence intervals) |
| ☐ | ☒ | For null hypothesis testing, the test statistic (e.g. *F*, *t*, *r*) with confidence intervals, effect sizes, degrees of freedom and *P* value noted *Give P values as exact values whenever suitable.* |
| ☒ | ☐ | For Bayesian analysis, information on the choice of priors and Markov chain Monte Carlo settings |
| ☒ | ☐ | For hierarchical and complex designs, identification of the appropriate level for tests and full reporting of outcomes |
| ☐ | ☒ | Estimates of effect sizes (e.g. Cohen's *d*, Pearson's *r*), indicating how they were calculated |

*Our web collection on statistics for biologists contains articles on many of the points above.*

## Software and code

Policy information about availability of computer code

| Data collection | BD Influx Sortware v1.2.0.142 (flow cytometry), Freedom EVOware v2.7 (library preparation), Illumina MiSeq control software v3.1.0.13 and NovaSeq 6000 control software v1.6.0/RTA v3.4.4 (sequencing), Olympus cellSens Dimension 1.8 (image acquisition) |
|---|---|
| Data analysis | bedtools 2.27, methylpy 1.4.2, scanpy 1.4.3, juicer tools 1.14.08, REPTILE (https://github.com/yupenghe/REPTILE.git), scHiCluster (https://github.com/zhoujt1994/scHiCluster.git) the mapping pipeline for snmC-seq2 data: https://cemba-data.readthedocs.io/en/latest/, including the following packages: bismark 0.20, bowtie2 2.3, cutadapt 1.18, picard 2.18, samtools 1.9, htslib 1.9; The ALLCools package for post-mapping analysis and snmC-seq2 related data structure: https://github.com/lhqing/ALLCools; The jupyter notebooks for specific analysis: https://github.com/lhqing/mouse_brain_2020. |

For manuscripts utilizing custom algorithms or software that are central to the research but not yet described in published literature, software must be made available to editors and reviewers. We strongly encourage code deposition in a community repository (e.g. GitHub). See the Nature Research guidelines for submitting code & software for further information.

## Data

Policy information about availability of data

All manuscripts must include a data availability statement. This statement should provide the following information, where applicable:
- Accession codes, unique identifiers, or web links for publicly available datasets
- A list of figures that have associated raw data
- A description of any restrictions on data availability

Single-cell raw and processed data included in this study were deposited to NCBI GEO/SRA with accession number GSE132489 (each experiment has a separate accession number recorded in GSE132489, see Supplementary Table 12), and to the NeMO archive: https://assets.nemoarchive.org/dat-vmivr5x. Single-cell

# Field-specific reporting

Please select the one below that is the best fit for your research. If you are not sure, read the appropriate sections before making your selection.

☒ Life sciences ☐ Behavioural & social sciences ☐ Ecological, evolutionary & environmental sciences

For a reference copy of the document with all sections, see nature.com/documents/nr-reporting-summary-flat.pdf

# Life sciences study design

All studies must disclose on these points even when the disclosure is negative.

| | |
|---|---|
| Sample size | At least 3,072 nuclei (eight 384-well plates) from each dissected region (1,536 nuclei from each replicate). The sample size allowed us to obtain high coverage methylomes for each subtype, and perform confident downstream analyses. |
| Data exclusions | We filtered the cells based on these main mapping metrics: 1) mCCC level < 0.03, 2) overall mCG level > 0.5, 3) overall mCH level < 0.2, 4) total final reads > 500,000, 5) bismark mapping rate > 0.5. Other metrics such as genome coverage, PCR duplicates rate, index ratio were also generated and evaluated during filtering. However, after removing outliers with the main metrics 1-5, few additional outliers can be found. Note the mCCC level is used as the estimation of the upper bound of bisulfite non-conversion rate. The criterion include pre-established ones in Luo. et al 2018, and new ones to exclude additional outliers as justified in the manuscript. |
| Replication | Each dissected region has at least two replicates, each replicate was pooled from 6-30 animals separately for nuclei preparation and downstream analyses. Data are highly consistent between replicates (Extended Data Fig. 2d-g). |
| Randomization | Randomization is not applicable, since the cells collected are random by nature. |
| Blinding | Blinding is not applicable, since all data are collected from mice. |

# Reporting for specific materials, systems and methods

We require information from authors about some types of materials, experimental systems and methods used in many studies. Here, indicate whether each material, system or method listed is relevant to your study. If you are not sure if a list item applies to your research, read the appropriate section before selecting a response.

## Materials & experimental systems

| n/a | Involved in the study |
|---|---|
| ☐ | ☒ Antibodies |
| ☒ | ☐ Eukaryotic cell lines |
| ☒ | ☐ Palaeontology and archaeology |
| ☐ | ☒ Animals and other organisms |
| ☒ | ☐ Human research participants |
| ☒ | ☐ Clinical data |
| ☒ | ☐ Dual use research of concern |

## Methods

| n/a | Involved in the study |
|---|---|
| ☒ | ☐ ChIP-seq |
| ☐ | ☒ Flow cytometry |
| ☒ | ☐ MRI-based neuroimaging |

## Antibodies

| | |
|---|---|
| Antibodies used | AlexaFluor488-conjugated anti-NeuN antibody (MAB377X, Millipore) |
| Validation | All antibodies have been previously published for use in immunohistochemistry and flow cytometry experiments. See vendor's page here: https://www.emdmillipore.com/US/en/product/Anti-NeuN-Antibody-clone-A60-Alexa-Fluor488-conjugated,MM_NF-MAB377X |

## Animals and other organisms

Policy information about studies involving animals; ARRIVE guidelines recommended for reporting animal research

| | |
|---|---|
| Laboratory animals | Adult (P56) C57BL/6J male mice. Housing condition: Temperature: 21-23 C, relative humidity: 61-63% |
| Wild animals | the study did not involve wild animals |

| Field-collected samples | the study did not involve samples collected from the field |
|---|---|
| Ethics oversight | All experimental procedures using live animals were approved by the Salk Institute Animal Care and Use Committee under protocol number 18-00006. |

Note that full information on the approval of the study protocol must also be provided in the manuscript.

# Flow Cytometry

## Plots

Confirm that:

☒ The axis labels state the marker and fluorochrome used (e.g. CD4-FITC).

☒ The axis scales are clearly visible. Include numbers along axes only for bottom left plot of group (a 'group' is an analysis of identical markers).

☒ All plots are contour plots with outliers or pseudocolor plots.

☒ A numerical value for number of cells or percentage (with statistics) is provided.

## Methodology

| Sample preparation | Isolated nuclei were labeled by incubation with 1:1000 dilution of AlexaFluor488-conjugated anti-NeuN antibody (MAB377X, Millipore) and a 1:1000 dilution of Hoechst 33342 at 4°C for 1 hour with continuous shaking. Fluorescence-Activated Nuclei Sorting (FANS) of single nuclei was performed using a BD Influx sorter with an 85μm nozzle at 22.5 PSI sheath pressure. Single nuclei were sorted into each well of a 384-well plate preloaded with 2 μl of Proteinase K digestion buffer (1μl M-Digestion Buffer, 0.1μl 20 μg/μl Proteinase K and 0.9μl H2O). The alignment of the receiving 384-well plate was performed by sorting sheath flow into wells of an empty plate and making adjustments based on the liquid drop position. Single-cell (1 drop single) mode was selected to ensure the stringency of sorting. For each 384-well plate, columns 1-22 were sorted with NeuN+ (488+) gate, and column 23-24 with NeuN- (488-) gate, reaching an 11:1 ratio of NeuN+ to NeuN- nuclei. |
|---|---|
| Instrument | BD Influx |
| Software | BD Influx Sortware v1.2.0.142 |
| Cell population abundance | We sort NeuN+ (488+) gate and NeuN- (488-) gate with an 11:1 ratio into each 384-well plate. |
| Gating strategy | Intact nuclei were first discriminated from debris by virtue of their bright DNA labeling (Hoechst Height signal) followed by light scattering profiles (Forward Scatter (FSC) Height vs Side Scatter (SSC) Height). Events with high Pulse Width measurements for FSC and SSC were then excluded as aggregates. Next, NeuN-AlexaFluor 488 positive or negative nuclei were selected, reaching an 11:1 ratio of NeuN+ to NeuN- nuclei. |

☒ Tick this box to confirm that a figure exemplifying the gating strategy is provided in the Supplementary Information.

