## [Peer Review File · Nature]

Manuscript Title: DNA Methylation Atlas of the Mouse Brain at Single-Cell Resolution

Reviewer Comments & Author Rebuttals**Reviewer Reports on the Initial Version:**

Referees' comments:

Referee #1 (Remarks to the Author):

In this manuscript, the authors present a single nucleus DNA methylation dataset of over 110K nuclei from 45 regions of the mouse brain. In addition to a very useful resource for the field, they report some interesting findings. They identified 161 cell clusters with distinct spatial locations and projection targets and showed that the methylation landscape of excitatory neurons in the cortex and hippocampus varied continuously along spatial gradients. They further integrated the DNA methylomes with the single cell chromatin accessibility to predict the enhancer-gene interaction for given cell types and combined the Hi-C dataset to annotate the regulatory genome of those cell types from the mouse brain. This is a comprehensive datasets and will be very useful for the field. I have several comments for the authors to improve their manuscript.

Specific comments:

1. Summary: The authors used NeuN-based sorting and analyzed mostly neuronal methylome at the single-cell level (92%) and only 8% cells were NeuN- non neuronal cells. The coverage of non neuronal cell types is much less compared to neurons and it is not the focus of the current study. This information should be clear in the summary for the general readers.
2. In the introduction, some references should be cited. The first paper identified nonCpG in the brain is PMID: 22608086. The first paper identified MeCP2 as the nonCpH reader is PMID: 24362762, which was then characterized in detail in PMID: 28498846. It should be also acknowledge that 5hmC is highly abundant in the brain and will not be differentiated from 5mC in the current study.
3. What reference source did the author use to annotate the cell types from the methylation data?
4. Please provide QC comparison on sn-m3C-seq, sn-mC-seq2, sc-ATAC-seq, in terms of how many reads/cell, genes/cell, Pearson correlation among individual sample as a supplementary table.
5. The authors need to do a better job to sequentially refer to figures in the text. For example, Fig. 2a and 2b were not mentioned before Fig. 2c.
6. It will be useful to validate methylation detection for both mCG sites and mCH sites in a cell-type specific context using a different approach.
7. When mapping methylome data using bismark with bowtie, did the author allow mismatching? How many mismatched was accepted? This is particularly important for mCH calling.
8. For the ANP anp-dg cluster, are they most similar to DG granule neurons?
9. Can the authors provide example of functional validation of their predicted enhancer-gene pairs?
10. The high overlapping of DMRs and open chromatin regions in the hippocampus confirmed the idea that the open chromatin regions are mostly cis-regulatory regions, but not necessary confirmed the correct match of cell-type identities. The author should modify their statement or give more explanation.
11. The analysis on key transcriptional factor identification in specific cell types is interesting. Considering the low expression level of TFs in single cell RNA-seq dataset, the analysis in this study provide a unique angle on cell-type specification. It will be interesting to perform an integrated analysis on those candidate TFs' ChIP-seq dataset.
12. Is the role of mCG and mCH in cell type specification different or similar? Are there any genomic region of DMRs, like promoters, enhancers, TSSs or gene bodies, play predominant roles?
13. For sn-m3C-seq analysis, were nuclei sorted by NeuN?
14. Several genes show a gradient of expression along the dorsal-ventral (septo-temporal) axis in the

adult mouse gyrus, for example sfrp3 (PMID: 26337530). How does the methylation status of these genes look like in UMAP (Fig. 5g)?

15. Are there sufficient cells to train the artificial neural network to predict non neuronal cells? Do non neuronal cells exhibit strong region specificity?

Referee #2 (Remarks to the Author):

Summary

Liu and Zhou et al. present a large-scale epigenetic landscape of the mouse brain in single cells. Specifically, more than 100,000 nuclei from 45 annotated regions within the cortex, hippocampus, striatum, pallidum, and olfactory spaces. This manuscript in conjunction with associated companion manuscripts, take multiple "views" of the same tissue using previously published epigenetic and structural 'omics approaches at varying resolutions to provide a more comprehensive analysis of the data. Overall, the data and analysis are well carried out and address a critical need in the field. As evidenced by the multiple companion manuscripts, the sheer scale of the data and necessary analyses are somewhat prohibitive in a single paper to move to more granular understanding and validation of predicted subtypes. Nevertheless, this paper represents a critical resource that will be generally of interest to the field. Further, the authors should be commended for making the data and code available for such a valuable resource, although the sn-m3C-seq data seem to be missing from the NeMO website; I was only able to find the snmc-seq data. There is also a visualization website that presents the data in a more accessible form. However, with this volume and high dimensionality of the data, it is clear that they faced similar challenges of many such large-scale papers, where it is often hard to get into the granularity, and cover enough details.

Main comments:

1) Binning: Lines 112 to 114 discuss the level of methylation coverage represented across the mouse genome in 100-kb bins being on average 95% with gene bodies at 81% on average. This is highly confusing and somewhat misleading as currently written, since the data were summarized into 100-kb bins and the discrepancy between genome-wide and gene body percentages is unclear. Each cell received about 1.5M reads (after filtering), or ~6% of the genome. Using 100-kb bins helps reducing the impact of such sparsity in the data, but at the 100-kb size range, different information is captured than at more granular levels. Long-range effects such as PMD hypomethylation will tend to dominate, while smaller features such as CpG islands, promoters, and enhancers will be lost. Despite the binning, heavy imputation was needed nonetheless (and performed; Line 742 onwards). It seems the global mean of the cell was used as the prior. The rationale for this choice is not clear. If anything, it should be the mean per locus across the cells. And why impute features with less cov (coverage?, not covariance or coefficient of variation, correct?) to have a mean methylation level value close to 1? It would be helpful to show the number of CpG or CpH sites covered by at least one read, since tiling the genome at a single resolution introduces a bin-size bias in the representation of the data. There may be added value to varying the bin sizes and combining them for a more multi-scale view of the data, as opposed to PMD-level binning, given the size of the data. That being said, going back and reanalyzing for TF motifs and DMRs/DMGs was great and does address this concern to some extent.

2) Filtering: The filtering choices are not always clear. For example, 100-kb genomic bin features were filtered by removing bins with mean total cytosine base calls < 250 or > 3000. It makes sense to filter the sparse bins, but why filter bins with high coverage? Is this an attempt to remove CpG islands? Samples were filtered based on a mCCC rate of 3%. It is not clear that this is good enough for studies that focus on mCH methylation, where the average level falls well below 3%. In addition, another filter of

20% mCH was applied. It is curious that after the 3% mCCC filtration there would still be cells with >20% mCH methylation (how often?). This would actually indicate that there's true CpH methylation in those cases?

3) Neuron Classification: How are the excitatory and inhibitory neurons classified in Figure 1b and 1c? These are ad hoc drawn clusters so presumably a priori information was used to delineate these cell types. However, this is not described well in the manuscript. This is critical given it is the highest level in the hierarchical clustering performed that underlies the main conclusions of the paper.

4) The authors should be commended for their robust exploration of clustering parameters by a grid search for hyperparameter (namely the resolution parameter) tuning - however it remains unclear how the cluster labels for each level of iterative cluster were generated with regards to cell type/subtype.

5) The initial data matrix construction of PCs for downstream clustering analysis was done using the elbow method, which is somewhat arbitrary. How many PCs were chosen? The plots of variance explained for the mc rates, along with an indication of the cutoff used, need to be shown as a supplemental figure for possible reproducibility. Additionally, splitting up the CG and CH methylation into separate matrices, performing PCA, and concatenating the matrices is unclear in the motivation to do so. Specifically, the correlation structure between the two types of methylation is destroyed despite having biological ties to one another. The correlation between the PCs from CG and CH also should be shown to understand the degree to which this correlation exists between them. Why not use NMF or ICA as an initialization using the full matrix (e.g. both CG and CH normalized bins)? Finally, why initialize with PCA as opposed to the normalized, 100kb binned methylation values since the binning approach drastically reduces the total number of features per cell, which is often the goal of data compression with something like PCA?

6) The use of a neural network to simultaneously predict subtypes and dissection regions is great. However, the motivation or need to use a neural network for the prediction approach versus something more interpretable like a penalized regression model is not clear. This would allow a somewhat more direct interpretation of the contribution of individual features. Does the neural network outperform the use of somewhat "simpler" regression models?

7) DMRs: The total number of significant DMRs found is startling - using a permutation based goodness of fit approach is intriguing, but the ability to control type I error is unclear with so many sites and cells/subtypes. This needs to be commented on to discuss the possible impact of false discovery rate in such a large-scale analysis. Using randomly sampled groups there were 2,003 DMRs, as shown in Fig 5H, and removing them from the gradient DMR set was an interesting approach but the logic is not entirely clear. One would not expect the same false positives in both experiments, likely also the reason for the minimal overlap. What is the FDR rate (2,003/how many total)? Knowing the FDR rate and keeping the whole set is going to be more informative than removing a mere 0.04% overlap with DMRs from scrambled setup.

8) The use of gene-body CpH methylation as a surrogate for inverse expression levels, and correlating it with CpG DMRs as surrogate or regulatory elements is clever. However, gene-body CpG methylation is positively correlated with gene expression. Why not use the more abundant (and therefore higher signal-to-noise ratio) CpG methylation, rather than the noisier CpH methylation? More importantly, CpH level is dependent on cell type and there are cell types with much lower level or no detectable CpHs. Gene body CpH levels in those cells are overall lower than those with high DNMT3A (and high global CpH), but not directly associated with host gene expression level difference. Also, it was stated that there were sn-RNA-seq data, and I feel those data could have been better used in this part of analysis - it could have been used for expression in this model, or if not, for assessing the suitability of gene body CpH and CpG methylation as surrogates for expression in this analysis, or as validation for discoveries.

9) The use of "rate" for methylation levels throughout the manuscript is confusing, since the word "rate" implies a time-dependent function. In most cases, the word "rate" could be replaced by the word "fraction". I realize that the authors have used this terminology in past papers, but I am not enthusiastic about promoting this usage. Scientists studying other types of genomic events have moved away from the use of the word "rate" and tend to use words like "density", "frequency", or "load", as in genomic mutational density per Mb, as opposed to mutation rate, which would instead be per cell division or time unit.

10) sn-m3C-seq: R1 and R2 reads were mapped separately, and they then claimed a higher mapping rate than previous methods. However, mapping separately would achieve that. They split up the unmapped reads into first 40bp, middle segment and final 40bp, likely to deal with the 3C part. There may not be that many options with Bismark, but HiC data processing methods may reveal smarter ways of doing this than just the 40bp split - e.g. searching for ligation site etc. It is not clear how they dealt with chimeric reads generated by Klenow to distinguish them from chromatin contacts.

11) Controls: The authors should be commended for including controls for most analyses. However, sometimes the control did not seem to be used properly (the Figure 5h example above), or the results were not impressive compared to controls. For example, in Figure 4k, the observed data (22-23 is not very high level of enrichment compared to random (16.5). Similar with 4h. Statistical significance is one thing and easy to achieve when there are so many data points in the genome, but if the effect size is not that big it calls into question how good the algorithm actually is. Sometimes controls are not included when they could. e.g., mCCC plots as controls for bisulfite conversion in a lot of the mCH analyses.

Minor comments:

1) Lines 134-136 about using UMAP to visualize differences based on cell location is unclear given that UMAP optimizes a cost function for both local and global distances rendering "location" somewhat confounded with the general classification of different brain regions. Were global distances weighted in the UMAP embeddings to account for spatial distance?

2) Line 152 suggesting high RNA expression of Unc5c given the low methylation - this needs to be evaluated in terms of actual gene expression from the Allen Brain Institute scRNA-seq brain atlas.

3) Line 243, how was 0.3 chosen as a reasonable threshold for impact score and thus, gene assignment to each branch in the tree?

4) Line 768, the word "significant" is used without any association with a p-value or other significance value.

5) Some of the results were puzzling and not explained very well. For example, why would CpH methylation correlate with # of intra-domain contacts? Does the same trend show up if mCCCs are plotted instead of CpH? Or in Fig. 4h, why would the number of enhancer-gene pairs differ so much between the two cell types? Or, in Fig 3i and 3k - why are the TF motifs different for CpH-DMGs and CpG-DMRs for the same cluster?

6) In figure 3A, 2 of the 3 CG-DMRs for BCL11b seemed to be at about 50% methylated in cell type B. This seems to suggest cell type B is still heterogeneous (or this gene is monoallelicly methylated). Since this is single cell data, can the authors look within B and see if it was indeed a mixture of methylated and unmethylated cells in B, or if it was monoallelic methylation.

7) In figure 2, m/n/p panel colors - do these stand for subtype assignments? Many colors are so similar that it is hard to assess the UMAPs

Author Rebuttals to Initial Comments:**Referee #1 (Remarks to the Author):**

In this manuscript, the authors present a single nucleus DNA methylation dataset of over 110K nuclei from 45 regions of the mouse brain. In addition to a very useful resource for the field, they report some interesting findings. They identified 161 cell clusters with distinct spatial locations and projection targets and showed that the methylation landscape of excitatory neurons in the cortex and hippocampus varied continuously along spatial gradients. They further integrated the DNA methylomes with the single-cell chromatin accessibility to predict the enhancer-gene interaction for given cell types and combined the Hi-C dataset to annotate the regulatory genome of those cell types from the mouse brain. This is a comprehensive dataset and will be very useful for the field. I have several comments for the authors to improve their manuscript.

R1.1

We thank the reviewer for appreciating the significance and impact of our manuscript and providing helpful comments to improve our study. In response to the reviewer's concerns raised below, our major efforts have focused on constructing a comprehensive web application (<http://neomorph.salk.edu/omb>) to help the general audience access the description and marker genes of the cell types we annotated and expanding the discussion on the cell types we described. Specifically, we added more discussion on the methylation gradient of dentate gyrus granule cells, which suggests the genes showing methylation variation that correlated with global mCH changes are related to granule cell maturation. Below we provide detailed responses to each of the comments.

Specific comments:

1. Summary: The authors used NeuN-based sorting and analyzed mostly neuronal methylome at the single-cell level (92%), and only 8% of cells were NeuN- nonneuronal cells. The coverage of non-neuronal cell types is much less compared to neurons, and it is not the focus of the current study. This information should be clear in the summary for the general readers.

R1.2

We have added the following sentence to the summary paragraph to clarify the number of neuron/non-neuronal cells:

“We carried out a comprehensive assessment of the epigenomes of mouse brain cell types by applying single nucleus DNA methylation sequencing to profile 103,982 nuclei (including 95,815 neurons and 8,167 non-neuronal cells) from 45 regions of the mouse cortex, hippocampus, striatum, pallidum, and olfactory areas.”

2. In the introduction, some references should be cited. The first paper identified nonCpG in the brain is PMID: 22608086. The first paper identified MeCP2 as the nonCpH reader is PMID: 24362762, which was then characterized in detail in PMID: 28498846.

R1.3.1

We have added these important references to the introduction.

(Note from the authors about “the first paper to describe nonCpG in the brain”. After revisiting the reviewers suggested citation (PMID: 2260806), we found this study to irrelevant to the topic. The paper (PMID 2260806) describes profiling of 5hmC in embryonic stem cells using the TAB-seq technology, no brain tissue was profiled in the manuscript. Instead we have cited Hon et al. (PMID: 23995138) for mouse brain and Lister et al. for human brain (PMID: 23828890) as the appropriate citations)

It should also be acknowledged that 5hmC is highly abundant in the brain and will not be differentiated from 5mC in the current study.

R1.3.2

We have added a paragraph in the discussion as follows:

“Notably, snmC-seq2 is a sodium bisulfite-based method which does not distinguish between 5-methylcytosine and 5-hydroxymethylcytosine (Huang et al., 2010), which has been shown to accumulate to significant levels in certain brain regions (Khare et al., 2012; Szulwach et al., 2011). New high-throughput methods will be needed to simultaneously measure the full complement of cytosine base modifications in single brain cells.”

3. What reference source did the author use to annotate the cell types from the methylation data?

R1.4

For annotation purposes, we did not use a single database but instead used a combination of well-described marker genes, dissection location, and gating information from sorting. Below we describe the information used for cell-type annotation. All details about cell type annotation can be found in Supplementary Table 7. To help the general audience better access these detailed annotations, we have developed a web application to interactively visualize the gene-level DNA methylation and metadata of single cells (e.g., browser page for IT-L2/3, http://neomorph.salk.edu/omb/cell_type?ct=IT-L23). Each cell type has a corresponding data browser, whose URL can also be found in the Supplementary Table 7.

1. Cell class (level 1 clustering) annotation.

We annotated non-neuronal cells based on both the NeuN- gate origin and low global mCH fraction (Fig. R1). Given the strong anti-correlation between CH methylation and

gene expression, we used hypo-CH-methylation at gene bodies \pm 2 kb (-2 kb from TSS and +2 kb from TES) of pan-excitatory markers such as *Slc17a7*, *Sv2b* and pan-inhibitory markers such as *Gad1*, *Gad2* (Fig. R2) to annotate excitatory and inhibitory cell classes, respectively.

Fig. R1. Non-neuronal cell class identification. **a, b**, L1-UMAP embedding of all cells pass QC (n=103,982), colored by NeuN+/- gate information during FANS (a) and cell global mCH fraction (b). **c**, scatter plot of major cell type mean NeuN-% (x-axis) and global mCH fraction (y-axis), non-neuronal cell types have low global mCH and high NeuN-%.

Fig. R2. Pan-excitatory and pan-inhibitory marker genes. Colors showing z-score (per row) of major cell types' gene body mean mCH fraction.

2. Major type (level 2) and subtypes (level 3) annotations.

Here we used gene body ± 2 kb hypo-CH-methylation (or hypo-CG-methylation for non-neurons) of well-known marker genes along with dissection information to annotate neuron and non-neuron clusters. All cluster marker genes are listed in Supplementary Table 7, together with the description of the cluster names and references to the marker gene information. The major cell types were annotated based on well-known marker genes reported in previous studies (Habib et al., 2016; Krienen et al., 2019; Lein et al., 2007; Luo et al., 2017; Tasic et al., 2018; Yao et al., 2020; Zeisel et al., 2018). Whenever possible, we name these clusters with canonical names (e.g., IT-L23, L6b) or using descriptive names that reflect the specific spatial location of the cluster (e.g., EP, CLA, IG-CA2). For subtypes, we named the clusters via its parent major type name followed by a short description or just a subtype marker gene name.

The above information has been added to the Methods. A description of the new web application for browsing these data has been added to the main text and the data availability section of the manuscript:

“A web application (Brain Cell Methylation Viewer) is provided to aid in visualization of cell- and cluster-level methylation data (<http://neomorph.salk.edu/omb>), as well as to access annotations and descriptions of marker genes for each cluster.”

We will continue to improve this tool by adding more additional information as it becomes available along with detailed documentation.

4. Please provide QC comparison on sn-m3C-seq, sn-mC-seq2, sc-ATAC-seq, in terms of how many reads/cell, genes/cell, Pearson correlation among individual samples as a supplementary table.

R1.5

We thank the reviewer for this suggestion. We now provide QC metrics for snmC-seq2 (Supplementary Table 5) and snm3C-seq (Supplementary Table 8) cells, including the number of reads/cell, the number of genes detected/cell and other essential mapping metrics for each technology, respectively. The data generation, mapping, and QC metrics for snATAC-seq data are detailed in the companion paper by Li et al.(Li et al., 2020)

For snmC-seq2, the final number of reads/cell is 1.54 ± 0.63 million (Mean \pm SD, Extended Data Fig. 2c). The number of genes detected/cell is $45,193 \pm 4,598$ (81% of total genes annotated in GENCODE vm22). Pearson correlations among individual samples were provided in Supplementary Table 4 and plotted in Extended Data Fig. 2d. The correlation between replicates ranged from 0.883 to 0.998.

For snm3C-seq, the final number of reads/cell is 1.56 ± 0.64 million, the number of genes detected/cell is $44,492 \pm 5,319$. snm3C-seq is based on snmC-seq2 protocol(Lee et al., 2019), therefore, the mapping metric is very similar. For the chromatin contact modality, the long-range cis contact number is 152 ± 61 k (both ends of the fragment mapped to the same chromosome with insertion size > 1 kb), the cis-trans ratio is 2.03 ± 0.42 (# cis contact / # trans contact).

For snATAC-seq, the preprocessing and quality control steps were described in Li et al.(Li et al., 2020) In brief, the number of fragments per cell is $4,930 \pm 4,109$. The number of genes detected/cell is $2,398 \pm 1,400$.

5. The authors need to do a better job to sequentially refer to figures in the text. For example, Fig. 2a and 2b were not mentioned before Fig. 2c.

R1.6

We thank the reviewer for pointing out the discrepancy in the sequence order of figures. We have revised the text and figures to ensure that the order of figure panels match their first mention in the text. The changes are:

1. In Fig.1, we swapped the four anatomical schematics panels with the UMAP and sunburst chart.
2. In Fig. 2, we added the description of Fig. 2a and b, and swapped Fig. 2j and k.
3. In Fig. 3, we rewrote the description to sequentially refer to each panel.

6. It will be useful to validate methylation detection for both mCG sites and mCH sites in a cell-type-specific context using a different approach.

R1.7

Since bisulfite sequencing is considered the gold standard for DNA methylation detection,

we used whole-genome bisulfite sequencing on purified cell types (Mo et al., 2015) to validate the cell type specificity of our single-cell sequencing and clustering approaches. UMAP embedding of methylomes from bulk purified excitatory neurons, PV and VIP neuron nuclei and snmC-Seq2 data showed excellent correspondence among all of the major types (Fig. R3a). Based on both mCH and mCG profiles, the strong correlations were observed between the bulk and single-cell samples (Fig. R3b). These results validated the detection of mCG and mCH at the major cell type level.

Fig. R3. Validation of cell-type-specific methylation pattern with bulk methylomes.
a, The UMAP embedding of snmC-Seq2 cells in our study (n=104,340) together with bulk methylomes from purified neural types (n=3) using both mCH and mCG. The major types relevant to the bulk cell types are colored and the other major types are greyed. **b**, The Pearson correlation coefficient between the major types and the bulk methylomes across 3,000 highly variable 100 kb bins using mCH (top) and mCG (bottom). The two replicates of bulk methylomes were shown separately.

7. When mapping methylome data using bismark with bowtie, did the author allow mismatching? How many mismatches were accepted? This is particularly important for mCH calling.

R1.8

We used default parameters of bismark (v0.20.0 and bowtie2 v2.3.4) that allowed mismatches (`--score_min L,0,-0.2`) in the final alignments but did not allow mismatch in seed alignments (`-N 0`). For final alignments, "`--score_min L,0,-0.2`" roughly corresponds to four mismatches per read, which is already more stringent than bowtie2's default setting (`L,-0.6,-0.6`). In addition, all sources of possible mismatches are well-controlled, as described below.

Bismark parameters

In bismark, the parameter "`--score_min L,0,-0.2`", i.e. $f(x) = 0 + -0.2 * x$ (where x is the read length) is for lowest alignment score. For a trimmed read of ~127bp, this would mean that a read can have the lowest alignment score of -25 before an alignment would become invalid, roughly four mismatches. Bismark uses a three-base mapping strategy to map the fully converted three-base reads onto similarly converted three-base genomes (Krueger and Andrews, 2011). Therefore, only reads with bisulfite converted mutations (C to T in the forward strand or G to A in the reverse strand) will be fully aligned without mismatches in the bismark mapping step, regardless of how many bisulfite-converted mutations it has. All distinguishable mismatches are non-bisulfite mismatches.

Source of non-bisulfite mismatches are well-controlled

The non-bisulfite mismatches that might impact alignment or methylation calling of the reads come from 1) SNPs, 2) technical sequencing errors (such as SNP/Indel caused by random primer and adaptase, or low-quality bases), and 3) bisulfite conversion-related non-bisulfite mismatches (mismatch positions which have a C in the BS-read but a T in the genome, which may occur due to the way bisulfite read alignments are performed, see the documentation of bismark https://rawgit.com/FelixKrueger/Bismark/master/Docs/Bismark_User_Guide.html#iv-running-bismark-methylation-extractor).

SNPs impact methylation calling in heterozygous organisms (e.g., humans), but our study used inbred mice. Therefore, The source of variance is well-controlled.

To minimize technical sequencing errors in the snmC-seq2 reads, we trimmed the reads before mapping, as described in Luo et al.(Luo et al., 2018) to remove any low-quality bases ($q > 20$). After trimming, $66.2\% \pm 3.0\%$ of the reads were uniquely mapped by bismark which is comparable to previous PBAT based bulk (66.3%)(Miura et al., 2012) or single cell ($64.7\% \pm 2.6\%$)(Luo et al., 2018) mouse WGBS-seq. The high mapping rate indicates the good quality of the trimmed reads.

For the last case, the bisulfite conversion-related non-bisulfite mismatches only account for $0.07\% - 0.09\%$ of the total mismatches (counted from 50 high-quality random cells). This small portion indicated that most of the alignments were selected from the correct alignment combination by bismark. Note that this type of mismatch was also discarded in methylation calling.

Non-bisulfite mismatches not likely confounding mCH/mCG calling

Finally, mismatches happening in the base following a cytosine might impact methylation context thus causing error in mC calling: changed from A, C, T to G (CH to CG) or from G to A, C, T (CG to CH). We counted such changes in 50 high-quality random cells and found such changes are infrequent and neglectable: $0.17\% - 0.21\%$ (CH to CG / all CG sites) and $0.01\% - 0.02\%$ (CG to CH / all CH sites).

8. For the ANP anp-dg cluster, are they most similar to DG granule neurons?

R1.9

This cluster is mainly dissected from DG and is very similar to DG granule cells (see below). In addition, the ANP anp-dg were labeled as non-neuronal cells due to a low global mCH / CH fraction and their large proportion (41% of the cells) in the NeuN- gate during FANS (Fig. R1 in R1.4).

In the level 1 UMAP embedding (Fig. R4a), we noticed that the “ANP anp-dg” cells have a continuous link to the DG granule neurons while the other ANP subtype “ANP anp-olf-cnu” links to the olfactory inhibitory neurons (OLF). Both DG and OLF are known to be related to adult neurogenesis (Ming and Song, 2011). The UMAP embedding and the partial NeuN- identity of these ANP cells indicate these cells may be the adult neuron precursor populations for DG and OLF, respectively.

The spatial location of these ANP cells further supports their precursor identity. The “ANP anp-dg” cells mostly come from DG dissection (Fig. R4b, c), corresponding to the subgranular zone (SGZ) within the dentate gyrus of the hippocampus where

neurogenesis happens (Ming and Song, 2011). The “ANP anp-olf-cnu” cells mainly come from olfactory and striatum dissection regions that overlap with the Rostral Migratory Stream (RMS), a tube-like structure that starts from the subventricular zone (SVZ) of the lateral ventricles and ends at the olfactory bulb (Fig. R4d).

Fig. R4. Spatial distribution of the ANP cells. **a, b**, L1-UMAP embedding of related cell types (n=7878) colored by subtype (**a**) or sub-region (**b**). **c**, bar plots showing dissection region composition of the two ANP subtypes. **d**, adult mouse brain schematic adapted from (Ming and Song, 2011).

9. Can the authors provide an example of the functional validation of their predicted enhancer-gene pairs?

R1.10

We appreciate the reviewer's comment but providing functional validation of enhancer-gene interactions in the context of our cell-type atlas is beyond the scope of our study. Recent functional enhancer studies in mice limb development (Osterwalder et al., 2018)

and the human neural stem cell line (Geller et al., 2019) both indicated that enhancer redundancy is common in mammalian genomes. Multiple enhancers interacting with the same gene confer phenotypic robustness to deletion or perturbation of individual enhancers (Geller et al., 2019; Osterwalder et al., 2018). As a result, knock-out (or inhibition through CRISPRi) of single specific enhancers may not be sufficient to reveal their true target genes. This feature reveals the complexity of enhancer-gene regulation; therefore, providing *in vivo* functional validation of enhancer-gene pairs would need substantial new experiments that we believe are beyond this study's scope.

10. The high overlapping of DMRs and open chromatin regions in the hippocampus confirmed the idea that the open chromatin regions are mostly cis-regulatory regions, but not necessarily confirmed the correct match of cell-type identities. The author should modify their statement or give more explanation.

R1.11

We agree with the reviewer that only showing the overlap between DMR and peaks in specific clusters is not sufficient to support the correct match of clusters. Therefore, we further plotted the mCG fraction and chromatin accessibility level of each set of DMRs across all hippocampus subtypes (Fig. R5). The same set of regions show similar cell-type-specificity in both modalities, which together with the overlap score (Fig. 2o) indicates the correct matching of cell-type identities.

We added Fig. R5 as Extended Data Figure 5c and revised the text of the results related to Fig. 2p as follow:

“We then performed co-clustering of the integrated data and calculated Overlap Scores (OS, see Methods) between the original methylation subtypes (m-types) and the chromatin accessibility subtypes (a-types, from Li et al.), which further quantified the matching of subtypes between the two modalities (Fig. 2o, Extended Data Fig. 5b, Supplementary Table 7). In addition, the CG-DMRs highly overlap with open chromatin peaks in corresponding subtypes of the hippocampus (Fig. 2p). Their mCG fractions and chromatin accessibility levels show similar cell-type-specificity across hippocampus subtypes, confirming the correct match of cell-type identities (Extended Data Fig. 5c).”

Fig. R5. mCG fraction (left) and chromatin accessibility (right) of cluster-specific CG-DMRs (columns) in hippocampus subtypes (rows).

4. The analysis on key transcriptional factor identification in specific cell types is interesting. Considering the low expression level of TFs in the single-cell RNA-seq dataset, the analysis in this study provides a unique angle on cell-type specification. It will be interesting to perform an integrated analysis on those candidate TFs' ChIP-seq datasets.

R1.12

We greatly appreciate the reviewers' acknowledgment of this unique aspect of our study. However, Existing ChIP-seq datasets generated from different tissues or developmental stages do not overlap with the cell type class and brain regions. We found only one example ((RARB ChIP-seq ([GSM1656748](https://www.ncbi.nlm.nih.gov/geo/query/acc.cgi?acc=GSM1656748)))) in adult mouse brain ventral striatum (Niewiadomska-Cimicka et al., 2017) that fits with data in our study. Integration of the RARB dataset reveals that distal peaks identified in bulk ChIP-seq are in excellent agreement with our cell-type-specific hypo-mCG and gene body hypo-mCH. Specifically, the *Rarb* gene is highly expressed in the mouse striatum and is an important regulator of normal striatal functions (Zetterström et al., 1999). As expected, the gene body of *Rarb* shows specific hypo-mCH in striatal major neuronal cell types, including "MSN-D1", "MSN-D2", "D1L-Fstl4", "D1L-PAL", "Foxp2" etc. (Fig. R6a, b, and in the web application here: <http://neomorph.salk.edu/omb/gene?gene=Rarb>). We overlapped peaks identified from the ChIP-seq with eDMRs identified in our study. Consistent with its gene body methylation pattern, the binding peaks of the RARB protein are specifically enriched in the hypo-eDMRs of corresponding cell types (one-sided Fisher's exact test with Benjamini-Hochberg correction, see FDRs in Fig. R6c and ChIP-seq peaks' mCG / CG in

Fig. R6d).

Fig. R6. RARB ChIP-seq integration. a, b, subtype level normalized gene body mCG

(a) and mCH (b) fraction of RARB gene. **c**, $-\log_{10}(\text{FDR})$ of the RARB ChIP-seq peak enrichment in each cell-type-specific hypo-DMR region. **d**, heatmap of ChIP-seq peaks' mCG / CG at 161 subtypes.

11. Is the role of mCG and mCH in cell-type specification different or similar? Are there any genomic regions of DMRs, like promoters, enhancers, TSSs or gene bodies, play predominant roles?

R1.13

We interpret the “role” of mCH and mCG as the level of the cell-type-specific information contained in each methylation type in a specific set of genomic regions. To quantify this level, we used the ratio of between-cluster variance versus within-cluster variance for different methylation types in each region set. Based on the new analysis described below, we found that the mCG has higher cell type-specificity than mCH in distal enhancer-like regions. In contrast, the gene-body mCH in neurons is more cell-type-specific than gene-body mCG. Neither methylation type shows strong cell-type-specificity in promoter and CGI regions. To study the “mechanistic role” of mCG and mCH in cell-type specification would require developmental and functional experimental data, which we believe is beyond the scope of this study.

Region sets definition

First, we define the region sets as follow: 1) “Gene”: all protein-coding gene body region defined in GENCODE vm22; 2) “CGI”: mm10 CpG island regions downloaded from UCSC table browser on July 1st, 2020; 3) “Promoter”: TSS \pm 2 kb of each protein-coding gene transcripts defined in GENCODE vm22; 4) “enhancer (eDMR)”: we selected all eDMRs from subtypes related to the dataset we used (see below) and removed any regions that overlapped with “Promoter” or “CGI”. 5) “enhancer (peak)”: since the eDMR regions were identified based on CpG differential methylation analysis, which might be biased in comparing different methylation types, here we also added the ATAC peaks from Li et al. (Li et al., 2020) as another enhancer-like region set. The peaks were identified using snATAC-seq data only and removed any region that overlapped with “Promoter” or “CGI.” We calculated the methylation fractions in CH and CG context of all these regions in each cell from two replicates of the MOP-2 dissection as examples (each replicate has ~1,200 nuclei passing QC).

Comparing mCH and mCG

We then used these methylation fractions to calculate ratios of between/within-cluster variance. For each region set, the between-cluster variance var_b is calculated as the variance of the mean of each cluster; the within-cluster variance var_w is calculated as the variance across all cells, after subtracting the cluster mean. A higher ration (var_b/var_w) means the methylation fraction of that region contains larger cluster-specific variance. We then calculate the Cumulative Distribution Function (CDF) of the ratios for each set of genomes regions for cross-comparison (Fig. R7).

In promoter and CGI regions (Fig. R7a, b), we found that both mCH and mCG fractions have smaller ratios than other region sets. This low level of cell-type-specificity agrees with the previous report that these regions are hypo-methylated consistently in adult tissues and do not preserve much cell type or tissue-specific methylation patterns (Roadmap Epigenomics Consortium et al., 2015; Schultz et al., 2015).

In contrast, distal enhancer-like elements defined by both eDMR and ATAC peaks (Fig. R7c, d) show substantially higher ratios, indicating stronger cell type-specific methylation patterns in enhancers compared to promoters and CGIs. Also, in these enhancer-like regions, mCG has higher cell type-specificity than mCH (Fig. R7c, d).

In gene bodies, the distribution of mCH and mCG is different in neurons and non-neurons (Fig. R7e, f). The ratio calculated with neurons indicates that mCH consists of higher variance across cell types than mCG. Such differences become minor when calculating the ratio using non-neurons, whose mCH level is also much lower than neurons.

Fig. R7. Between-cluster variance vs. within-cluster variance ratio in different sets of genome

regions. Please see the illustration on the left.

12. For sn-m3C-seq analysis, were nuclei sorted by NeuN?

R1.14

We did not perform NeuN sorting for the snm3C-seq experiment; all cell-types were profiled in these experiments. We also added this sentence in the method section:

“Nuclei were then stained with Hoechst 33342 (but not stained with NeuN antibody) and filtered through a 0.2 μ M filter and sorted similarly to the snmC-seq2 samples.”

13. Several genes show a gradient of expression along the dorsal-ventral (Septo-temporal) axis in the adult mouse gyrus, for example, *sfrp3* (PMID: 26337530). How does the methylation status of these genes look like in UMAP (Fig. 5g)?

R1.15

We plotted the gene body mCH and mCG level of several ventral genes (e.g., *Sfrp3/Frzb*) and dorsal genes (e.g., *Abcb10*) (Zhang et al., 2018) on our DG UMAP, and no methylation gradient patterns were observed (Fig. R8a). We also looked at these genes in Zhang et al., where both bulk RNA-Seq and bisulfite sequencing were performed

in manually dissected dorsal and ventral DG. Both *Frzb* and *Abcb10* are dorsal-ventral DEGs, but no corresponding dorsal-ventral DMRs can be identified surrounding their gene bodies (Fig. R8b). These data are consistent with our findings, indicating that the differential expression of *Frzb* or *Abcb10* is not associated with changes of DNA methylation.

Given that the brains were dissected into 600-micron coronal slices in our study, we do not have a gold-standard label of the dorsal-ventral axis of DG. Nevertheless, to more thoroughly explore whether the DG methylation gradient corresponds to known dorsal-ventral methylation differences, we compared our data with the published dorsal/ventral methylome (Zhang et al., 2018). As described in the previous version of our manuscript, we divided the granule cells into four groups based on their global mCH, and identified DMRs between the groups whose mCG levels were positively correlated with global mCH (+DMRs) or negatively correlated with global mCH (-DMRs). We found that 15% of the dorsal-hypo DMRs from Zhang et al. (Zhang et al., 2018) overlapped with +DMRs ($P < 0.001$, ave. shuffle overlap $3.0 \pm 0.1\%$), and 4% of the dorsal-hypo DMR overlapped with -DMRs ($P < 0.001$, ave. shuffle overlap $1.7 \pm 0.1\%$) (Fig. R8f). Similarly, 13% of the ventral-hypo DMRs overlap with -DMRs ($P < 0.001$, ave. shuffle overlap $0.06 \pm 0.02\%$) (Fig. R8f). Together, these moderate but significant overlaps imply that the axis of global mCH gradient is partially explained by the traditionally defined dorsal-ventral axis. It would require corresponding dissections, or spatially-resolved single-cell epigenetic data to comprehensively evaluate the methylation differences along the dorsal-ventral axis of DG granule cells.

Fig. R8. Supporting information for the dorsal-ventral DEGs. a, L3-UMAP colored by mCH and mCG fraction of *Frzb* (first column) and *Abcb10* (second column). **b**, genome browser from Zhang et al (https://brainome.ucsd.edu/annoj/mm_dentate_ee_sh/browser.html), showing mCH, mCG and RNA expression level surrounding *Frzb* and *Abcb10*. **c**, dorsal-ventral DMRs identified from Zhang et al. overlapping with +/- DMRs (f) in our DG cells.

Finally, we examined what the implications of the global mCH gradient could be. Our hypothesis is the global mCH difference may be related to the maturation of granule cells. Previous studies have also demonstrated the accumulation of global mCH during the development of different brain structures (He et al., 2020; Lister et al., 2013).

To further explore whether mCH gradients are related to granule cell maturation, we selected genes that are enriched for +DMRs in their gene bodies (+DMRgenes) or enriched for -DMRs (-DMRgenes), and then examined the expression of these genes across development time points. We used a scRNA-seq dataset (Dataset C in Hochgerner et al.) that contains 24,185 DG cells from eight developmental time points (E16.5-P132). Eight cell types were selected from the original study, including radial glia-like cells (RGL), neuronal intermediate progenitor cells (nIPC), neuroblast, immature, and

mature granule cells, which cover the whole developmental trajectory of granule cells. Intriguingly, the +DMRgenes and -DMRgenes have substantially different dynamic expression patterns across the eight cell types along their developmental trajectories. The +DMRgenes have higher expression in the immature cell types than mature cell types, while the -DMRgenes show the reverse trend (Fig. R9a-c). This finding indicates the +DMRgenes and -DMRgenes may be repressed and activated during granular cell maturation, respectively. Two example genes *Rfx3* and *Tcf4* are shown in more detail to better illustrate the consistent trend between methylation gradient and developing RNA expression profile (Fig. R9d, e).

In summary, these results are consistent with our hypothesis that young DG granule cells have low global mCH, and low methylation at genes associated with neural precursors. Older DG granule cells have accumulated greater global mCH, and have low methylation at genes associated with mature neurons.

Fig. R9 +/- genes are developmentally associated. **a**, violin plot summarizing cluster mean CPM of +/- genes in the eight clusters identified in Hochgerner et al. **b**, **c**, cluster mean-centered CPM of individual genes in eight clusters. Clusters are ordered based on their position in developmental trajectory from left to right, i.e., from immature to mature. **d**, **e**, Two examples from +gene *Tcf4* (left) and -gene *Rfx3* (right). From top to bottom showing 1) gene model; 2) two DMR groups; 3) mCG fraction (range 0-1) of the DG cell groups and the “ANP dg-all” subtype; 4) mCH fraction (range 0-0.03) of the DG cell groups and the “ANP dg-all” subtype; 5) normalized gene body mCG fraction of granule cells (n=6,179); 6) RNA expression level of the developing DG scRNA-seq dataset, image downloaded from <http://linnarssonlab.org/dentate/>

4. Are there sufficient cells to train the artificial neural network to predict nonneuronal cells? Do nonneuronal cells exhibit strong region specificity?

R1.16

We thank the reviewer for this suggestion. After further examination of the non-neuronal cell data, we indeed have evidence that non-neuronal cell types also exhibit regional specificity. By incorporating non-neuronal cells into the ANN model, we observed an average accuracy of 95% to predict cell types, and 42% to predict their anatomic location. The accuracy to predict region specificity is highly dependent on the cell type. Specifically, we noted stronger predictability in astrocytes (ASC, 62%) and oligodendrocytes (ODC, 52%) that are comparable to cortical inhibitory neurons, while weaker predictability comparable to random in the other non-neuronal types (OPC, 13%; MGC, 2%; Others, 17%). The sample sizes of oligodendrocyte progenitor cells and microglia are comparable with the subtypes of ASC and ODC. Thus, the differences of performances are less likely due to insufficient cell numbers. It is worth noting that unsupervised clustering did not separate non-neuronal cells from different regions into different clusters, which further emphasizes the utility of supervised analyses to study the regional specificity of these cell types.

These results are further supported by previously published literature. It has been reported that astrocytes are derived from regionally patterned radial glia (Bayraktar et al., 2014), while the signature of oligodendrocytes depends on the local environment rather than their progenitor cells (Floriddia et al., 2019). Indeed, we observed a high performance of region prediction for ODC, but baseline level performance in OPC. These results demonstrated that ASC and ODC show moderate regional specificity, while OPC and other non-neuronal cells do not. We have updated the main text and corresponding figure in Extended Data Fig. 11 to include this analysis.

“We also observed moderate spatial specificity of non-neuronal cells. With the same network structure and features, we achieved 42% accuracy to predict the cell body location of non-neuronal cells. The accuracy depends highly on the cell types, with stronger predictability in astrocytes (ASC, 62%) and oligodendrocytes (ODC, 52%) that are comparable to cortical inhibitory neurons (Extended Data Fig. 11d). This is further supported by the evidence that astrocytes are derived from regionally patterned radial glia (Bayraktar et al., 2014), while the signature of oligodendrocytes depends on the local environment (Floriddia et al., 2019). It is worth noting that unsupervised clustering did not separate non-neuronal cells from different regions into different clusters, which further emphasizes the utility of supervised analyses to study the regional specificity of these cell types.”

Fig. R10. The dissection region prediction accuracy of nonneuronal cells. The colored points denote the prediction accuracy of the model, while the grey ones denote the random guess accuracy when cell subtypes and corresponding spatial distributions are given.

Referee #2 (Remarks to the Author):

Summary

Liu and Zhou et al. present a large-scale epigenetic landscape of the mouse brain in single cells. Specifically, more than 100,000 nuclei from 45 annotated regions within the cortex, hippocampus, striatum, palladium, and olfactory spaces. This manuscript in conjunction with associated companion manuscripts, take multiple "views" of the same tissue using previously published epigenetic and structural 'omics approaches at varying resolutions to provide a more comprehensive analysis of the data. Overall, the data and analysis are well carried out and address a critical need in the field. As evidenced by the multiple companion manuscripts, the sheer scale of the data and necessary analyses are somewhat prohibitive in a single paper to move to more granular understanding and validation of predicted subtypes. Nevertheless, this paper represents a critical resource that will be generally of interest to the field. Further, the authors should be commended for making the data and code available for such a valuable resource, although the sn-

m3C-seq data seem to be missing from the NeMO website; I was only able to find the snmC-seq data. There is also a visualization website that presents the data in a more accessible form. However, with this volume and high dimensionality of the data, it is clear that they faced similar challenges of many such large-scale papers, where it is often hard to get into the granularity, and cover enough details.

R2.1

We thank the reviewer for appreciating the significance of the study and the critical need for these data in the field, and for providing very helpful comments to improve our study. To address the reviewer's comments raised below, we performed analyses showing the robustness of our clustering results as well as justifying the preprocessing steps. We also compare our snmC-Seq data with scRNA-Seq data to validate the corresponding cell types in the two modalities and the gene-enhancer pairs linked by the correlation analyses. Below we provide responses to each of the comments.

For the data sharing, we have uploaded the missing snm3C-seq data to GEO (GSE132489, the Reviewer Access Token is oxcbgqsazienxcl). To better share our data and analysis with the general audience, we now provide [an interactive web application \(http://neomorph.salk.edu/omb/home, please see R2.4\)](http://neomorph.salk.edu/omb/home) for cell-level and cluster-level data visualization. We have rewritten the data availability section of the manuscript, and now include the above data links.

Main comments:

1) Binning: Lines 112 to 114 discuss the level of methylation coverage represented across the mouse genome in 100-kb bins being on average 95% with gene bodies at 81% on average. This is highly confusing and somewhat misleading as currently written since the data were summarized into 100-kb bins and the discrepancy between genome-wide and gene body percentages is unclear.

R2.2.1

We thank the reviewer for pointing this out. There are, in fact, two metrics that we used to measure coverage. The first is the coverage of single cytosine sites (6%), which represents the proportion of cytosines in the genome that has been covered by at least one read. The second is the coverage of larger regions (95% of 100 kb bins and 81% of genes), which represents the proportion of regions that the methylation fraction can be reliably computed. Based on the estimation of a binomial model, the reliability of methylation level quantification depends on the length of the region ((Luo et al., 2017), Fig. S1). Specifically, a larger region is covered by more sites and reads, which reduces the uncertainty of methylation quantification. Since a large number of genes are shorter

than 100 kb, the average percentage for accurate quantification is higher for 100 kb bins than gene bodies. Given that most of our downstream analyses were applied to the bins and genes, we believe that both metrics are important and are explicitly described in the text. We have made the following edits to this section to make this more clear.

“In total, we profiled the DNA methylomes of 110,294 single nuclei yielding, on average, 1.5 million stringently filtered reads/cell (mean \pm SD: $1.5 \times 10^6 \pm 5.8 \times 10^5$), covering $6.2 \pm 2.6\%$ of the cytosines in the mouse genome in each cell. These parameters allowed reliable quantification of the DNA methylation fraction for 25905 ± 1090 ($95 \pm 4\%$) 100 kb bins and 44944 ± 4438 ($81 \pm 8\%$) gene bodies (Fig. 1i). The global non-CG methylation levels range from 0.2% to 7.6%, and global CG methylation levels range from 61.6% to 88.8% (Extended Data Fig. 2b, c).”

Each cell received about 1.5M reads (after filtering), or ~6% of the genome. Using 100-kb bins helps reduce the impact of such sparsity in the data, but at the 100-kb size range, different information is captured than at more granular levels. Long-range effects such as PMD hypomethylation will tend to dominate, while smaller features such as CpG islands, promoters, and enhancers will be lost.

R2.2.2

As the reviewer suggested here and below (R2.2.5), we evaluated the clustering ability of mCH and mCG in eight different bin sizes (from 5 kb to 1 Mb) on the MOp-2 dataset (2386 nuclei from two replicate), and observed robust clustering results across these parameters.

For each bin size, we performed the same preprocessing steps as described in the methods. We used different cutoffs corresponding to each bin size (e.g., coverage cutoff is linearly scaled up or down for longer or shorter bins, respectively). All filtering steps aim to keep a similar percentage of features as the original 100 kb bin analysis. After preprocessing, we then performed PCA (Fig. R11a, b), UMAP embedding on PCA space (Fig. R11c), UMAP embedding without PCA (see discussion in R2.6.3), and clustering (Fig. R11d) at each bin size.

For both mCH and mCG, the trend of explained variance ratio for top PCs and the shape of PC1 & PC2 embeddings are consistent among most binning sizes (Fig. R11a, b). While the UMAP embeddings are altered by different binning sizes, the overall cell type structures are still preserved (Fig. R11c). We further checked the consistency of clustering for different methylation types and bin sizes combinations by calculating the adjusted rand score (ARS) (Hubert and Arabie, 1985) between the clustering result of each combination and the original subtype labels. Again, both mCH and mCG have consistent ARS across bin sizes. In summary, the bin-size does not have a notable impact on the clustering results of the snmC-seq2 data and we chose the 100 kb bin-size to be consistent with prior analysis (Luo et al., 2017). Although we

agree with the reviewer, that smaller features might better capture enhancer level information, the results here indicate that methylation features at different sizes contain redundant information to determine clusters. The benefit of using 100 kb bins is that it's usually an order of magnitude faster than smaller bins such as 5 kb, which allows us to explore more hyperparameters given the same amount of computation resources.

Fig. R11. Comparing the bin-size effect on embedding and clustering results. **a**, explained variance ratio of top PCs calculated from mCH and mCG matrix. **b**, **c**, scatter plot of PC1-PC2 and UMAP embeddings (n=2386 for all plots). Each group is clustered separately, but here all scatter plots are colored using original subtype labels to allow easy visual comprehension. **d**, ARS comparing the clusters identified by each feature set with original subtype labels.

Despite the binning, heavy imputation was needed nonetheless (and performed; Line 742 onwards). It seems the global mean of the cell was used as the prior. The rationale for this choice is not clear. If anything, it should be the mean per locus across the cells. And why impute features with less cov (coverage?, not covariance or coefficient of variation, correct?) to have a mean methylation level value close to 1?

R2.2.3

The use of the beta-binomial distribution to estimate the methylation level has been described previously for single-cell methylome data (Smallwood et al., 2014). The original study only focused on mCG and used a constant beta prior ($a=1$, $b=1$). Since the methylation levels of CH and CG are very distinct, we also tested the clustering and embedding performances using two more specific priors for each methylation type, which are 1) calculating the prior based on the mean and variance of each cell, 2) as the reviewer mentioned, calculating the prior based on the mean and variance of each feature/locus. Similar to R2.2.2, we evaluated the impact of three different priors, using the MOp-2 dataset and 100 kb bins as features.

We found that the constant beta prior ($a=1$, $b=1$) does not work for the CH methylation, and also gives worse performance for the CG methylation (Fig. R12a, b). Calculating priors by cell mean or by feature mean gives very similar clustering results which are considerably better than the constant beta prior (Fig. R12a, b). From an implementation aspect, when doing such calculations for the full dataset, we need to use a chunk-by-chunk implementation as the total amount of the data is very large and the requirement for physical memory is large. Because the data is natively stored by cell chunks (as the data is generated continuously along the time), calculating mean per cell is practically easier than calculating mean per feature. As the comparison here indicates, the two calculations performed equally well, and thus we chose cell-level normalization which is more efficient to implement.

For the last question about normalization to 1, the purpose of this step is to normalize the bin-level methylation fractions in a cell by its global methylation level. This was applied to not only the low coverage features but all features. We have modified the methods section to make this more clear. This approach is similar to the use of CPM normalization in the single-cell RNA-seq analysis (Luecken and Theis, 2019), to prevent

PC1 from being dominated by the global methylation level. We also compared the embedding and clustering with or without normalization (Fig. R12c, d). Similar clustering results were observed (Fig. R12d), but the UMAP embeddings without normalization are more stretched (Fig. R12c). In some cases, such an effect makes the embedding less visually clear to illustrate the cluster structure.

Fig. R12 Comparing the effect of priors and normalization on UMAP embedding and clustering. **a, b**, UMAP embedding using different methylation types (columns), different prior calculation methods (rows), and whether normalize the fraction per cell (a is normalized, b is no normalization). Each group is clustered separately, but here all scatter plots are colored using original subtype labels to allow easy visual comprehension. **c, d**, ARS comparing clusters identified by each condition with original subtype labels.

It would be helpful to show the number of CpG or CpH sites covered by at least one read, since tiling the genome at a single resolution introduces a bin-size bias in the representation of the data.

R2.2.4

As described in R2.2.1, the original text “covering $6.2 \pm 2.6\%$ of the mouse genome per cell” in the manuscript corresponds to the number of cytosines covered by at least one read. We used the total number of cytosine covered by at least one read divided by total cytosine in the mappable genome to get this percentage. We have rephrased the text and methods to make this more clear.

“In total, we profiled the DNA methylomes of 110,294 single nuclei yielding, on average, 1.5 million stringently filtered reads/cell (mean \pm SD: $1.5 \times 10^6 \pm 5.8 \times 10^5$), covering $6.2 \pm 2.6\%$ of the **cytosines in the mouse genome in each cell.**”

Many previous single-cell methylome studies (Luo et al., 2017; Mulqueen et al., 2018; Smallwood et al., 2014) used tiling to reduce feature number and data sparsity before embedding and clustering since the total number of cytosine detected is too large (10^9) to be used as individual features. Please see our response to R2.2.2 showing that bin-size does not have a large impact on clustering.

There may be added value to varying the bin sizes and combining them for a more multi-scale view of the data, as opposed to PMD-level binning, given the size of the data. That being said, going back and reanalyzing for TF motifs and DMRs/DMGs was great and does address this concern to some extent.

R2.2.5

We thank the reviewer’s recognition of the post-clustering analysis of gene and regulatory elements. Please see our reply to question R2.2.2, where we discuss our analysis of different bin-sizes of features (from 5 kb to 1 Mb). Since bin-size does not have a notable impact on the clustering results of the snmC-seq2 data (Fig. R11), we chose 100 kb as a suitable size for large-scale clustering analysis.

2) Filtering: The filtering choices are not always clear. For example, 100-kb genomic bin features were filtered by removing bins with mean total cytosine base calls < 250 or > 3000 . It makes sense to filter the sparse bins, but why filter bins with high coverage? Is this an attempt to remove CpG islands?

R2.3.1

We do not attempt to remove the CpG islands here. This step is designed to remove regions that have “mapping issues”. The minimum and maximum cutoff for total cytosine base calls are selected based on the distribution of total cytosine base calls of all 100 kb bins. As shown in Fig. R13, the minimum cutoff cov. < 250 is intended to remove regions

that are poorly covered or unmappable ($n=977$, 3.7% of total 100 kb bins). The maximum cutoff (cov. > 3000) is intended to remove some regions showing extremely high coverage ($n=6$, 0.023% of total). Five out of the six extremely high-coverage regions are also overlapped with “High Signal Regions” in the ENCODE mm10 blacklist v2 (Amemiya et al., 2019), which means these regions are known to have mapping issues and not recommended to be included in the analysis. To clarify this step as the reviewer mentioned, we have added a better explanation of our filtering strategy in the Methods.

“Feature filtering. 100 kb genomic bin features were filtered by removing bins with mean total cytosine base calls < 250 (low coverage) or > 3000 (unusually high-coverage regions). Regions that overlap with the ENCODE blacklist (Amemiya et al., 2019) were also excluded from further analysis.”

Fig. R13. Mean total cytosine base calls of all 100-kb non-overlapping genome bins.

Samples were filtered based on a mCCC rate of 3%. It is not clear that this is good enough for studies that focus on mCH methylation, where the average level falls well below 3%.

R2.3.2

The maximum cutoff of mCCC fraction was chosen based on the overall mCCC distribution, as shown in Fig. R14. Because this step was applied before any clustering analysis, we decided to use a loose cut off to prevent removing any population that might

be biologically meaningful.

Fig. R14. Distribution of mCCC fraction across single cells (n=110,294).

We further evaluated the impact of decreasing the mCCC cutoff down to 1%. Among the 1636 cells with mCCC fraction falling between 0.01 to 0.03, 253 of them passed all QC filters and were assigned to clusters, the other cells were filtered out due to other criteria. These 253 cells (“high mCCC”) come from 70 different subtypes, while most of them only contribute a tiny portion of the total subtype populations. Notably, a subset of them with 47 cells were assigned to the “PAL-Inh Chat” subtype (cell type browser http://neomorph.salk.edu/omb/cell_type?ct=PAL-Inh%20Chat), which contributes to 24% of the total “PAL-Inh Chat” population. This subtype indeed also has the highest mean mCH fraction (0.039 ± 0.008) among all subtypes we defined and is clearly marked by the choline acetyltransferase gene *Chat* (Fig. R15a), corresponding to cholinergic neurons in the striatum and pallidum. Other cells from the same experiments that contain these “high mCCC” cells do not show abnormal mCCC levels in general (For example, two replicates of CP-1 dissection region, Fig. R15b).

In addition, we also had unmethylated lambda DNA spike-in in all of our experiments as non-conversion control. In these “high mCCC” cells, the average lambda DNA mC fraction is low (0.0087), further confirming the high mCCC is not due to non-conversion.

Note that not every cell has enough lambda DNA reads to calculate the non-conversion rate, so we used the mCCC fraction as a proxy at the single-cell level (see discussion in R2.12.3).

Based on this evidence, we believe that the 3% mCCC cutoff is appropriate for initial pre-clustering filtering. We respectfully disagree with the reviewer that “the average level falls well below 3%” (and apologies for not being more clear). As shown in Fig.R15c below, the mean mCH fraction of many cell types is close to or even above 0.03. Indeed, the reviewer is correct that the average mCH fraction is below 3% based on the previous reports using bulk methylome data (He et al., 2020; Lister et al., 2013). However, our more extensive survey of cell types in this new dataset provides an update to this initial observation. We find many neuronal cell types can have a global mCH fraction as high as 0.03. The lower global mCH levels detected in bulk samples represent averages between neurons having high global mCH levels and non-neuronal cells with <1% mCH. We have added the global mCH range of all the cells in the main text.

“The global non-CG methylation levels range from 0.2% to 7.6%, and global CG methylation levels range from 61.6% to 88.8% (Extended Data Fig. 2b, c).”

Fig. R15. Supporting information for mCCC cutoff. a, *Chat* normalized gene body mCH in PAL-Inh cells, L3-UMAP (n=4307). **b**, mCCC distribution of two replicates of CP-1. “PAL-Inh Chat” cells shown in red. **c**, cell global mCH fraction on L1-UMAP (n=103,982)

In addition, another filter of 20% mCH was applied. It is curious that after the 3% mCCC filtration there would still be cells with >20% mCH methylation (how often?). This would actually indicate that there's true CpH methylation in those cases?

R2.3.3

We thank the reviewer for pointing out these two redundant cutoffs. We only found 2 cells having >20% mCH fraction but <3% mCCC fraction. These cells have only 8 reads and

1012 reads, which didn't pass the final reads > 500,000 cutoff. These "cells" might correspond to empty wells or debris, where the extreme shallow coverage made it meaningless to evaluate its whole genome methylation level. By combining both the methylation level and final reads filters, these abnormal cells were excluded.

3) Neuron Classification: How are the excitatory and inhibitory neurons classified in Figure 1b and 1c? These are ad hoc drawn clusters so presumably, a priori information was used to delineate these cell types. However, this is not described well in the manuscript. This is critical given it is the highest level in the hierarchical clustering performed that underlies the main conclusions of the paper.

R2.4

For the cell class annotation, we first did the unsupervised clustering (see methods) and then annotated the cell classes using ad hoc information. Specifically, we annotated non-neuronal cell clusters based on both the NeuN- gate origin and low global mCH fraction (Fig. R16). Given the strong anti-correlation between CH methylation and gene expression, we used hypo-CH-methylation at gene bodies ± 2 kb of pan-excitatory markers such as *Slc17a7*, *Sv2b* and pan-inhibitory markers such as *Gad1*, *Gad2* (Fig. R17) to annotate excitatory and inhibitory cell classes, respectively.

All details about cell type annotation can be found in Supplementary Table 7. To help the general audience better access these detailed annotations, we have developed a web application (<http://neomorph.salk.edu/omb/home>) to interactively visualize the gene-level DNA methylation and metadata of single cells. Each cell type has a corresponding page, whose URL can also be found in the Supplementary Table 7.

Fig. R16. Non-neuronal cell class identification. **a, b,** L1-UMAP embedding of all cells pass QC (n=103,982), colored by NeuN+/- gate information during FANS (a) and cell global mCH fraction (b). **c,** scatter plot of major cell type mean NeuN-% (x-axis) and global

mCH fraction (y-axis), non-neuronal cell types have low global mCH and high NeuN-%.

Fig. R17. Pan-excitatory and pan-inhibitory marker genes. Colors showing z-score (per row) of major cell types gene body mean mCH fraction.

4) The authors should be commended for their robust exploration of clustering parameters by a grid search for hyperparameter (namely the resolution parameter) tuning - however, it remains unclear how the cluster labels for each level of iterative cluster were generated with regards to cell type/subtype.

R2.5

Below we describe the process of Major type (level 2) and subtypes (level 3) annotations. Similar to R2.4, the details about cell type annotation can be found in Supplementary Table 7, and the interactive web application for cell type and genes (<http://neomorph.salk.edu/omb/home>).

We used gene body \pm 2 kb hypo-CH-methylation (or hypo-CG-methylation for non-neurons) of well-known marker genes along with dissection information to annotate neuron and non-neuron clusters. All cluster marker genes are listed in Supplementary Table 7, together with the description of the cluster names and references to the marker gene information. The major cell types were annotated based on well-known marker genes reported in previous studies (Habib et al., 2016; Krienen et al., 2019; Lein et al., 2007; Luo et al., 2017; Tasic et al., 2018; Yao et al., 2020; Zeisel et al., 2018). Whenever possible, we name these clusters with canonical names (e.g., IT-L23, L6b) or using descriptive names that reflect the specific spatial location of the cluster (e.g., EP, CLA, IG-CA2). For subtypes, we named the clusters via its parent major type name followed by a sort description or just a subtype marker gene name.

The above information has been added to the Methods. A description of the new web application for browsing these data has been added to the main text and the data availability section of the manuscript:

“A web application (Brain Cell Methylation Viewer) is provided to aid in the visualization of cell- and cluster-level methylation data (<http://neomorph.salk.edu/omb>), as well as to access annotations and descriptions of marker genes for each cluster.”

We will continue to improve this tool by adding more additional information as it becomes available along with detailed documentation.

5) The initial data matrix construction of PCs for downstream clustering analysis was done using the elbow method, which is somewhat arbitrary. How many PCs were chosen? The plots of variance explained for the mc rates, along with an indication of the cutoff used, need to be shown as a supplemental figure for possible reproducibility.

R2.6.1

As suggested by the reviewer, we plotted PCs from the L1 (Fig. R18a, b), L2-Inh (Fig. R18c, d), and L3-MSN-D1 analysis (Fig. R18e, f) as three examples that paired with Fig. 1b. For the purpose of reproducibility, we have provided all of the parameters for each level of clustering analysis in Supplementary Table 6.

Fig. R18 Three example PCA plot. From Level 1 (a, b, from all cells), Level 2 (c, d, from all inhibitory cells), Level 3 (e, f, from MSN-D1 cells) analysis. **a, c, e**, showing explained variance ratio of top PCs of the mCH or mCG matrix, **b, d, f**, showing scatter plots using eight pairs of PCs, colored by corresponding cell type labels from original study to help visually evaluate the information content in each PC.

Additionally, splitting up the CG and CH methylation into separate matrices, performing PCA, and concatenating the matrices is unclear in the motivation to do so. Specifically, the correlation structure between the two types of methylation is destroyed despite having biological ties to one another. The correlation between the PCs from CG and CH also should be shown to understand the degree to which this correlation exists between them. Why not use NMF or ICA as an initialization using the full matrix (e.g. both CG and CH normalized bins)?

R2.6.2

As suggested by the reviewer, we now provide Pearson correlations between the PCs of the mCH matrix and the mCG matrix (Fig. R19a). The top PCs (which are most informative for clustering and embedding) from the two matrices are highly correlated, indicating that both methylation types capture similar dominant variances. This also supports the previous results described in our responses in R2.2 or R2.6.1 that indicate both methylation types are able to give very similar clustering results.

Since the PCs of the two matrices do not represent the same space, we also provided Pearson correlations between the mCH and mCG fractions on the original 100 kb bin feature spaces. We calculate the correlation between mCG and mCH using the raw 100 kb bin matrices before PCA, and also the reconstructed 100 kb bin matrices by the top 30 PCs of each methylation type. As shown in Fig. R19b, the mCH, and mCG fractions of most 100 kb bin features are positively correlated, and selecting top PCs indeed enhanced such correlation as it “denoised” the matrices by removing later PCs. These results demonstrate that the intrinsic correlation structure between the two types of methylation is not destroyed by performing PCA separately on the two matrices.

Fig. R19. Pearson correlation of mCH and mCG. **a**, Correlation between top 30 PCs of mCH (y-axis) and mCG (x-axis) matrix. **b**, Distribution of the feature mCH and mCG Pearson correlation of each cell. The Raw is the correlation calculated using input matrix; the Reconstruct is the correlation calculated using the top 30 PCs reconstructed matrix.

Next, we tested how different decomposition methods might impact clustering results. Using the 100 kb bin features of the MOp-2 dataset (the same as R2.2), we tested six decomposition strategies: 1) PCA on the mCH matrix only, 2) PCA on the mCG matrix only; 3) Concatenate PCs from 1 and 2; 4-6) Concatenate mCH and mCG matrix, then run PCA (4), ICA (5), NMF (6). For all three decomposition algorithms (PCA, ICA, NMF), we used the implementations from the “sklearn.decomposition” (scikit-learn 0.23.1) with default parameters and a different number of components from 3 to 30. As shown in Fig.

R20, all six strategies resulted in very similar UMAP embedding and clustering results, indicating the choice of decomposition method does not have a large impact on

clustering results.

Fig. R20. Testing different decomposition strategies. **a**, UMAP embedding using selected components in each group. Each group is clustered separately, but here all scatter plots are colored using original subtype labels to allow easy visual comprehension.

b, ARS comparing clusters identified by each condition with original subtype labels.

Finally, why initialize with PCA as opposed to the normalized, 100kb binned methylation values since the binning approach drastically reduces the total number of features per cell, which is often the goal of data compression with something like PCA?

R2.6.3

There are 27,269 100-kb bins in the mm10 genome, and even after filtering and highly variable feature selection, we still maintain ~3000 100-kb bins, many of which are highly correlated with each other. Therefore, as suggested by widely used single-cell analysis packages (Luecken and Theis, 2019; Satija et al., 2015; Wolf et al., 2018), we performed PCA to further reduce dimensions to reduce the “curse of dimensionality” that may occur in the nearest neighbor graph building (Wolf et al., 2018).

When answering R2.2.2, we ran UMAP and clustering directly on all the remaining features without doing PCA. As shown in Fig. R21, although in some mCH conditions (e.g. mCH of 100 kb bins), direct embedding on feature space gives comparable results; it does merge many clusters and gives less clear UMAP embeddings in most mCG conditions and some mCH conditions when bin size is smaller. These results indicate that running PCA before embedding and clustering is a more robust approach.

Fig. R21. UMAP embedding of the same datasets used in R2.2.2, with or without PCA.

- 1) The use of a neural network to simultaneously predict subtypes and dissection regions is great. However, the motivation or need to use a neural network for the prediction approach versus something more interpretable like a penalized regression model is not clear. This would allow a somewhat more direct interpretation of the contribution of individual features. Does the neural network outperform the use of somewhat "simpler" regression models?

R2.7

We benchmarked the neural network against random forest and logistic regression. All three models achieved >95% accuracy for subtype prediction (Fig. R22a, top). However,

our neural network model outperformed both random forest and logistic regression models to predict the dissected region. The performances increased by 16% and 7% respectively measured by exact accuracies (Fig. R22a, middle; only the exact region is considered correct prediction), or 6% and 4% measured by fuzzy accuracies (Fig. R22a, bottom; either the exact region or its anatomical neighbors is considered correct prediction; as described in Methods in the previous version of the manuscript). The increases in performance were observed for almost all of the major cell types (Fig. R22b). These results demonstrate the necessity of using the neural network for the prediction task, and this information has been added to the main text and Extended Data Fig. 11.

“The performance of ANN on subtype prediction is comparable with logistic regression (LR) and random forest (RF), while its performance on location prediction is substantially improved against the other two models (Extended Data Fig. 11b). This suggests that distinguishing the cells from different dissected regions would require non-linear relationships between genomic regions.”

Note that the major aim of Fig. 6 is to emphasize the predictability of detailed cell location in the brain by DNA methylation. In comparison, the cell-type-specific features, including gene and regulatory elements, were elaborately explored in Fig. 3-5 with simpler and statistical models. Thus, we decided to use the neural network in Fig. 6 which performed the best in these classification tasks.

Fig. R22. Performance of the neural network and baseline models. **a**, The overall exact accuracies (top and middle) or fuzzy accuracies (bottom) of neural network (NN), logistic regression (LR, multinomial), and random forest (RF, $n_{\text{estimator}}=1000$) for subtype prediction (top) and region prediction (middle and bottom). Scikit-learn was used for LR and RF implementation. **b**, **c**, Performance of region prediction in each major type. Each circle shows the exact (**b**) or fuzzy (**c**) accuracy of NN and the improved accuracy against LR (top) or RF (bottom) in a major type. The sizes of circles are proportional to the cell numbers of the major type, and the colors of circles represent different categories of major types.

6) DMRs: The total number of significant DMRs found is startling - using a permutation-based goodness of fit approach is intriguing, but the ability to control type I error is unclear with so many sites and cells/subtypes. This needs to be commented on to discuss the possible impact of the false discovery rate in such a large-scale analysis.

R2.8.1

We used methylpy 1.4.4 (<https://github.com/yupenghe/methylpy>) “DMRfind” to identify DMRs as we have previously described (He et al., 2020; Schultz et al., 2015). In brief, methylpy first tests differential methylated sites (DMS) and then merges DMS within 250

bp into DMRs. For the sequential-permutation-based (Besag and Clifford, 1991) P values of the goodness-of-fit statistic at DMS level, methylpy did implement a false discovery rate (FDR) control method designed to compare multiple sequential permutation-derived P values (Bancroft et al., 2013) to the desired rate of 1%.

To further estimate the empirical FDR at the DMR level, we shuffled the cell type labels and merged single-cell profiles according to the “fake cell types”. Then we called shuffle-DMRs using the same process. In the shuffled run, we identified 143,615 total shuffle-DMRs before applying any additional filters. Compared to 4,722,053 total subtype-DMRs when merged based on the real subtypes, the overall false discovery rate is 3.0%. After using the same robust-mean and blacklist filtering (see methods), the FDR is 2.7% (105,310 shuffle-DMRs and 3,947,795 subtype-DMRs).

We then divided the DMRs into different groups based on the number of DMSs and the effect size of the DMR, and computed the FDR within each group. For each DMR, the effect size was calculated by subtracting the minimum mCG fraction across samples from the robust mean (see methods) of samples. We then assigned both shuffle-DMR and subtype-DMR into joint bins of the number of DMSs and effect size ranges (Fig. R23a, b). Most (93%) of the shuffle-DMRs only have a single DMS (no other DMS within ± 250 bp), while this proportion decreases to 35% for subtype-DMRs. For effect size > 0.3 bins, the FDR for DMS = 1 bins range from 0.071 to 0.093. The FDR for the remaining bins having DMS > 1 is close to or well below 0.01 (Fig. R23c).

Fig. R23 Distribution of DMRs by effect size and number of DMSs. a, b, Portion of DMRs fall into corresponding effect size and # of DMS pairs blocks. **c,** empirical FDR of each block, calculated by (# of shuffle-DMRs / # of subtype-DMRs) in each block. Effect size < 0.3 is not used in analysis.

The remaining question is whether we should keep those 1,645,355 (35%) single-DMS DMRs. Using the single-DMS shuffle-DMR as background regions, we performed motif enrichment analysis on single-DMS subtype-DMRs (sDMRs) and multi-DMS subtype-DMRs (mDMRs). We found 108 / 174 motifs enriched in mDMRs are also enriched in sDMRs, and the odds ratio of all mDMR-enriched motifs are highly correlated (Fig. R24, Pearson's $r=0.86$, $P = 1e-53$). These results indicate that although the single-DMS DMRs are noisier than multi-DMS DMRs, they are still biologically relevant. Removing these single-DMS DMRs will improve the overall FDR from 3.0% to 0.3%, with the cost of reducing the power to identify true positives. We now provide the number of DMRs that remained for each subtype using different filtering criteria in Supplementary Table 12, while keeping the single-DMS DMRs in our genome-wide analysis. For prioritizing DMRs for further experimental purposes (such as enhancer-AAV testing (Hrvatín et al., 2019; Mich et al., 2020)), one could filter DMR by $DMS > 1$ and increase the effect size cutoff to select candidates with higher confidence.

Fig. R23 and Fig. R24 are added as Extended Data Fig. 7 and the discussion of FDR

controlling has been added to the Methods.

Fig. R24. The odds ratio of TF motif enrichment in single-DMS DMRs (sDMRs) and multi-DMS DMRs (mDMRs). Each dot represents a TF motif. The TF motifs significantly enriched in both sDMRs and mDMRs were colored in green, and significant TF motifs in only sDMR or mDMR are colored in orange or blue, respectively. Non-significant TFs are greyed.

Using randomly sampled groups there were 2,003 DMRs, as shown in Fig 5H, and removing them from the gradient DMR set was an interesting approach but the logic is not entirely clear. One would not expect the same false positives in both experiments, likely also the reason for the minimal overlap. What is the FDR rate (2,003/how many total)? Knowing the FDR rate and keeping the whole set is going to be more informative than removing a mere 0.04% overlap with DMRs from scrambled setup.

R2.8.2

We agree with the reviewer that keeping the whole set of DMRs and reporting the FDR would be more reasonable. The FDR of DG DMRs (related to Fig. 5h) is 0.8% (2,003 / 243,312). We have updated the figures and Methods to make this more clear. Given the tiny overlap between the random DMRs and real DMRs, the results are unchanged.

7) The use of gene-body CpH methylation as a surrogate for inverse expression levels, and correlating it with CpG DMRs as surrogate or regulatory elements is clever. However, gene-body CpG methylation is positively correlated with gene expression. Why not use the more abundant (and therefore higher signal-to-noise ratio) CpG methylation, rather than the noisier CpH methylation?

R2.9.1

We thank the reviewer for appreciating our correlation analyses to link enhancers and genes based on methylation levels. Here we systematically compare gene body mCH and mCG, and concluded that in neuronal cell types, 1) as we have previously published ((Luo et al., 2017; Mo et al., 2015)), both mCH and mCG are negatively correlated with gene expression, 2) mCH is a better surrogate for gene expression, and 3) the abundance and signal-to-noise ratio of mCH and mCG are comparable.

First, to evaluate the correlation between gene expression and the two types of methylation, we used our snmC2T-Seq data of human cortical neurons (Luo et al., 2019), where DNA methylation and RNA expression are simultaneously quantified in the same single nuclei. These analyses revealed that both mCG and mCH at gene bodies and promoters are anti-correlated with gene expression (Fig. R25a). This finding is consistent with our previous studies using brain cells (Luo et al., 2017; Mo et al., 2015), and in contrast to embryonic stem cells (Lister et al. 2009) and cancer cells (Yang et al., 2014) where mCG is positively correlated with gene expression. The results also indicated that the feature having the strongest anti-correlation with gene expression is mCH at gene-body, which is also consistent with our previous conclusion using bulk methylome in a limited number of purified cell types (Mo et al., 2015). As an example shown in the genome browser (Fig. R25b, *Cux2* as an IT-L23 marker), the cell-type-specific hypo CH methylation usually spans the whole gene body, while the cell-type-specific hypo CG methylation majorly locates in limited smaller regions (CG-DMRs) inside or surrounding the gene body. This illustrates why mCH has stronger correlations with RNA in many cell type marker genes.

Besides, many predicted enhancers are intragenic. Therefore, gene body mCG therefore contains the information of enhancer mCG, and leads to higher correlations with enhancer mCG intrinsically.

It is also worth noting that the total number of methylated CH sites is usually comparable with methylated CG sites in neuronal genomes (Fig. R25c). This is because the total number of CH sites is 30 times more than CG sites, although the methylation level is 20-85 fold higher for CG. Nevertheless, we acknowledge that the relative error of mCH detection could be higher than mCG (Fig. R25d). However, since we used cluster-level pseudo-bulk methylome (usually >10x coverage) in this analysis, the relative error of the mCH level estimation is smaller than 5% (signal/noise > 20) for more than 80% of genes in subtypes (Fig. R25d). Therefore, we consider the estimation to be reasonably accurate, and we remain confident to use mCH in this analysis as a surrogate of gene expression given their higher correlation.

To emphasize the correlation between gene expression and mCH/mCG, we have revised the main text as follow:

“A unique advantage of single-neuron methylome profiling is that it captures the information of both cell-type-specific gene expression and predicted regulatory elements. Specifically, **both gene body mCH and mCG are negatively correlated with gene expression in neurons, with mCH showing a stronger correlation than mCG** (Lister et al., 2013; Luo et al., 2017, 2019; Mo et al., 2015). CG-DMRs provide predictions about cell-type-specific regulatory elements and TFs whose motifs enriched in these CG-DMRs predict the crucial regulators of the cell type (He et al., 2020; Luo et al., 2017; Mo et al., 2015).”

Fig. R25. Comparison of mCH and mCG as surrogates of gene expression. **a**, The Pearson correlation coefficient between gene expression and mCH (blue) or mCG (orange) level at different genomic locations. Correlations were computed across 17 major cell types of the human prefrontal cortex for 2,154 differentially methylated genes (FDR < 1e-10, Kruskal-Wallis test). **b**, Genome browser view of a well-known cell type marker gene *Cux2* for IT-L2/3. CT-L6 subtypes are used as negative examples. Red arrows pointing to IT-L2/3 hypo-DMRs located inside the *Cux2* gene body. **c**, The proportion of mC in CH and CG context in each major type. The number of mC were averaged across cells of each major type and then normalized to the sum of 1. **d**, Histogram of estimated signal to noise ratio of gene body methylation (n=33,099x145) for all DMGs (n=33,099) in all subtypes (n=145), see methods noted below.

Methods of Fig. R25d: To calculate the signal-to-noise ratio on mC fraction calculation, assuming $m \sim Bi(cov, p)$, where p is the methylation level, the mean is $cov \cdot p$ or mc , and standard deviation is $\sqrt{cov \cdot p \cdot (1 - p)}$ or $\sqrt{mc(cov - mc)}$ for each gene in each subtype. The mean and standard deviation of binomial distribution were used to estimate the signal and noise respectively.

More importantly, CpH level is dependent on cell type and there are cell types with much lower level or no detectable CpHs. Gene body CpH levels in those cells are overall lower than those with high DNMT3A (and high global CpH), but not directly associated with host gene expression level difference.

R2.9.2

We agreed with the reviewer that the difference in the global mCH fraction between cell types will impact differential analysis (e.g. elevating the number of CH-DMG if using unnormalized mCH fraction). Therefore, in all of our analyses we have normalized the mCH fraction by the global mCH fraction of the cluster when comparing between clusters (e.g., in CH-DMG analysis). We improved the method section to make this important point more clear:

“We used the **single-cell level mCH fraction normalized by the global mCH level** (same as the “Computation and normalization of the methylation level” in the clustering step above) to calculate markers between all neuronal clusters. We compared non-neuron clusters separately using the **mCG fraction normalized by the global mCG level.**”

Also, it was stated that there were sn-RNA-seq data, and I feel those data could have been better used in this part of analysis - it could have been used for expression in this model, or if not, for assessing the suitability of gene body CpH and CpG methylation as surrogates for expression in this analysis, or as validation for discoveries.

R2.9.3

The only currently snRNA-seq data (Yao et al., 2020) covers cortex and hippocampus, which does not fully match the regions that we have profiled in our study. Nevertheless, we perform RNA-based correlation analyses only using cell types from these two tissue/age matched brain regions in order to add orthogonal validation to our methylation-based correlation analysis. Specifically, we integrated snmC-seq2 data from this study with snRNA-seq data, using the two matched brain regions. Similar to integration with snATAC-seq data, we separately carried out integration for each of the five major neuronal cell-type groups: 1) cortical IT and CLA neurons, 2) cortical ET neurons, 3) cortical CT, NP, and L6b neurons, 4) hippocampal excitatory neurons, and 5) cortical and hippocampal MGE and CGE inhibitory neurons. In general, all cell groups integrated well as the cluster structure from both datasets are maintained in the integrative UMAP embedding (Fig. R26a). Within each group, we used the overlap score to match transcriptome cell types to methylome cell types at the subtype level (Fig. R26b and Supplementary Table 11). In total, we were able to match 233 snRNA-seq cell types (deepest level in Yao et al.) with 90 snmC-seq2 neuronal subtypes. We used the matched cluster to calculate the RNA expression profile of the 90 methylation subtypes.

Fig. R26. Integration of snmC-seq2 with snRNA-seq data. **a**, Integration UMAP embeddings of five cell-type groups. Left column shows snRNA-seq cells colored by snRNA-seq cell types using the same palette as Yao et al. 2020, right column shows snmC-seq2 cells colored by subtype labels. **b**, Heatmap showing overlap score between snmC-seq2 subtypes (x-axis) and snRNA-seq cell type identified in Yao et al (y-axis).

We then compared eDMR-gene correlation pairs using three different types of gene information in these 90 methylation subtypes: 1-2) using mCH (1) and mCG (2) fraction of gene body ± 2 kb; 3) using RNA expression from integrated snRNA-seq data. First, all three different calculations generate similar null distributions (Fig. R27a), therefore, we

kept the correlation cutoff as described in our original manuscript (> 0.3 for positively correlated and < -0.3 for negatively correlated). The correlation calculated using RNA expression is reversed to match the correlation calculated using the two methylation types (Table R1).

eDMR Measure	Gene Measure	Correlation	Type
mCG	mCH, mCG	Positive	Activation
		Negative	Repression
	RNA	Positive	Repression
		Negative	Activation

Table R1. Correlation analysis groups

In total, we identified more activation pairs than repression pairs in all groups (Fig. R27b). The mCG group gives the highest number of correlated pairs; however, this elevation of correlation is expected since many eDMRs are intragenic and using mCG as the measurement in both eDMRs and genes contains overlapping information. We compared the mC-based pairs with RNA-based pairs, and observed 27% (24%) of the activation pairs based on mCH (mCG) overlapped with RNA-based activation pairs (Fig. R27c, e). The observed repression pairs based on mCH (mCG) overlapped with RNA-based repression pairs were 18% (17%) (Fig. R27d, f).

In addition to the overlap, we also plotted the correlation of eDMR-gene pairs based on different gene measurements (Fig. R28). Both mCH and mCG are negatively correlated with the RNA-based correlation (Fig. R28), indicating general agreement of the mC and RNA based analysis regardless of the cutoffs.

Together, the correlation between gene RNA expression and eDMR mCG fraction validates the correlation based on gene mCH/mCG fractions and provides an additional source of information that prioritize the most-likely active eDMR-gene interactions in adult mouse brain neuronal types.

Fig. R27. Overlap between RNA-based pairs with mC-based pairs. **a**, Null distribution of the eDMR-gene correlations, gray lines and numbers mark the portion of shuffled correlation exceeding the cutoff of -0.3 and 0.3. **b**, number of positive or negative pairs found in each group. **c**, **d**, portion of mC-based pairs that overlapped with RNA-based pairs. **e**, **f**, portion of RNA-based pairs that overlapped with mC-based pairs.

Fig. R28. Scatter plots showing all individual eDMR-gene pairs. a, b, comparing RNA-based (x-axis) eDMR-gene correlation with mCH (a) or mCG (b) based eDMR-gene correlation (y-axis). **c, d**, Zoom-in view of pairs having < -0.3 correlation in both groups. **e, f**, Zoom-in view of pairs having > 0.3 correlation in both groups.

9) The use of “rate” for methylation levels throughout the manuscript is confusing, since the word “rate” implies a time-dependent function. In most cases, the word “rate” could be replaced by the word “fraction”. I realize that the authors have used this terminology in past papers, but I am not enthusiastic about promoting this usage. Scientists studying other types of genomic events have moved away from the use of the word “rate” and tend to use words like “density”, “frequency”, or “load”, as in genomic mutational density per Mb, as opposed to mutation rate, which would instead be per cell division or time unit.

R2.10

We agreed with the reviewer and have changed the all related terms into “methylation fraction” or “methylation level” throughout the manuscript.

10) sn-m3C-seq: R1 and R2 reads were mapped separately, and they then claimed a higher mapping rate than previous methods. However, mapping separately would achieve that.

R2.11.1

It is standard in Hi-C data processing pipelines to separately map the paired-end reads to the genome and re-pair them after quality control (alternatively, parameters can be used to “skip-pairing” in alignment where read 1 and 2 are mapped independently, such as in BWA-MEM “-SP5M” based alignment). In our previous Nature Methods paper where snm3C-Seq was developed (Lee et al., 2019), we demonstrated that we could achieve increased mapping rates by splitting un-aligned reads, compared with alternative methods that either did not split unmapped reads or that used aligners that attempt to split (i.e. soft or hard clip) reads during alignment (i.e. bwa-meth).

They split up the unmapped reads into the first 40bp, middle segment and final 40bp, likely to deal with the 3C part. There may not be that many options with Bismark, but HiC data processing methods may reveal smarter ways of doing this than just the 40bp split - e.g. searching for a ligation site, etc.

R2.11.2

The reviewer is correct that alternative mapping strategies for Hi-C data have been developed, including using aligners that split reads by soft- or hard-clipping (BWA-MEM) or where potential ligation junctions can be identified due to the sequence at the ligation site. In our previous Nature Methods paper where snm3C-Seq was developed (Lee et al., 2019), we tested read both using split-read aligners (bwa-meth) and splitting based on restriction enzyme cutting sites, and observed improved alignment accuracy and rates using our manual read splitting approach. In terms of splitting reads based on the ligation site, the reason this is less useful in our snm3C-seq method (as compared to in standard Hi-C experiments) is that we do not fill-in ends of DNA with nucleotides to label DNA ends with biotin. As a result, the ligation junctions are formed with sticky ends. As a 4-base NlaIII cutter is used for digestion, the ligation site motif is only 4 base in length, CATG. After bisulfite conversion (C to T and G to A), there is additional ambiguity that is introduced (for example, the motif could be present as CATG or TATG as a result of the presence or absence of cytosine methylation). As a result, the motif becomes very frequent in the converted genome. This is problematic as 1) it will lead to many “false positive” ligation sites being identified if considering all the possible converted or unconverted motifs, and 2) read splitting splits into small fragments leading to reduced

mapping qualities. This is in contrast to a normal Hi-C protocol, where blunt ends were used and no bisulfite conversion was performed. The motif in this case will be CATGCATG and the reads are mapped to the normal genome, which considerably reduces the frequency.

It is not clear how they dealt with chimeric reads generated by Klenow to distinguish them from chromatin contacts.

R2.11.3

Based on our previous experiments, Klenow usually generated <2% fragments that consisted of two parts mapping to two genomic loci > 1 kb apart from each other. Compared to the snm3C-Seq data, this number is 22%. We therefore reasonably estimate the false discovery rate of cis contacts is less than 10%. Additionally, the enrichment of cell-type-specific genes and enhancers at the loop anchors (Fig. 4) also supports the high quality of the chromatin contact data.

11) Controls: The authors should be commended for including controls for most analyses. However, sometimes the control did not seem to be used properly (the Figure 5h example above),

R2.12.1

Please see R2.8 for Figure 5h related response.

or the results were not impressive compared to controls. For example, in Figure 4k, the observed data (22-23 is not a very high level of enrichment compared to random (16.5). Similar with 4h. Statistical significance is one thing and easy to achieve when there are so many data points in the genome, but if the effect size is not that big it calls into question how good the algorithm actually is.

R2.12.2

We agree with the reviewer that statistical significance is easier to achieve when the sample size is larger, which does not equate to large absolute differences or fold-change. Accordingly, we have modified the statement in the text to describe the moderate but significant differences. We note that statistical tests consider the variance of the null distribution, which is also an important aspect in addition to the absolute differences. For example in Fig. 4k, although the means are not different by more than a few fold, the variability of random distribution is small, which leads to a large enrichment (Z-score ~ 60).

In Fig. 4h, the blue and orange colors correspond to the different y-axes, as shown on the left and right, representing the number of gene-enhancer pairs located on separate anchors of the same loop, or within the same loop, respectively. Therefore, we intended

to show the trend of increasing numbers of gene-enhancer pairs supported by chromatin contacts (y-axis) with the increasing of gene mCH and enhancer mCG correlation (x-axis), rather than comparing the two lines in the same subpanel. We have added more information into the figure legend to make this more clear.

Note that all analyses are presenting the patterns that we observed from the data using well-established algorithms, rather than determine the performance of these algorithms. The algorithms used in these panels involve standard processing and loop calling of Hi-C data, and standard correlation analysis between gene and enhancer methylation. The REPTILE enhancer calling is also published and benchmarked (He et al., 2017; Sethi et al., 2020).

Sometimes controls are not included when they could. e.g., mCCC plots as controls for bisulfite conversion in a lot of the mCH analyses.

R2.12.3

The mCCC fraction is used as the proxy of the upper bound of the non-conversion rate for cell-level QC. However, mCCC contains biological signals and are not due to failed conversion. In the cells that passed QC filters, we believe it was not appropriate to use mCCC as quantitative controls in the other computational analyses (see discussion below). We added additional clarification of this information in Methods.

In all of our experiments, every well contains a spike-in of unmethylated lambda DNA as non-conversion control. The lambda mC fraction at each library preparation batch is below 1.5%. As an additional ‘metric’ for a more complex mammalian genomic, we monitor the methylation level at CCC sites as it is the lowest among all of the different 3 base-contexts, and, in fact, it is very close to the unmethylated lambda mC fraction (Fig. R29a). Therefore, in addition to lambda, we use mCCC fraction as the single-cell level QC metric for non-conversion fraction in mammalian genomic DNA.

However, mCCC also positively correlates with mCH at the single-cell level (Pearson’s $r=0.53$, $P = 1e-300$). Such a high correlation does not exist between lambda mC and mCCC or mCH (Fig. R29b), indicating that the mCCC fraction, although being very low, is also impacted by the methyltransferase expression (*Dnmt3a*). In addition, using “PAL-Inh Chat” as an example, we also discussed in R2.3.2 that some fraction of mCCC may represent real signals. Thus, mCCC should not be treated as “pure noise” and we have not used mCCC as a control for any quantitative computational analyses such as the eDMR-gene correlation analysis.

Fig. R29. Comparison of unmethylated lambda DNA mC, mCCC and mCH. **a**, Lambda mC fraction, mCCC fraction, mCH fraction distribution in all cells with lambda DNA reads (n=59,720). **b**, Pearson correlation between three types of methylation fractions using all cells in (a).

Minor comments:

1) Lines 134-136 about using UMAP to visualize differences based on cell location is unclear given that UMAP optimizes a cost function for both local and global distances rendering "location" somewhat confounded with the general classification of different brain regions. Were global distances weighted in the UMAP embeddings to account for spatial distance?

R2.13

We agree with the reviewer. The UMAP coordinates were computed only based on DNA methylation, and the cells were then colored based on their spatial localization for visualization. Thus, the locations of different colors on the UMAP represent the similarities of DNA methylation between the cells located in different regions. Therefore, we have now changed "based on" to "between".

2) Line 152 suggesting high RNA expression of *Unc5c* given the low methylation - this needs to be evaluated in terms of actual gene expression from the Allen Brain Institute scRNA-seq brain atlas.

R2.14

We plotted *Unc5c* gene mCH fraction and RNA expression (from AIBS snRNA-seq) using the integrated UMAP embedding from R2.9.3 (Fig. R31). Note that *Unc5c* gene has lowest mCH and highest expression in the “Unc5c” cell types, but it is also expressed in a few other cell types. This cluster corresponds to the *Lamp5*⁺/*Lhx6*⁺ type in the snRNA-seq study (Yao et al., 2020). We also plotted *Lamp5* and *Lhx6* below (Fig. R31). All three genes have corresponding expression and methylation patterns, which clearly indicates this *Unc5c* cell type is found in both modalities.

Fig. R30. Hypo-mCH in the gene body corresponding to high level RNA expression. **a, b**, integration UMAP showing snmC-seq2 and snRNA-seq cell types. **c-h**, Normalized gene body mCH fraction (first row) and RNA expression level (second row, from snRNA-seq) of *Unc5c* (c, d), *Lamp5* (e, f), and *Lhx6* (g, h) that marked the “Unc5c” cell type.

3) Line 243, how was 0.3 chosen as a reasonable threshold for impact score and thus, gene assignment to each branch in the tree?

R2.15

The threshold of 0.3 was empirically chosen. At each node in the dendrogram, all DMGs or differential enriched motifs were assigned with impact scores. With the cutoff of 0.3, 71% of DMGs (75% of differential enriched motifs) on average were selected to represent the differential features between the two branches of the node.

4) Line 768, the word "significant" is used without any association with a p-value or other significance value.

R2.16

We agreed with the reviewer and have now removed the word “significant” here. We also discussed the PC selection in R2.6.

5) Some of the results were puzzling and not explained very well. For example, why would CpH methylation correlate with # of intra-domain contacts? Does the same trend show up if mCCCs are plotted instead of CpH?

R2.17.1

As we have discussed in the results, this observation might indicate the local chromosome architecture is more compact in mature cells where global mCH is higher.

This trend is also true in mCCC. However, as discussed in R2.12.3, mCCC is only used for a quality control metric, and is not considered as an exact quantitative measurement of the non-conversion rate.

Or in Fig. 4h, why would the number of enhancer-gene pairs differ so much between the two cell types?

R2.17.2

The numbers of enhancer-gene pairs are similar in these two cell types. However, the numbers of loops identified in the two cell types are different. In snm3C-Seq data, we have 1,933 cells labeled as DG while 686 cells are labeled as CA1. Thus, the coverage of DG contact map is 3-fold more than CA1, which leads to twice as many loops being called by HICCUPS in DG compared with CA1, and more longer-range loops were also identified. Therefore, the numbers of gene-enhancer pairs that overlap or within loops were different, as shown in Fig. 4h. To make it more explicit, we have added the following discussion in the Methods.

“Note that the abundance of cell types is highly variable, leading to different coverages of contact maps after merging all the cells from each cell type. Since *HICCUPS* loop calling is sensitive to the coverage, the more loops were identified in the abundant cell types (e.g. 12,614 loops were called in DG, containing 1933 cells) compared to the less abundant ones (e.g. 1,173 loops were called in MGE, containing 145 cells). Therefore, we do not compare the feature counts related to the loops across cell type directly in the following analyses.”

Or, in Fig 3i and 3k - why are the TF motifs different for CpH-DMGs and CpG-DMRs for the same cluster?

R2.17.3

Fig. 3h (3i in the original version) is based on CH-DMG analysis of all TF genes, while Fig. 3j (3k) is based on motif enrichment analysis in CG-DMRs, computed for a subset of TFs (719 / 1309 TFs). Many TF genes in Fig. 3h do not have known motifs (e.g.,

Zfp462, Hivep1) in the JASPAR2020 database (Fornes et al., 2020) due to lack of ChIP-Seq/SELEX data. Besides, some TFs (e.g., Fos, Jun) show universally hypo-mCH in their gene bodies across all cell types. These TFs were not identified in CH-DMG analysis, but their motifs were differentially enriched in CG-DMRs of different cell types.

In summary, although most of the examples shown in the results are consistent between gene body and regulatory motif analyses, we still acknowledge that the two types of analyses represent different information and are expected to have different results. The TFs that are the top candidates in both analyses represent those with the strongest evidence of cell-type specificity.

6) In figure 3A, 2 of the 3 CG-DMRs for BCL11b seemed to be at about 50% methylated in cell type B. This seems to suggest cell type B is still heterogeneous (or this gene is monoallelicly methylated). Since this is single cell data, can the authors look within B and see if it was indeed a mixture of methylated and unmethylated cells in B, or if it was monoallelic methylation.

R2.18

Since the coverage of our data is around 6% of the genome for every single cell, a single cytosine site is only covered by one read in most of the cells. Therefore, the partial methylation of a DMR is due to a mixture of methylated and unmethylated cells in that cell type. Besides, we can not determine alleles from the pseudo bulk data, because our data were collected from inbred BL6 mice, where most alleles show little sequence variation.

7) In figure 2, m/n/p panel colors - do these stand for subtype assignments? Many colors are so similar that it is hard to assess the UMAPs

R2.19

The reviewer is correct, colors in Fig. 2m,n,p indicate cell subtypes. The original color was intended to be consistent with other figures and Li et al (Li et al., 2020). We have now updated the figure with more distinguishable colors.

Reference

Amemiya, H.M., Kundaje, A., and Boyle, A.P. (2019). The ENCODE Blacklist: Identification of Problematic Regions of the Genome. *Sci. Rep.* 9, 9354.

Bancroft, T., Du, C., and Nettleton, D. (2013). Estimation of false discovery rate using sequential permutation p-values. *Biometrics* 69, 1–7.

Bayraktar, O.A., Fuentealba, L.C., Alvarez-Buylla, A., and Rowitch, D.H. (2014). Astrocyte development and heterogeneity. *Cold Spring Harb. Perspect. Biol.* 7, a020362.

Besag, J., and Clifford, P. (1991). Sequential Monte Carlo p-Values. *Biometrika* 78, 301–304.

Floriddia, E.M., Zhang, S., and van Bruggen, D. (2019). Distinct oligodendrocyte populations have different spatial distributions and injury-specific responses. *bioRxiv*.

Fornes, O., Castro-Mondragon, J.A., Khan, A., van der Lee, R., Zhang, X., Richmond, P.A., Modi, B.P., Correard, S., Gheorghe, M., Baranašić, D., et al. (2020). JASPAR 2020: update of the open-access database of transcription factor binding profiles. *Nucleic Acids Res.* 48, D87–D92.

Geller, E., Gockley, J., Emera, D., Uebbing, S., Cotney, J., and Noonan, J.P. (2019). Massively parallel disruption of enhancers active during human corticogenesis.

Habib, N., Li, Y., Heidenreich, M., Swiech, L., Avraham-Davidi, I., Trombetta, J.J., Hession, C., Zhang, F., and Regev, A. (2016). Div-Seq: Single-nucleus RNA-Seq reveals dynamics of rare adult newborn neurons. *Science* 353, 925–928.

He, Y., Gorkin, D.U., Dickel, D.E., Nery, J.R., Castanon, R.G., Lee, A.Y., Shen, Y., Visel, A., Pennacchio, L.A., Ren, B., et al. (2017). Improved regulatory element prediction based on tissue-specific local epigenomic signatures. *Proc. Natl. Acad. Sci. U. S. A.* 114, E1633–E1640.

He, Y., Hariharan, M., Gorkin, D.U., Dickel, D.E., Luo, C., Castanon, R.G., Nery, J.R., Lee, A.Y., Zhao, Y., Huang, H., et al. (2020). Spatiotemporal DNA methylome dynamics of the developing mouse fetus. *Nature* 583, 752–759.

Hrvatin, S., Tzeng, C.P., Aurel Nagy, M., Stroud, H., Koutsoumpa, C., Wilcox, O.F., Griffith,

E.C., and Greenberg, M.E. (2019). PESCA: A scalable platform for the development of cell-type-specific viral drivers.

Huang, Y., Pastor, W.A., Shen, Y., Tahiliani, M., Liu, D.R., and Rao, A. (2010). The behaviour of 5-hydroxymethylcytosine in bisulfite sequencing. *PLoS One* 5, e8888.

Hubert, L., and Arabie, P. (1985). Comparing partitions. *J. Classification* 2, 193–218.

Khare, T., Pai, S., Koncevicius, K., Pal, M., Kriukiene, E., Liutkeviciute, Z., Irimia, M., Jia, P., Ptak, C., Xia, M., et al. (2012). 5-hmC in the brain is abundant in synaptic genes and shows differences at the exon-intron boundary. *Nat. Struct. Mol. Biol.* 19, 1037–1043.

Krienen, F.M., Goldman, M., Zhang, Q., del Rosario, R., Florio, M., Machold, R., Saunders, A., Levandowski, K., Zaniewski, H., Schuman, B., et al. (2019). Innovations in Primate Interneuron Repertoire.

Krueger, F., and Andrews, S.R. (2011). Bismark: a flexible aligner and methylation caller for Bisulfite-Seq applications. *Bioinformatics* 27, 1571–1572.

Lee, D.-S., Luo, C., Zhou, J., Chandran, S., Rivkin, A., Bartlett, A., Nery, J.R., Fitzpatrick, C., O'Connor, C., Dixon, J.R., et al. (2019). Simultaneous profiling of 3D genome structure and DNA methylation in single human cells. *Nat. Methods* *16*, 999–1006.

Lein, E.S., Hawrylycz, M.J., Ao, N., Ayres, M., Bensinger, A., Bernard, A., Boe, A.F., Boguski, M.S., Brockway, K.S., Byrnes, E.J., et al. (2007). Genome-wide atlas of gene expression in the adult mouse brain. *Nature* *445*, 168–176.

Li, Y.E., Preissl, S., Hou, X., Zhang, Z., Zhang, K., Fang, R., Qiu, Y., Poirion, O., Li, B., Liu, H., et al. (2020). An Atlas of Gene Regulatory Elements in Adult Mouse Cerebrum.

Lister, R., Mukamel, E.A., Nery, J.R., Urich, M., Puddifoot, C.A., Johnson, N.D., Lucero, J., Huang, Y., Dwork, A.J., Schultz, M.D., et al. (2013). Global epigenomic reconfiguration during mammalian brain development. *Science* *341*, 1237905.

Luecken, M.D., and Theis, F.J. (2019). Current best practices in single-cell RNA-seq analysis: a tutorial. *Mol. Syst. Biol.* *15*, e8746.

Luo, C., Keown, C.L., Kurihara, L., Zhou, J., He, Y., Li, J., Castanon, R., Lucero, J., Nery, J.R., Sandoval, J.P., et al. (2017). Single-cell methylomes identify neuronal subtypes and regulatory elements in mammalian cortex. *Science* *357*, 600–604.

Luo, C., Rivkin, A., Zhou, J., Sandoval, J.P., Kurihara, L., Lucero, J., Castanon, R., Nery, J.R.,

Pinto-Duarte, A., Bui, B., et al. (2018). Robust single-cell DNA methylome profiling with snmC-seq2. *Nat. Commun.* *9*, 3824.

Luo, C., Liu, H., Xie, F., Armand, E.J., Siletti, K., Bakken, T.E., Fang, R., Doyle, W.I., Hodge, R.D., Hu, L., et al. (2019). Single nucleus multi-omics links human cortical cell regulatory genome diversity to disease risk variants.

Mich, J.K., Graybuck, L.T., Hess, E.E., Mahoney, J.T., Kojima, Y., Ding, Y., Somasundaram, S.,

Miller, J.A., Weed, N., Omstead, V., et al. (2020). Functional enhancer elements drive subclass-selective expression from mouse to primate neocortex.

Ming, G.-L., and Song, H. (2011). Adult neurogenesis in the mammalian brain: significant answers and significant questions. *Neuron* *70*, 687–702.

Miura, F., Enomoto, Y., Dairiki, R., and Ito, T. (2012). Amplification-free whole-genome bisulfite sequencing by post-bisulfite adaptor tagging. *Nucleic Acids Res.* *40*, e136.

Mo, A., Mukamel, E.A., Davis, F.P., Luo, C., Henry, G.L., Picard, S., Urich, M.A., Nery, J.R., Sejnowski, T.J., Lister, R., et al. (2015). Epigenomic Signatures of Neuronal Diversity in the Mammalian Brain. *Neuron* *86*, 1369–1384.

Mulqueen, R.M., Pokholok, D., Norberg, S.J., Torkency, K.A., Fields, A.J., Sun, D., Sinnamon, J.R., Shendure, J., Trapnell, C., O'Roak, B.J., et al. (2018). Highly scalable generation of DNA methylation profiles in single cells. *Nat. Biotechnol.* *36*, 428–431.

Niewiadomska-Cimicka, A., Krzyżosiak, A., Ye, T., Podleśny-Drabiniok, A., Dembélé, D., Dollé, P., and Krężel, W. (2017). Genome-wide Analysis of RAR β Transcriptional Targets in Mouse Striatum Links Retinoic Acid Signaling with Huntington's Disease and Other Neurodegenerative Disorders. *Mol. Neurobiol.* *54*, 3859–3878.

Osterwalder, M., Barozzi, I., Tissières, V., Fukuda-Yuzawa, Y., Mannion, B.J., Afzal, S.Y., Lee, E.A., Zhu, Y., Plajzer-Frick, I., Pickle, C.S., et al. (2018). Enhancer redundancy provides phenotypic robustness in mammalian development. *Nature* *554*, 239–243.

Roadmap Epigenomics Consortium, Kundaje, A., Meuleman, W., Ernst, J., Bilenky, M., Yen, A., Heravi-Moussavi, A., Kheradpour, P., Zhang, Z., Wang, J., et al. (2015). Integrative analysis of 111 reference human epigenomes. *Nature* *518*, 317–330.

Satija, R., Farrell, J.A., Gennert, D., Schier, A.F., and Regev, A. (2015). Spatial reconstruction of single-cell gene expression data. *Nat. Biotechnol.* *33*, 495–502.

Schultz, M.D., He, Y., Whitaker, J.W., Hariharan, M., Mukamel, E.A., Leung, D., Rajagopal, N., Nery, J.R., Urich, M.A., Chen, H., et al. (2015). Human body epigenome maps reveal noncanonical DNA methylation variation. *Nature* *523*, 212–216.

Sethi, A., Gu, M., Gumusgoz, E., Chan, L., Yan, K.-K., Rozowsky, J., Barozzi, I., Afzal, V., Akiyama, J.A., Plajzer-Frick, I., et al. (2020). Supervised enhancer prediction with epigenetic pattern recognition and targeted validation. *Nat. Methods* *17*, 807–814.

Smallwood, S.A., Lee, H.J., Angermueller, C., Krueger, F., Saadeh, H., Peat, J., Andrews, S.R., Stegle, O., Reik, W., and Kelsey, G. (2014). Single-cell genome-wide bisulfite sequencing for assessing epigenetic heterogeneity. *Nat. Methods* *11*, 817–820.

Szulwach, K.E., Li, X., Li, Y., Song, C.-X., Wu, H., Dai, Q., Irier, H., Upadhyay, A.K., Gearing, M., Levey, A.I., et al. (2011). 5-hmC-mediated epigenetic dynamics during postnatal neurodevelopment and aging. *Nat. Neurosci.* *14*, 1607–1616.

Tasic, B., Yao, Z., Graybuck, L.T., Smith, K.A., Nguyen, T.N., Bertagnolli, D., Goldy, J., Garren, E., Economo, M.N., Viswanathan, S., et al. (2018). Shared and distinct transcriptomic cell types across neocortical areas. *Nature* *563*, 72–78.

Wolf, F.A., Angerer, P., and Theis, F.J. (2018). SCANPY: large-scale single-cell gene expression data analysis. *Genome Biol.* *19*, 15.

Yang, X., Han, H., De Carvalho, D.D., Lay, F.D., Jones, P.A., and Liang, G. (2014). Gene body methylation can alter gene expression and is a therapeutic target in cancer. *Cancer Cell* *26*, 577–590.

Yao, Z., Nguyen, T.N., van Velthoven, C.T.J., Goldy, J., Seden-Cortes, A.E., Baftizadeh, F., Bertagnolli, D., Casper, T., Crichton, K., Ding, S.-L., et al. (2020). A taxonomy of transcriptomic cell types across the isocortex and hippocampal formation.

Zeisel, A., Hochgerner, H., Lönnerberg, P., Johnsson, A., Memic, F., van der Zwan, J., Häring, M., Braun, E., Borm, L.E., La Manno, G., et al. (2018). Molecular Architecture of the Mouse Nervous System. *Cell* *174*, 999–1014.e22.

Zetterström, R.H., Lindqvist, E., Mata de Urquiza, A., Tomac, A., Eriksson, U., Perlmann, T., and Olson, L. (1999). Role of retinoids in the CNS: differential expression of retinoid binding proteins and receptors and evidence for presence of retinoic acid. *Eur. J. Neurosci.* *11*, 407–416.

Zhang, T.-Y., Keown, C.L., Wen, X., Li, J., Vousden, D.A., Anacker, C., Bhattacharyya, U., Ryan, R., Diorio, J., O'Toole, N., et al. (2018). Environmental enrichment increases transcriptional and epigenetic differentiation between mouse dorsal and ventral dentate gyrus. *Nat. Commun.* *9*, 298.

Reviewer Reports on the First Revision:

Referees' comments:

Referee #1 (Remarks to the Author):

The authors have been very responsive and addressed most of my previous concerns. For the rest they made arguments, which are reasonable to me. The authors can try to incorporate many of the figures made for reviewers into supplementary figures in the manuscript if space allows. This work will be an important resource for the field and the authors' effort to make it more accessible to readers is great.

Author Rebuttals to First Revision:

N/A